# Oxidative stress and GPX2 control pancreatic vs. non-pancreatic cell fate in human endoderm

Joanna Szpotkowska [1], Wojciech J. Szlachcic [1], Katarzyna Blaszczyk[1], Maja Baginska[1], Magdalena Socha [1] & Malgorzata Borowiak [1,2] ✉

Cell fate decisions in human endoderm development are tightly regulated, yet the role of metabolic products remains elusive. The endodermal posterior foregut gives rise to pancreas, liver, and intestine. Here, we identify Glutathione Peroxidase 2 as a critical regulator of human posterior foregut differentiation, revealing oxidative stress as a key determinant of pancreatic versus non-pancreatic cell fate. Cells lacking Glutathione Peroxidase 2 under pancreas-promoting conditions differentiate also into hepatic-like progenitors. Through bulk and single-cell transcriptomics, chromatin accessibility profiling, and functional studies, we reveal that Glutathione Peroxidase 2 orchestrates lineage commitment by regulating key transcription factors, leading to emergence of multilineage liver and intestinal progenitors. Mechanistically, Glutathione Peroxidase 2 deficiency triggers extracellular matrix remodeling, activating bone morphogenetic protein signaling and skewing differentiation from the pancreatic lineage. Manipulating oxidative stress recapitulates or rescues Glutathione Peroxidase 2 loss effects, establishing oxidative stress as a gatekeeper of pancreatic fate. Controlling oxidative stress during in vitro differentiation could advance regenerative medicine applications.

Organogenesis relies on the precise specification of progenitor cell populations into distinct tissues. The developmental trajectories of the pancreas, liver, and duodenum exemplify the intricate lineage relationships and inherent cellular plasticity characterizing this process. Evidence from human genetic syndromes substantiates the concept of plasticity, given the congenital malformations manifesting as multi-organ phenotypes, affecting these related organs[1]. Alterations in cellular plasticity can cause aberrant differentiation and potentially increase susceptibility to malignant transformation.

Recent advances in single-cell transcriptomic analysis have revealed the dynamic nature of multipotent progenitors throughout foregut development. Experimental manipulations of developmental signals in mouse embryos and, importantly, directed differentiation of human pluripotent stem cells (hPSCs) in vitro, have demonstrated the significant plasticity within foregut endoderm territory, enabling alterations in cell fate[2,3]. Comprehensive CRISPR-Cas9 genomic screens and single-cell transcriptomics have elucidated the transcriptional programs governing lineage commitment and plasticity restriction during hPSC-derived endoderm differentiation[4–7]. However, despite extensive investigations into transcriptional networks, signaling pathways, and cellular interactions, the precise molecular mechanisms, including the contribution of metabolism and free radicals, driving lineage diversification from foregut endoderm, particularly in hPSC differentiation, remain incompletely characterized.

The broad expression of pioneering transcription factors, such as GATA Binding Protein 4 (GATA4), GATA Binding Protein 6 (GATA6), and Forkhead Box A2 (FOXA2), across endoderm domains suggests the involvement of additional mechanisms in lineage segregation.

[1]Institute of Molecular Biology and Biotechnology, Adam Mickiewicz University, Uniwersytetu Poznanskiego 6, Poznan, Poland. [2]McNair Medical Institute and Baylor College of Medicine, One Baylor Plaza, Houston, TX, USA. ✉e-mail: malbor3@amu.edu.pl

Comprehensive analyses of single-cell RNA-seq datasets across developmental stages have revealed dynamic alterations in genes associated with oxidative phosphorylation, oxidative stress response, glutathione metabolism, and lipid peroxidation at critical lineage bifurcation points[8,9]. This observation prompted our investigation into the role of cellular metabolic status, with particular emphasis on oxidative stress during human endoderm specification.

Oxidative stress emerges from a disruption in the equilibrium between reactive oxygen species (ROS) production and cellular antioxidant defense mechanisms. ROS, encompassing free radicals and peroxides, are natural byproducts of aerobic metabolic processes. Historically, ROS were characterized as deleterious agents contributing to cellular damage, aging, and pathogenesis[10]. Contemporary research has nuanced this perspective by recognizing the critical regulatory roles of physiological ROS levels in developmental processes, cell signaling, and cellular homeostasis. Emerging evidence demonstrates the multifaceted involvement of ROS in diverse developmental contexts. In *Drosophila*, ROS orchestrate meiotic progression and early embryonic divisions[11,12]. Similarly, in developing sea urchin embryos, ROS modulate Nodal gradient activation, secondary axis establishment, and ectoderm differentiation[13,14]. Zebrafish studies further illuminate ROS functionality, with hydrogen peroxide ($H_2O_2$) mediating neural development through regulation of the Engrailed 2 gradient[15].

The endoderm, situated within the developing embryo and generating the metabolically active organs, is highly susceptible to ROS fluctuations. Studies in mice have shown that aberrant ROS levels, caused by antioxidant enzyme SOD1 deficiency, lead to abnormalities in gut tube development[16]. Despite accumulating evidence of ROS developmental significance, the mechanistic functions of ROS in human endoderm development, particularly in the context of hPSC differentiation into clinically relevant cell types, remain largely unexplored.

To address this gap, we investigated the impact of oxidative stress on pancreatic versus non-pancreatic cell fate decisions during in vitro human endoderm differentiation. By examining key oxidative stress response components, we specifically targeted selenoprotein Glutathione Peroxidase 2 (GPX2): an antioxidant enzyme critical for $H_2O_2$ neutralization and characterized by heightened expression during endoderm specification. Our findings reveal an unexpected role for GPX2 in modulating pancreatic and hepatic cell fate decisions and provide insights into the intricate interplay between oxidative stress and cell fate determination during hPSC differentiation into endoderm derivatives, with emphasis on pancreatic endocrine cell development.

## Results

### Oxidative stress and GPX2 in developing β cells

To investigate the role of oxidative stress in β-cell development, we utilized an hPSC pancreatic differentiation using our well-established differentiation protocol[17,18]. Differentiation efficiency was assessed across different stages using immunofluorescence (IF) and quantitative imaging or flow cytometry (FC). We consistently obtained 90–98% of SRY-box Transcription Factor 17 (SOX17)$^+$ and FOXA2$^+$ definitive endoderm (DE) cells at day 4, 75% of cells expressing a posterior foregut (PFG) marker Hematopoietically Expressed Homeobox (HHEX) at day 6, 80–90% pancreatic and duodenal homeobox 1 (PDX1)$^+$ and SRY-Box Transcription Factor 9 (SOX9)$^+$ pancreatic endoderm (PEN), 68–80% pancreatic progenitors (PP) co-expressing PDX1 and NK6 Homeobox 1 (NKX6-1)-at day 12, 65% of Chromogranin-A (CHGA)$^+$ and PDX1$^+$ endocrine progenitors (EP) and 30–50% β cells co-expressing C-peptide (C-PEP) and NKX6-1 (Fig. 1A, Supplementary Fig. 1A, B). We then quantified the ROS levels utilizing a fluorescence-based sensor, $H_2DCFDA$, at seven time points corresponding to the consecutive developmental stages, which revealed a progressive increase in ROS levels throughout the differentiation (Fig. 1B). At the

SC-β cell stage, ROS levels were nearly three times higher than at the initial hPSC stage. Consistent with these in vitro observations, the expression of genes associated with the oxidative stress response pathway increased in developing human fetal pancreas from 4 to 11 post-conception weeks (PCW) (Supplementary Fig. 1C). This significant increase in ROS levels suggested a potential role for oxidative stress in β cell development.

Cellular responses to oxidative stress are primarily mediated by antioxidant enzymes from the superoxide dismutase (SOD), peroxiredoxin (PRDX), and glutathione peroxidase (GPX) families (Fig. 1C). SOD converts superoxide radicals ($O_2^•$) into oxygen ($O_2$) and hydrogen peroxide ($H_2O_2$). Subsequently, GPX neutralizes $H_2O_2$ to $H_2O$ and $O_2$ through glutathione oxidation. PRDX also neutralizes $H_2O_2$ to $H_2O$ via a cysteine-dependent oxidation-reduction reaction. Catalase (CAT) is another antioxidant enzyme responsible for $H_2O_2$ degradation to $H_2O$ and $O_2$. We analyzed the mRNA expression of GPX family enzymes (*GPX1-8*), PRDX family members (*PRDX1-6*), and SOD family members (*SOD1-3*) across the hPSC, DE, and PP stages using our RNA-seq data (Fig. 1D). Expression levels of all antioxidative enzyme transcripts remained relatively stable between the hPSC and DE stages. However, the expression of *GPX2* and *SOD3* increased significantly at the PP stage, contrasting with the unchanged expression of other enzymes. *CAT* transcripts were not detected in our RNA-seq data.

To investigate the effect of oxidative stress on pancreatic differentiation, we focused on GPX2, as it functions downstream of SOD. Single-cell analysis of mouse gastrulation and early organogenesis[19] revealed remarkably high *Gpx2* expression in DE and gut at embryonic days (E) E6.5-8.5 (Fig. 1E). Additionally, *Gpx2* expression was prominent in bipotent progenitors (BP) and Neurogenin3 (*Ngn3*)-expressing early endocrine progenitors (EP-Ngn3) in the E14.5 pancreas[20] at the time point of extensive morphogenesis and cell adhesion remodeling (Fig. 1F). Given that DE contributes to the development of multiple organs within gut tube, and BPs differentiate into either endocrine or ductal cells, while *Ngn3*$^+$ EPs give rise to pancreatic endocrine cells, these findings suggest a potential role for GPX2 in the determination of endoderm or pancreatic progenitor cell fate.

Next, we examined the GPX2 expression during human pancreatic development. In the fetal pancreas at 10 and 13 PCW, GPX2 protein was expressed in pancreatic cells, including PDX1$^+$ cells (Fig. 1G). Transcriptomic analysis of the human fetal pancreas further revealed that *GPX2* mRNA is detectable even at the earlier stages, peaking at 6 PCW (Fig. 1H). Finally, we examined GPX2 protein expression during hPSC differentiation into pancreatic β cells. GPX2 expression was first detected at the PFG stage and increased progressively through the PEN, PP, and EP stages, but declined at the SC-β cell stage (Fig. 1I). Collectively, these findings suggest a potential role for GPX2 in early pancreatic development but not in mature β cells.

### GPX2 deletion alters endoderm in vitro differentiation

To directly assess the GPX2 role in β cell differentiation, we generated a homozygous GPX2 loss-of-function model in hPSCs by deleting the entire *GPX2* gene (genomic coordinates, chr14:64,939,301-64,942,926, Hg38) (Fig. 2A) using CRISPR/Cas9 and a set of three single guide RNAs (sgRNAs). Successful *GPX2* knockout (*GPX2* KO) in selected clonal hPSC lines was confirmed by PCR and Sanger sequencing, and the protein level by western blot (WB) (Fig. 2A, Supplementary Fig. 2A). No significant differences were observed between WT and *GPX2* KO hPSCs in terms of colony morphology (Supplementary Fig. 2B), proliferation rates (Supplementary Fig. 2C), and pluripotency marker POU Class 5 Homeobox 1 (OCT3/4) protein expression (Supplementary Fig. 2D).

Compared to WT, *GPX2* KO PEN cells exhibited increased cytoplasmic stress, as measured by $H_2DCFDA$, while no changes in mitochondrial stress levels were detected by MitoSOX staining (Fig. 2B). This suggests that GPX2 deficiency leads to specific cellular stress responses during endoderm differentiation.

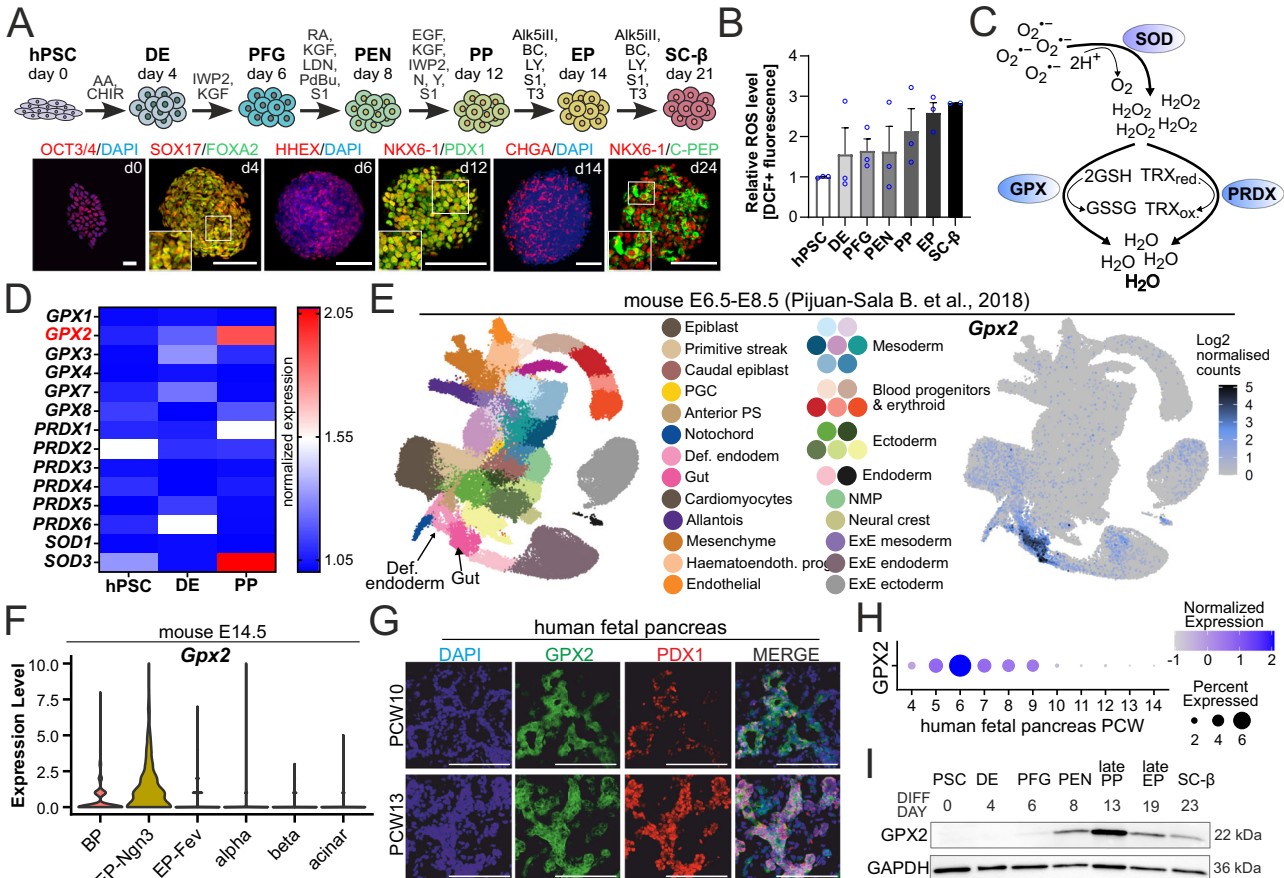

**Fig. 1 | Oxidative stress and GPX2 expression in the developing pancreatic β cells. A** Schematic overview of hPSC differentiation into SC-β cells. Stages and sample collection time points are indicated. Representative immunofluorescence (IF) images show the expression of stage-specific markers. **B** Changes in oxidative stress levels during in vitro SC-β cell differentiation measured by $H_2DCFDA$ intensity relative to hPSCs. Each dot represents one image, with mean and ± SEM indicated. $N = 3$ independent experiments. **C** Diagram illustrates the mechanism of superoxide dismutase (SOD), glutathione peroxidase (GPX), and peroxiredoxin (PRDX) in combating oxidative stress. First, SOD catalyzes the dismutation of superoxide radicals ($O_2^-$) into hydrogen peroxide ($H_2O_2$), which is then converted into water ($H_2O$) and oxygen ($O_2$) by GPX, utilizing reduced glutathione (GSH) as a cofactor, or by PRDX. red., reduced; ox., oxidized. **D** Heatmap illustrates changes in the expression of oxidative stress response genes during in vitro differentiation to the PP stage. Data are row-normalized to highlight differences in gene expression

across stages. Higher gene expression is in red, lower in blue. **E** Molecular single-cell transcriptome atlas of mouse embryos, spanning embryonic days (E) E6.5 to E8.5, encompasses gastrulation and early organ development (https://crukci.shinyapps.io/mousegastrulation2018/, E-MTAB-6967)[19]. On the left, definitive endoderm (DE) and gut are indicated. On the right, the spatial distribution of *Gpx2* mRNA shows elevated mRNA expression in DE and the gut anlage. **F** *Gpx2* expression in E14.5 mouse pancreas, specifically in bipotent progenitors (BP) and Ngn3-expressing early, endocrine progenitors (EP-Ngn3)[20]. **G** Human fetal pancreas at 10 post-conception weeks (PCW) and 13 PCW stained for GPX2 (green) and PDX1 (red) proteins. DAPI marks nuclei (blue). Scale bar = 100 μm. **H** *GPX2* mRNA expression during human pancreas development (4-14 PCW) from scRNA-seq data[36,108]. **I** Western blot depicts GPX2 protein levels during hPSC pancreatic differentiation. Days of cell collection are shown. GAPDH was used as a loading control. $N = 3$ biological repeats. Source data are provided as a Source data file.

Next, we evaluated the differentiation of *GPX2* KO hPSCs towards β cells in a 3D culture format. DE markers, SOX17[21] and FOXA2[22] displayed comparable expression levels in *GPX2* KO compared with WT cells (Fig. 2C, Supplementary Fig. 2F). Differences in gene expression between WT and *GPX2* KO cells became apparent at the PFG stage, coinciding with the onset of GPX2 expression. IF staining and quantification demonstrated increased expression levels of SOX17 by 30%, Caudal Type Homeobox 2 (CDX2) by 77%, and Hepatocyte Nuclear Factor 4 Alpha (HNF4A) by 22%, while HHEX expression decreased by 30% in PFG *GPX2* KO cells (Fig. 2D). At the PFG stage, these marker genes, all key regulators of gut development, play pivotal roles in lineage specification. SOX17 is crucial for gut endoderm morphogenesis[23], CDX2 is essential for initiating intestinal development[24,25], HNF4A controls hepatic and pancreatic progenitor specification[26], and HHEX is a key regulator of liver development and a gatekeeper of pancreatic lineage specification[7,27]. Additionally, WT and *GPX2* KO PFG cells co-expressing SOX17 and HNF4A (Fig. 2F) and SOX17 and HHEX (Supplementary Fig. 2E) were identified.

At the PEN stage, expression levels of PDX1, CDX2, and SOX9 were increased by 66%, 20%, and 61%, respectively, and HHEX levels remained lower by 24% in *GPX2* KO cells compared with WT (Fig. 2E), similar trend was observed in the second clonal GPX2 KO cell line (Supplementary Fig. 2G). PDX1 governs pancreas development[28] and is also expressed in duodenum[29] and SOX9 regulates pancreatic and liver progenitor development[30]. Moreover, at both stages, PFG and PEN, we observed the expression of the liver cell marker, Fibrinogen Beta Chain (FGB), in a fraction of HNF4A+ *GPX2* KO cells (Fig. 2F, G). Interestingly, the PEN HNF4A+/FGB+ cells did not co-express PDX1, which suggests liver-like cell fate specification. Further, co-expression of HNF4A and FGB proteins in WT and KO PFG spheroids was confirmed by FC. Although expressed at low levels overall, FGB expression in KO cells was approximately two-fold higher compared with WT (Fig. 2H, Supplementary Fig. 2H, I). A similar result was obtained at the PEN stage, with 18% of FGB+ *GPX2* KO cells co-expressing HNF4A compared with 8% FGB+/HNF4A+ WT cells (Fig. 2H, Supplementary Fig. 2J).

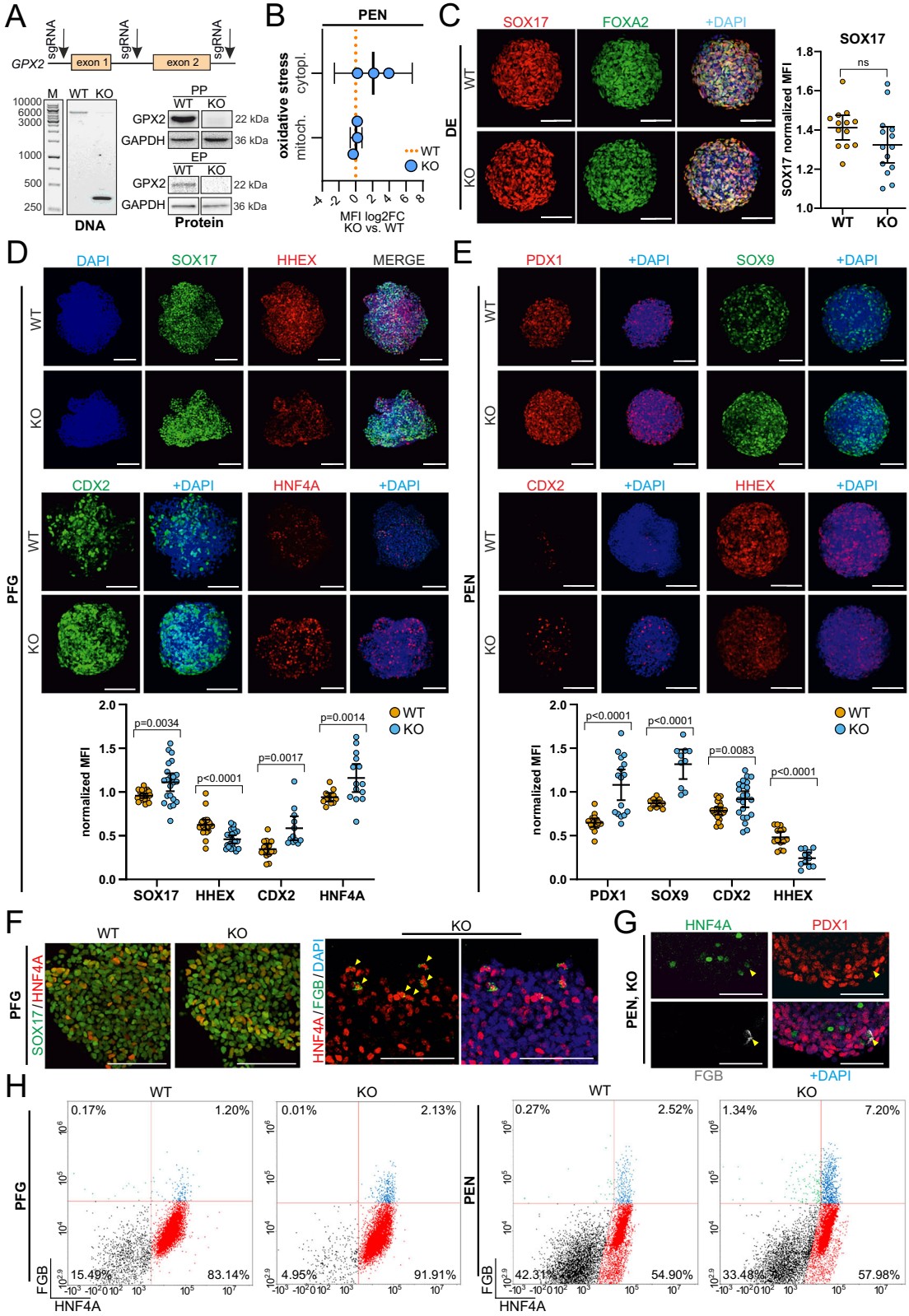

These findings strongly suggest that GPX2 plays a crucial role in directing endoderm differentiation towards a pancreatic lineage, and its absence results in the parallel adoption of alternative liver-like and intestinal-like fates. However, a subset of cells also expressed PDX1, suggesting they adopted a pancreatic progenitor fate in GPX2 absence.

## GPX2 deficiency disrupts posterior foregut patterning

To comprehensively characterize the effects of *GPX2* loss during pancreatic differentiation, we performed a set of RNA sequencing experiments (Fig. 3A). Bulk RNA sequencing of cells at day 6 of differentiation, corresponding to the PFG stage, showed a clear separation between WT and KO cells in the first principal component,

**Fig. 2 | GPX2 deficiency results in altered differentiation of the posterior foregut endoderm. A** Design and validation of *GPX2* knockout (KO) in hPSCs. Three sgRNAs (arrows) were used to delete exons 1 and 2 (orange blocks) of the *GPX2* gene. A representative gel electrophoresis image confirms the deletion at the DNA level. Western blot analysis validates the absence of GPX2 protein in KO (clonal line KO1) pancreatic progenitors (PP) and endocrine progenitors (EP). GAPDH was used as a loading control. **B** Changes in mitochondrial and cytoplasmic oxidative stress levels in KO (clonal line KO1) pancreatic endoderm (PEN) compared to WT PEN, assessed by flow cytometry. Mean and 95% confidence intervals (CIs) are indicated. *N* = 3 independent experiments. An unpaired, two-sided t-test was used to determine statistical significance. **C** Representative IF staining for SOX17 (red) and FOXA2 (green) in WT and KO definitive endoderm (DE). DAPI marks nuclei (blue). Scale bar = 100 μm. Quantification of the mean fluorescence intensity (MFI) of SOX17 relative to DAPI for WT (orange) and KO (clonal line KO1, blue) spheroids is shown as a dot plot. Each dot represents one image, with mean and 95% confidence intervals (CIs) indicated. *N* = 3 independent experiments. An unpaired, two-sided t-test was used to determine statistical significance. **D** Representative IF staining for SOX17 (green) and HHEX (red), or CDX2 (green) and HNF4A (red) proteins in WT and KO (clonal line KO1) posterior foregut (PFG). DAPI marks nuclei (blue). Scale bar = 100 μm. Quantification of marker mean fluorescence intensity (MFI) relative to DAPI for WT (orange) and KO (blue) spheroids is shown as a dot plot. Each dot represents one image, with mean and 95% confidence intervals (CIs) indicated. *N* = 3 independent experiments. An unpaired, two-sided t-test was used to determine statistical significance. *p*-value for each WT vs. KO comparison were: SOX17: *p* = 0.0034, HHEX: *p* < 0.0001, CDX2: *p* = 0.0017, HNF4A *p* = 0.0014.

**E** Representative IF staining for PDX1 (red), SOX9 (green), or CDX2 (red) and HHEX (red) proteins in WT and KO (clonal line KO1) pancreatic endoderm (PEN). DAPI marks nuclei (blue). Scale bar = 100 μm. Quantification of the marker mean fluorescence intensity (MFI) relative to DAPI for WT (orange) and KO (blue) spheroids is shown as a dot plot. Each dot represents one image, with mean and 95% confidence intervals (CIs) indicated. *N* = 3 independent experiments. An unpaired, two-sided t-test was used to determine statistical significance. *p*-value for each WT vs. KO comparison were: PDX1: *p* < 0.0001, SOX9: *p* < 0.0001, CDX2: *p* = 0.0083, HHEX: *p* < 0.0001. **F** Zoom inserts show SOX17 (green) and HNF4A (red) co-expression in WT and KO posterior foregut (PFG), and HNF4A (red) and FGB (green) in KO (clonal line KO1) PFG. Cells co-expressing HNF4A and FGB are indicated by yellow arrows. DAPI staining marks nuclei (blue). Scale bar = 100 μm. Due to low FGB presence in WT PFG cells, as shown by FC analysis in (**H**), co-staining for HNF4A and FGB is shown only in KO cells. *N* = 3 biological repeats. **G** Zoom inserts show HNF4A (green), PDX1 (red), and FGB (gray) expression in KO (clonal line KO1) pancreatic endoderm (PEN). The merged image includes DAPI-stained nuclei (blue). The yellow arrow marks a cell co-expressing HNF4A and FGB but not PDX1. Scale bar = 50 μm. Due to low FGB presence in WT PFG cells, as shown by FC analysis in (**H**), co-staining for HNF4A, PDX1, and FGB is shown only in KO cells. *N* = 3 biological repeats. **H** Representative flow cytometry analysis shows the expression levels of HNF4A (APC-H) and FGB (FITC-H) proteins in WT and KO (clonal line KO2) at PFG and PEN stages. In KO, a higher percentage of cells co-express HNF4A and FGB at both stages compared to the WT counterpart. Source data are provided as a Source data file.

capturing 57.2% of the variation (Fig. 3B, Supplementary Fig. 3A). Among the top differentially expressed genes (DEGs), we observed the upregulation of *CDX2* and *HNF4A* transcripts and downregulation of *HHEX* (Fig. 3C, Supplementary Data 1), corroborating the IF results. Additionally, we identified the upregulation of anterior DE and midgut markers, including Insulin-Like Growth Factor Binding Protein 5 (*IGFBP5*) and S100 Calcium Binding Protein A13 (*S100A13*)[31]. We also noted downregulation of the DE patterning gene Orthodenticle Homeobox 2 (*OTX2*) and increased expression of Transcription Factor AP-2 Gamma (*TFAP2C*), which is upregulated in *OTX2*-deficient human *DE*[32]. Concomitantly with the downregulation of the anterior marker *OTX2*, we found the upregulation of the posterior marker *CDX2*[33], suggesting altered temporal patterning in *GPX2* KO PFG. We also observed increased Bone Morphogenetic Protein 6 (*BMP6*) expression in KO PFG[34,35], suggesting a pro-liver-like bias in *GPX2* foregut endoderm. Together, these results demonstrate the dysregulation of multiple genes critical for foregut development and patterning.

We next investigated how the putative PFG patterning alterations, induced by the identified transcriptomic changes in *GPX2* KO, influence cell fate acquisition following pancreatic progenitor induction. To this end, we analyzed WT and KO cells at day 8 of differentiation, 24 h after directed pancreatic fate induction by a combination of Retinoic Acid (RA) and Protein Kinase C (PKC) pathways activation along with BMP and Sonic hedgehog (SHH) pathway inhibition, using single-cell combinatorial indexing RNA sequencing (scRNA-seq) (Fig. 3D, Supplementary Fig. 3B, C). The top variable genes within the dataset were also differentially expressed between *GPX2* KO and WT cells (Supplementary Fig. 4A, B). Using Uniform Manifold Approximation and Projection (UMAP) dimensionality reduction, we observed separation between WT and *GPX2* KO cells (Fig. 3D). Due to the low detection of peri-pancreatic endoderm markers *PDX1* and *SOX9* (Supplementary Fig. 4C) in early PEN, likely due to scRNA-seq sensitivity, we identified four PEN clusters based on other known markers, such as multipotent pancreatic progenitor markers One Cut Homeobox 2 (*ONECUT2*), SRY-Box Transcription Factor 11 (*SOX11*), ZFP36 Zinc Finger Protein Like 1 (*ZFP36L1*), and ZFP36 Zinc Finger Protein Like 2 (*ZFP36L2*)[36], as well as CD24 Molecule (*CD24*)[37] (Fig. 3E, Supplementary Fig. 4D, F). Among these populations, the PEN_3 and PEN_4 clusters had the highest expression of these early pancreatic markers, suggesting they were the most mature pancreatic progenitors at this stage. We

also identified two posterior foregut (PFG_1 and PFG_2) clusters marked by *GATA4* and PR/SET Domain 1 (*PRDM1*), and three smaller clusters referred to as non-pancreatic progenitors (NPP), marked by Thymocyte Selection Associated High Mobility Group Box (*TOX*), Sodium Voltage-Gated Channel Alpha Subunit 7 (*SCN7A*), and Phosphodiesterase 3A (*PDE3A*), respectively (Fig. 3E). Interestingly, a distinct cluster was marked by the *IGFBP5* gene (Fig. 3E), previously identified as upregulated in *GPX2* KO cells at day 6 (Fig. 3C), which might indicate intestinal cell commitment[38–40].

For each of the clusters, we identified highly enriched genes as putative novel markers of these PFG subpopulations (Supplementary Fig. 4E, Supplementary Data 2), together with genes that were deregulated between WT and KO genotypes, irrespective of cell type (Supplementary Fig. 4G, Supplementary Data 2). Interestingly, we found that the *HHEX* transcript, which was downregulated in *GPX2* KO cells at the protein level (Fig. 2D, E), was more abundant in PEN clusters enriched in WT cells (Supplementary Fig. 4F). Similarly, transcripts of *HNF4A*, which were enriched in KO cells (Fig. 2D), were less abundant in WT-enriched PEN clusters (Supplementary Fig. 4F). Therefore, scRNA-seq-based results corroborated the differential expression of proteins at the PEN stage.

Importantly, 65% of KO cells belonged to PFG clusters, compared to less than 30% of WT cells. Conversely, 41% of WT cells versus only 5% of *GPX2* KO cells belonged to the PEN_3 and PEN_4 clusters (Fig. 3F, Supplementary Fig. 4H), which represent the most progressed pancreatic progenitors. Based on these results, we concluded that *GPX2* KO cells exhibited a reduced response to forced pancreatic fate induction compared to WT cells.

To investigate the mechanisms underlying the observed cell fate bias, we analyzed pathway deregulation. For that, we used the pathway activity analysis tool PROGENy[41], which calculates pathway activity scores based on pathway regulons: a set of genes whose expression is influenced by pathway activation or inhibition. Greater changes in the expression of pathway-responsive genes result in a higher pathway activity score. This analysis revealed a downregulation of the hypoxia regulon and upregulation of the TGF-β regulon in *GPX2* KO cells as the most significantly deregulated pathways (Fig. 3G, Supplementary Data 2). Importantly, the relative activities of these regulons were consistently deregulated across most PFG and PEN clusters (Fig. 3H), suggesting a genotype-specific

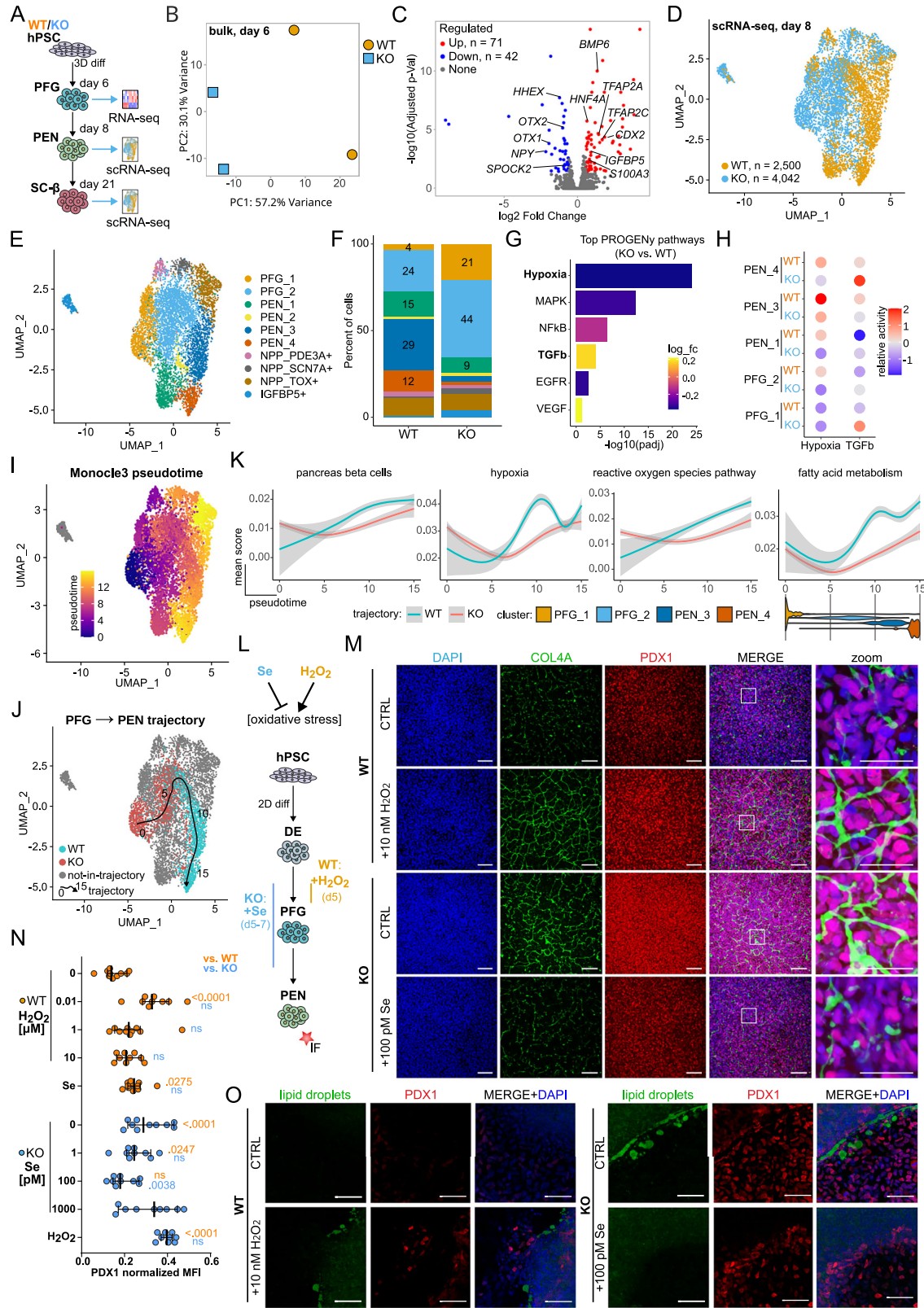

effect rather than cluster frequency bias. Trajectory analysis of WT and KO cells transitioning from PFG to PEN stages along pseudotime (Fig. 3I, J, Supplementary Fig. 4I) revealed that Molecular Signal Database (MSigDB) terms such as "pancreas beta cells", "hypoxia", "reactive oxygen species" and "fatty acid metabolism" were enriched in WT cells during transcriptomic progression to PEN stages (Fig. 3K). This led us to investigate whether *GPX2* loss-induced deregulation of

oxidative stress response pathways contribute to the delayed pancreatic fate acquisition in *GPX2* KO cells.

To further explore the role of oxidative stress, we treated WT and KO cells with varying concentrations of hydrogen peroxide ($H_2O_2$; 0.01–10 μM) to induce oxidative stress, and selenium (Se; 1–1000 pM) to reduce it. As controls, we treated WT cells with 100 pM Se and KO cells with 10 nM $H_2O_2$. $H_2O_2$ at concentrations exceeding 1 mM was

**Fig. 3 | Oxidative stress manipulation mimics GPX2 deficiency at the pancreatic endoderm stage. A** Scheme illustrates the sequencing approaches of WT and KO differentiating cells: bulk RNA-seq was performed at posterior foregut (PFG) stage, scRNA-seq was conducted at pancreatic endoderm (PEN) and stem cell-derived β (SC-β) cell stages. **B** Principal Component Analysis (PCA) analysis shows transcriptional profiles of WT (orange) and KO (clonal line KO1, blue) PFG. **C** Volcano plot illustrates downregulated (blue) and upregulated (red) DEGs in KO (clonal line KO1) PFG compared to WT PFG. Selected genes are labeled on the plot. Differential expression was assessed using the DESeq2 Wald test (two-sided), and $p$-values were adjusted for multiple comparisons using the Benjamini–Hochberg false discovery rate (FDR) method. **D** Uniform manifold approximation and projection (UMAP) plot shows differences between WT (orange) and KO (clonal line KO1, blue) PEN. Each dot represents a single cell. **E** UMAP plot demonstrates the main cell types in PEN, distributed into 10 different clusters. Each dot represents a single cell, and each cell type is marked with a different color. **F** Percentage of cells assigned to each cell type for WT and KO (clonal line KO1), colored according to (**E**). **G** Bar plot shows top significantly dysregulated signaling pathways in KO (clonal line KO1) vs. WT PEN, as identified by PROGENy. Upregulated and downregulated pathways are colored in yellow and blue, respectively. **H** Dot plot illustrates changes in hypoxia and TGF-β pathway activities between KO (clonal line KO1) and WT PEN across selected clusters. Increased pathway activity is shown in red, and decreased activity is shown in blue. **I** UMAP feature plot shows pseudotime values calculated using the Moncole3 package. Cells are ordered in pseudotime, starting from earlier cells marked in blue to later cells marked in yellow. Each dot represents a single cell. **J** UMAP plot visualizes the differentiation trajectory from foregut to pancreatic endoderm. Cells are color-coded by genotypes. The black arrow indicates the

trajectory. Gray cells are not assigned to the trajectory. Each dot represents a single cell. **K** Plots show the mean score of selected hallmark gene sets from the Molecular Signature Database along pseudotime for WT (blue) and KO (red) trajectories, as depicted in (**J**). Gray areas depict 95% confidence intervals. The violin plots at the bottom represent the pseudotime distribution of analyzed clusters for cells within the trajectories. **L** Scheme of the experimental design to investigate oxidative stress influence on cell differentiation. Differentiation stages, the timing of $H_2O_2$ (stress inducer) or selenium (stress reducer) treatments are shown. IF staining was performed at the PEN stage (a red star). **M** Representative images of WT and KO (clonal line KO1) PENs differentiated with 10 nM $H_2O_2$ or 100 pM Se. Cells were stained for PP marker PDX1 (red) or ECM component, COL4A1 (green). DAPI stains nuclei (blue). Scale bar = 100 μm. $N = 3$ independent experiments. Zoomed-in inserts are also shown. **N** Quantification of IF staining for WT and KO (clonal line KO1) PENs differentiated with or without oxidative stress modulation. WT (orange) and KO (blue) cells were treated with varying concentrations of $H_2O_2$ and selenium, respectively. As controls, WT cells were treated with 100 pM Se and KO cells with 10 nM $H_2O_2$. WT is marked in orange, and KO is shown in blue. Each dot represents one image, with mean and 95% confidence intervals (CIs) indicated. $N = 3$ independent experiments. One-way ANOVA with Sidak's multiple comparisons test was used to determine statistical significance. Exact $p$-values are shown and colored according to the reference (control) condition for each comparison, i.e., orange means comparison to WT without added $H_2O_2$ and blue to KO without selenium. **O** Representative images of WT and KO (clonal line KO1) PENs differentiated with or without regulation of oxidative stress and stained with Nile Red to visualize lipid droplets (green) and for PDX1 (red). DAPI stains nuclei (blue). Scale bar = 100 μm. $N = 3$ independent experiments. Source data are provided as a Source data file.

---

lethal to the cells. Following treatment at the DE and PFG stages, PDX1 protein expression was analyzed 3 days later (Fig. 3L). Treatment with 0.01 μM $H_2O_2$ significantly elevated PDX1 expression in WT cells, mimicking the levels observed in *GPX2* KO cells. Conversely, 100 pM Se treatment decreased PDX1 expression in KO cells (Fig. 3M, N) to levels comparable to untreated WT cells. PDX1 expression was not changed in the applied control conditions. We observed similar effects of $H_2O_2$ and Se on HNF4A expression (Supplementary Fig. 4J).

Next, we assessed collagen fiber integrity. WT and Se-treated KO cells exhibited shortened and disorganized collagen fibers, while KO and $H_2O_2$-treated WT cells presented more abundant and better-organized fiber structures (Fig. 3M). Furthermore, we observed the accumulation of lipid droplets in *GPX2* KO and $H_2O_2$-treated WT cells (Fig. 3O). This lipid accumulation is in line with the presence of oxidative stress in these cells[42,43]. Therefore, $H_2O_2$-induced oxidative stress recapitulates the developmental bias of *GPX2* KO, and Se treatment rescues this phenotype by restoring the expression of pancreatic and liver markers in KO cells. Together, our findings imply a critical role for oxidative stress in modulating developmental trajectories.

## GPX2-deficient cells show propensity for hepatic-like lineage

We next examined the potential of GPX2-deficient cells to differentiate towards β cells. Surprisingly, at the SC-β cell stage (day 21), we did not observe significant changes in the protein levels of key β cell markers between WT and *GPX2* KO cells, including PDX1, CHGA[44], NKX6-1[45], and C-PEP[46] (Fig. 4A–C, Supplementary Fig. 5A). WT and KO cells co-expressed NKX6-1 and C-PEP (Fig. 4B), as expected in β cells, with similar frequency. Thus, KO cells differentiated into β cells as efficiently as WT cells, despite the alterations observed during PFG development. However, we observed altered frequencies of lineage markers for other populations than β cells, e.g., an increased number of GCG+ α cells (Supplementary Fig. 5B, C). Moreover, we detected elevated expression of liver markers, i.e., HNF4A and α-fetoprotein (AFP)[26,47,48] (Fig. 4A, C), in day 21 KO cells, confirming a higher propensity of KO cells to adopt a liver-like progenitor fate. In *GPX2* KO cells, co-staining for C-PEP and FGB, as well as PDX1 and AFP (Fig. 4B), revealed that liver markers were expressed in cells lacking endocrine markers, suggesting the endocrine and hepatic-like specification

occurring in parallel. Together, our data suggest that under strong pro-pancreatic conditions during in vitro differentiation, *GPX2* KO cells differentiate into two distinct lineages: endocrine-like and multilineage progenitors, including liver-like progenitors. Consistently, quantification at the SC-β stage showed that FGB levels reached nearly 14% in *GPX2* KO cells compared to only ~1% in WT, and AFP protein levels were also significantly increased in KO cells relative to WT counterparts (Supplementary Fig. 5D, E). Consistent findings were also observed in the second clonal GPX2 knockout cell line (Supplementary Fig. 5F, G).

To uncover and analyze the cellular landscape of WT and *GPX2* KO cells at the SC-β cell stage, we performed scRNA-seq on differentiation day 21. A total of 6128 WT and 5610 KO cells at the SC-β cell stage were sequenced using 10x Chromium platform (Supplementary Fig. 6A–C). Following cell filtering and quality control, the unsupervised analysis identified 11 distinct clusters, visualized using UMAP (Supplementary Fig. 6D, E). Cell identities within each cluster were assigned based on the expression of lineage marker genes specific to each cluster (Supplementary Fig. 6D, F). Next, we excluded mesenchymal and proliferating cells from further analysis to focus on pancreatic and hepatic-like cell differentiation. After re-clustering, we retained 5515 cells for WT and 4648 cells for *GPX2* KO, grouped into 10 clusters (Fig. 4D–F). Endocrine cells from both genotypes grouped, forming SC-β, δ, and multihormonal (MH) cell clusters, with similar proportions of cells in each cluster (Fig. 4F). However, in KO samples, the endocrine cell cluster (EC) contained 10% more cells compared to WT. In contrast, the EP cluster constituted 22% of WT but only 4% of KO cells (Fig. 4F). Analysis of canonical endocrine markers in the SC-β, MH, EP, EC, and δ cell clusters confirmed the identity of each cluster while showing no significant differences in these marker expression levels between genotypes (Fig. 4G). At the same time, however, we identified upregulated and downregulated genes in *GPX2* KO cells compared to WT within each cluster (Fig. 4H, Supplementary Data 3), suggesting that while GPX2-deficient endocrine cells largely resemble WT, some transcriptional differences exist.

At day 21 of differentiation, we still observed a significant number of progenitor cells, a common occurrence in β-cell differentiation protocols[49]. Pancreatic progenitor cells (PP1 and PP2 clusters) constituted nearly 60% of WT populations but less than 2% in KO cells

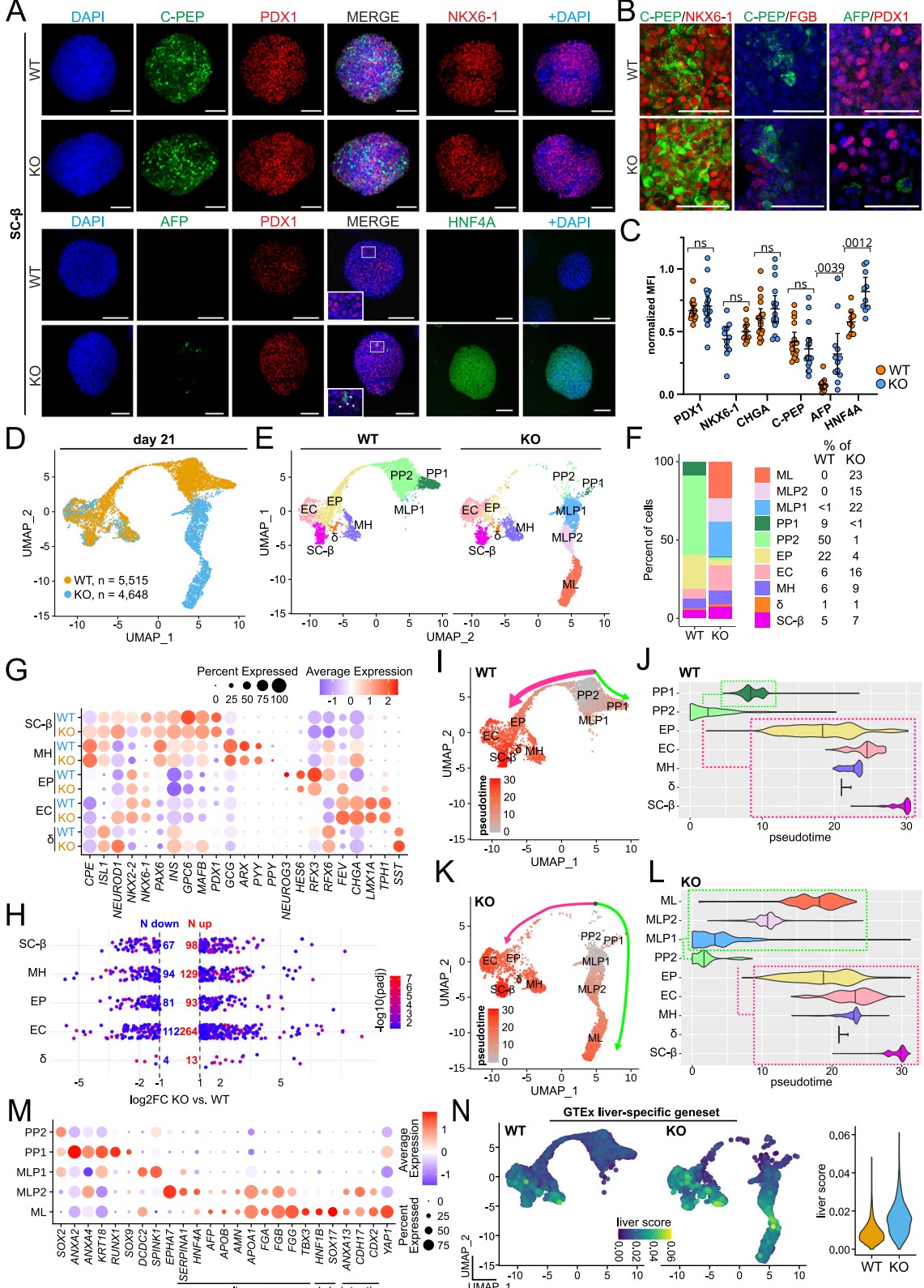

(Fig. 4F). Interestingly, in *GPX2* KO samples, three distinct clusters emerged: multilineage progenitors 1 (MLP1, 22%), multilineage progenitors 2 (MLP2, 15%), and multilineage (ML, 23%), which collectively comprised less than 1% in WT cells. Given these pronounced differences in cluster composition between WT and KO (Supplementary Fig. 6G), we performed pseudotime trajectory analysis using Monocle3[50]. In WT cells, differentiation trajectories progressed

towards endocrine cell fates, though a substantial proportion of cells remained as various pancreatic progenitors (Fig. 4I, J). In KO cells, however, we observed bifurcation into two distinct differentiation trajectories: endocrine and multilineage cell fates (Fig. 4K, L). Liver-like and intestine-like progenitor markers were highly expressed in the MLP2 and ML clusters (Fig. 4M). We noted the increased expression of Serine Peptidase Inhibitor Kazal Type 1 (*SPINK1*) in MLP1, Serpin Family

**Fig. 4 | GPX2-deficient cells show hepatic bias under pro-pancreatic conditions.**
**A** Representative IF staining of WT and KO (clonal line KO1) SC-β cells for C-PEP (green), PDX1 (red), NKX6-1 (red), AFP (green), and HNF4A (green). DAPI marks nuclei (blue). Scale bar = 100 μm. $N$ = 3 independent experiments. **B** Zoomed-in inserts show IF co-staining of C-PEP (green) with NKX6-1 (red), C-PEP (green) with FGB (red), and AFP (green) with PDX1 (red) in WT and KO (clonal line KO1) cells at the SC-β cell stage. FGB is not co-expressed with C-PEP, nor are AFP and PDX1 co-expressed. DAPI (blue) marks nuclei. Scale bar = 100 μm. **C** Quantification of mean fluorescence intensity (MFI) of PDX1, NKX6-1, CHGA, C-PEP, AFP, and HNF4A expression in WT (orange) and KO (clonal line KO1, blue) spheroids shown as a dot plot. Each dot represents one image. The median with 95% CI is shown on the graph. The $p$-values were calculated using an unpaired, two-sided t-test. $N$ = 3 independent experiments. **D** UMAP embedding projection of an integrated dataset of 10,163 WT and KO (clonal line KO1) cells at the SC-β cell stage (day 21). Proliferating cells in the S- or M-phase of the cell cycle were excluded from the dataset. Cells are labeled by sample origin (WT - orange, KO - blue). Each dot represents a single cell. **E** Clustering of datasets shown in (**D**) into various cell types present at the SC-β cell stage, organized into 10 different clusters. UMAP projections are shown separately for WT and KO (clonal line KO1). Each dot represents a single cell, and each cell type is color-coded. **F** Bar plot shows the percentage of cells assigned to each cluster for WT and KO (clonal line KO2), with colors corresponding to (**E**). **G** Dot plot represents the expression of marker genes across clusters characteristic for endocrine cells in KO and WT spheroids. SC-β - stem cells-derived β cells, MH - multihormonal cells, EP - endocrine progenitors, EC - endocrine cells, δ - delta cells. Genes with higher average expression are shown in red, and with lower in blue. Dot size represents the percentage of cells expressing selected genes. **H** Dot plot represents upregulated and downregulated genes in KO (clonal line KO1) compared to WT across endocrine cell clusters. SC-β - stem cells-derived β cells, MH - multihormonal, EP - endocrine progenitors, EC - endocrine cells, δ - delta cells. For each cluster, the number of differentially expressed genes (DEGs) is indicated, with

upregulated genes shown in red and downregulated genes in blue. Dot color intensity corresponds to the $p$-value for each gene. Differential expression was assessed using the DESeq2 Wald test (two-sided), and $p$-values were adjusted for multiple comparisons using the Benjamini–Hochberg false discovery rate (FDR) method. **I** UMAP projection of pseudotime analysis for in vitro differentiated WT cells. Cells predicted to appear later in time are highlighted in red. The main differentiation direction is marked by a wide, pink arrow. The green arrow indicates cells preferentially differentiating in directions other than endocrine cells. **J** Violin plot represents the pseudotime distribution of each cluster in WT samples. Differentiation directions indicated in (**I**) are highlighted with boxes outlined by dashed lines. The main differentiation direction is marked by a pink and alternative path by a green box. **K** UMAP projection of pseudotime analysis for in vitro differentiated KO cells. Cells predicted to appear later in time are highlighted in red. Cells differentiating into endocrine cells are indicated by a pink arrow, while those acquiring a multilineage progenitor fate are marked with a green arrow. **L** Violin plot represents the pseudotime distribution of each cluster. Differentiation directions indicated in (**K**) are highlighted with boxes outlined by dashed lines, pink for endocrine differentiation direction, and green for multilineage direction. **M** Dot plot represents the expression of marker genes across pancreatic progenitors and multilineage progenitor clusters in KO and WT cells. PP1 - pancreatic progenitors 1, PP2 - pancreatic progenitors 2, ML1 - multilineage progenitors 1, ML2 - multilineage progenitors 2, ML - multilineage. Marker genes for liver, intestine, and bile duct (bd) genes are indicated. Genes with higher average expression are shown in red, while those with lower are shown in blue. Dot size represents the percentage of cells expressing selected genes. **N** Feature and violin plots show individual cell scores for expression of adult human liver-specific genes, based on the GTEx database[6]. Cells exhibiting high pathway scores are marked in yellow, and those with low scores are shown in blue. A violin plot illustrates the distribution of pathway scores, with WT marked in orange and KO in blue. Source data are provided as a Source data file.

A Member 1 (*SERPINA1*), *HNF4A*, *AFP*, Apolipoprotein A1 (*APOA1*), and Apolipoprotein B (*APOB*) in MLP2; *HNF4A*, *APOA1*, *AFP* and HNF1 Homeobox B (*HNF1B*) were also highly expressed in ML (Fig. 4M). The mRNAs of hepatic endoderm markers, *SOX17* and T-Box Transcription Factor 3 (*TBX3*)[51,52], were also enriched in the ML cluster (Fig. 4M). We did not detect any albumin mRNA in our samples, which might reflect the immature, progenitor state of the cells or the insufficient sequencing depth. Furthermore, we scored each cell for expression of human liver-enriched genes[6] and found that KO cells were enriched for these genes compared to WT cells, with the emphasis on MLP2, ML, and surprisingly, endocrine cell clusters (Fig. 4N). Interestingly, analysis of a scRNA-seq dataset from developing human liver (PCW5-6) reveals that among the *PRDX*, *GPX*, and *SOD* gene families, *GPX2* is the only transcript expressed at a substantially reduced level in liver cells (Supplementary Fig. 6H)[9]. These results further supported the emergence of alternative differentiation fates upon *GPX2* deletion, even under pro-pancreatic signals.

**Oxidative stress in liver vs. pancreas cell fate decisions**
To isolate the intrinsic differentiation bias of *GPX2* KO cells, we differentiated hPSCs to DE and then cultured the cells as spheroids in basal medium, lacking endocrine lineage-specific signals (Fig. 5A). By day 8, KO cells exhibited significantly elevated protein levels of liver markers: FGB (3-fold), AFP (1.5-fold), and HNF4A (2.5-fold), as well as the intestinal marker CDX2 (3.5-fold) (Fig. 5B–D). Co-staining for HNF4A, a marker of early liver development, with AFP and FGB, associated with liver maturation[9,26] confirmed liver-like progenitor formation in KO, with all AFP⁺ or FGB⁺ cells also co-expressing HNF4A (Fig. 5C).

To investigate GPX2 suppressive role in liver-like fate acquisition, we generated doxycycline-inducible GPX2 overexpression (OE) hPSC line on HUES8 and H1 background (Supplementary Fig. 7A–D). 24-h treatment of GPX2 OE hPSCs with 1 μg/mL doxycycline resulted in 3-fold increase in GPX2 expression (Supplementary Fig. 7B). Inducing GPX2 OE at DE stage, followed by spontaneous differentiation, led to

an 80% decrease in AFP and FGB protein expression compared to doxycycline-treated WT cells at day 8 (Fig. 5E, F, Supplementary Fig. 7D). Consequently, overexpression of GPX2 blocks liver-like fate, an effect opposite to GPX2 loss.

We then explored the correlation between differentiation bias and oxidative stress levels. We determined that GPX2 protein expression emerges at day 6 of pancreatic differentiation (Fig. 1I). As GPX2 acts as an antioxidant, we assumed that its expression may be triggered by increased oxidative stress, which led us to induce oxidative stress shortly before GPX2 expression starts. Treatment of WT cells with 10 nM $H_2O_2$ increased (by >60%) HNF4A expression, and KO cells with 100 pM Se decreased (by 30%) HNF4A levels in KO cells (Fig. 5G, H). Therefore, $H_2O_2$ treatment of WT cells during spontaneous DE differentiation recapitulates the cell fate bias observed in GPX2-deficient PEN, suggesting that $H_2O_2$ induced oxidative stress impacts the cell fate decisions regardless of the in vitro differentiation approach.

Bulk RNA-seq on WT, WT treated with $H_2O_2$ (WT + $H_2O_2$), and *GPX2* KO spontaneously differentiated DE cells revealed distinct transcriptional profiles among all samples with KO and WT + $H_2O_2$ clustering more closely to each other than to WT (Supplementary Fig. 7E). KO cells, exhibited significant upregulation of liver-associated genes, including Cytochrome P450 Family 2 Subfamily E Member 1 (*CYP2E1*), Hepatocyte Growth Factor (*HGF*), extracellular matrix (ECM) components, including Collagen Type VII Alpha 1 Chain (*COL7A1*), Laminin Subunit Alpha 4 (*LAMA4*), and TGF-β pathway, including Nodal Growth Differentiation Factor (*NODAL*) and Bone Morphogenetic Protein 8a (*BMP8A*). Interestingly, WT + $H_2O_2$ cells exhibited similar transcriptional changes (Supplementary Fig. 7F, G). Focusing on liver-associated genes, we identified overlapping upregulated genes in *GPX2* KO cells and WT + $H_2O_2$ cells compared to WT (Fig. 5I, Supplementary Data 4). Gene Set Enrichment Analysis (GSEA) corroborated these findings, revealing enrichment of liver- or oxidative stress-associated terms, particularly cholesterol homeostasis, in both KO and WT + $H_2O_2$ cells (Fig. 5J) and enrichment of the ROS pathway in KO cells (Supplementary Fig. 7H).

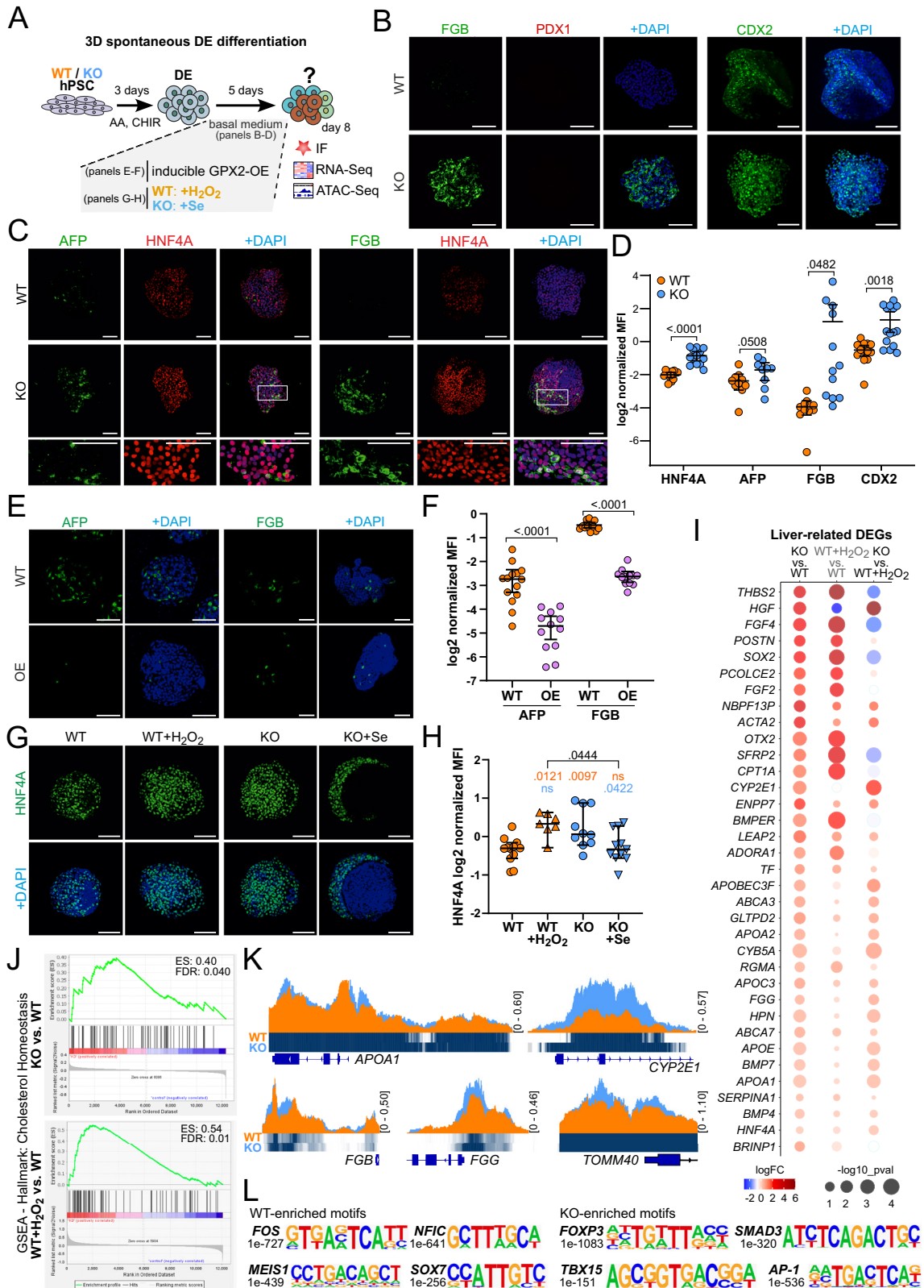

Next, we performed an Assay for Transposase-Accessible Chromatin using sequencing (ATAC-seq) of WT and *GPX2* KO cells differentiated spontaneously until day 8 (Supplementary Fig. 7J–N). This revealed significant changes in chromatin accessibility at liver genes or regulatory sequences, including *APOA1*, *CYP2E1*, *FGB*, Fibrinogen Gamma Chain (*FGG*), and Translocase of Outer Mitochondrial Membrane 40 (*TOMM40*) (Fig. 5K), consistent with the RNA-seq data. Motif

enrichment analysis in KO cells showed enrichment of motifs for transcription factors involved in TGF-β signaling activation, such as *JUN* (*AP-1*) and SMAD Family Member 3 (*SMAD3*)[53], as well as Forkhead Box P3 (*FOXP3*), which is crucial for TGF-β-mediated regulatory T cell differentiation and immune tolerance[54]. Additionally, a motif for T-Box Transcription Factor 15 (*TBX15*) was identified in KO cells (Fig. 5L, Supplementary Data 5). In mice, *Tbx3*, a related T-Box binding

**Fig. 5 | Spontaneous differentiation shows a higher propensity of *GPX2* KO DE cells to differentiate towards liver-like progenitors. A** Schematic representation of the experimental setup used to analyze spontaneous differentiation of definitive endoderm (DE) under oxidative stress. WT and KO hPSCs were differentiated into DE with activin A (AA) and CHIR99021 (CHIR) treatment. On day 5, WT cells were treated with hydrogen peroxide ($H_2O_2$), KO cells were treated with selenium (Se), or *GPX2* overexpression (OE) was induced with doxycycline. On day 8, cells were analyzed using immunofluorescence (IF), RNA sequencing (RNA-seq), and assay for transposase-accessible chromatin using sequencing (ATAC-seq). **B** Representative IF images of WT and KO (clonal line KO1) cells stained for liver marker FGB (green), pancreatic marker PDX1 (red), and intestinal marker CDX2 (green), with DAPI (blue) as a nuclear counterstain. Scale bar = 100 µm. $N = 3$ biological repeats. **C** Representative IF images of WT and KO (clonal line KO2) cells stained for liver markers: AFP (green) and FGB (green), both co-stained with liver marker HNF4A (red). DAPI (blue) was used as a nuclear counterstain. Scale bar = 100 µm. **D.** Dot plot shows quantification of mean fluorescence intensity (MFI) of HNF4A, AFP, FGB, and CDX2 relative to DAPI for WT (orange) and KO (clonal line KO1 and 2, blue) spheroids. Each dot represents one image. $N = 3$ independent experiments. Data are presented on the plot as means; error bars are 95% CIs. The *p*-values were calculated using an unpaired, two-sided t-test. *p*-value for each WT vs. KO comparison were: HNF4A: $p < 0.0001$, AFP: $p = 0.0508$, FGB: $p = 0.0482$, CDX2: $p = 0.0018$. **E** WT and GPX2 OE (in HUES8) cells at day 8 of spontaneous differentiation, stained for liver markers AFP (green) and FGB (green). DAPI (blue) marks nuclei. Scale bar = 100 µm. WT and OE cells were treated with 1 µg/mL doxycycline for 24 h at day 5 of differentiation. **F** Dot plot represents the quantification of mean fluorescence intensity (MFI) of AFP and FGB relative to DAPI for WT (orange) and OE (pink) cells at day 8 of spontaneous differentiation. Each dot represents one image. $N = 3$ independent experiments. Data are presented as mean on the plot; error bars are 95% CIs. The *p*-values were calculated using an unpaired, two-sided t-test. *p*-value for each WT vs.

OE comparison: AFP: $p < 0.0001$, FGB: $p < 0.0001$. **G** Representative IF images of WT and KO (clonal line KO2) cells at day 8 of spontaneous differentiation, stained for HNF4A (green). On day 5 of differentiation WT cells were treated with 10 nM $H_2O_2$ and KO cells were exposed to 100 pM Se for 72 h. DAPI (blue) marks nuclei. Scale bar = 100 µm. **H** Dot plot represents the quantification of mean fluorescence intensity (MFI) of HNF4A relative to DAPI for WT (orange), WT + $H_2O_2$ (orange), KO (clonal line KO2, blue), and KO + Se (blue) cells at day 8 of spontaneous differentiation. Each dot represents one image. $N = 3$ independent experiments. Data are presented as mean with 95% CIs. *p*-values were calculated using an ANOVA test. **I** Dot plot represents the differential expression of liver-related genes across the following comparisons: KO (clonal line KO1) vs. WT, WT + $H_2O_2$ vs. WT, and KO vs. WT + $H_2O_2$. WT, WT + $H_2O_2$ and KO cells were differentiated spontaneously until day 8. Genes with higher average expression are shown in red, and with lower are shown in blue. Dot size is proportional to the $-\log_{10}$(*p*-value). Differences in gene expression between samples were calculated using a two-sided empirical Bayes moderated t-test (limma-voom), with *p*-values adjusted for multiple testing using the Benjamini–Hochberg false discovery rate (FDR) method. **J** Enrichment plot from the gene set enrichment analysis (GSEA) of genes differentially expressed between KO (clonal line KO1) vs. WT and WT + $H_2O_2$ vs. WT cells in the RNA-seq experiment, compared to the cholesterol homeostasis signaling gene set from hallmark gene sets. We performed 1000 permutations in the GSEA analysis. Gene sets with a false discovery rate (FDR) ≤ 0.04 and enrichment score (ES) ≤ 0.54 are shown. **K** ATAC-seq tracks highlight the *loci* of *APOA1*, *CYP2E1*, *FGB*, *FGG*, and *TOMM40* in WT (orange) and KO (clonal line KO2, blue) cells differentiated spontaneously until day 8. The peaks represent normalized and combined biological replicates ($N = 2$). **L** Binding motifs of transcription factors identified by motif enrichment analysis in WT and KO (clonal line KO2) cells differentiated spontaneously until day 8. Source data are provided as a Source data file.

transcription factor, plays a crucial role in cell-fate determination and liver organogenesis[51] and *TBX3*-deficient hPSCs demonstrate reduced hepatocyte differentiation and an increased pancreatic progenitor differentiation[55].

We also investigated whether KO and WT + $H_2O_2$ cells exhibit an increased propensity for spontaneous intestinal differentiation. Analysis of intestine-associated gene expression in RNA-seq data indicated that KO cells exhibit higher expression levels of a subset of intestine marker genes, such as *CDX2*, Telomerase Reverse Transcriptase (*TERT*)[56], Doublecortin Like Kinase 1 (*DCLK1*)[57], and HOP Homeobox (*HOPX*)[58]. Here, we have also observed in WT + $H_2O_2$ cells increased intestinal marker expression, resembling the changes in KO cells (Supplementary Fig. 7I), and similarly to liver-related genes. Moreover, we found that *GPX2* KO cells display increased chromatin accessibility at the SRY-Box Transcription Factor 2 (*SOX2*) and *DCLK1 loci* compared to WT cells (Supplementary Fig. 7N). Together, these results suggest that, compared to WT cells, KO and WT + $H_2O_2$ cells exhibit an enhanced capacity for intestinal differentiation. However, based on the number of intestine- and liver-related changes in RNA-seq and ATAC-seq data, the liver-like progenitor characteristics appear to be preferred.

## GPX2 deficiency causes extracellular matrix alterations

We next investigated how changes in ROS metabolism influence cell fate determination. An analysis of scRNA-seq data from cells at PEN and SC-β cell stages revealed upregulation of ECM-associated genes in KO cells. These included Laminin Subunit Alpha 1 (*LAMA1*), Fibronectin 1 (*FN1*), and Laminin Subunit Beta 1 (*LAMB1*), as well as ECM signaling regulators, including Cadherin 1 (*CDH1*), Cadherin 2 (*CDH2*), and Glypican 3 (*GPC3*) (Fig. 6A–D). Consistent with these findings, a comprehensive analysis of the core matrisome (which includes ECM proteins, ECM-associated proteins, growth factors, and ECM-modifying enzymes[59]) at the SC-β cell stage showed increased gene set score in the ML, MLP1, and MLP2 clusters of KO cells. In WT cells, a similar increase in expression of core matrisome genes was observed in the PP2 cluster, albeit at a lower level. The ML cluster exhibited the highest

RNA levels of matrisome components compared to both KO endocrine cells and WT PP cells (Fig. 6E). To further investigate changes in ECM, we analyzed RNA-seq data from spontaneously differentiated WT, KO, and WT + $H_2O_2$ cells, focusing on ECM-associated marker genes. This revealed their upregulation in KO and WT + $H_2O_2$ cells (Fig. 6F, Supplementary Data 4). Additionally, changes in chromatin accessibility were observed for ECM markers, including Laminin Subunit Alpha 5 (*LAMA5*) and Fibulin 2 (*FN1*), among others in KO cells (Fig. 6G). Furthermore, immunostaining confirmed increased expression of LAMA1 alongside HNF4A in KO compared to WT cells (Fig. 6H). We also examined other ECM proteins, specifically LAMA5 and COL4A, and observed striking changes in their composition and formation of dense clusters specifically in *GPX2*-deficient cells (Fig. 6H). In summary, our results indicate substantial changes in ECM composition in KO cells, which may influence the direction of endoderm differentiation.

## BMP activation drives hepatic bias in GPX2-deficient PFG

To elucidate the molecular basis of the PFG differentiation bias observed in KO cells, we leveraged RNA-seq and scRNA-seq datasets to analyze differential signaling pathway activity across key developmental stages: PFG, PEN, and SC-β cell. Functional annotation of the upregulated genes in ML cells using Kyoto Encyclopedia of Genes and Genomes (KEGG) revealed enrichment in terms associated with liver function, including coagulation cascades[60,61], PPAR signaling pathway[62], protein digestion and absorption, platelet activation[63], and the metabolism of cholesterol and glutathione[64–66]. Notably, the TGF-β signaling pathway term, encompassing TGF-β and BMP signaling[67,68], emerged as one of the most significantly enriched pathways (Fig. 7A, Supplementary Data 3). PROGENy analysis revealed the significant upregulation of TGF-β and WNT pathway-responsive genes in KO cells, particularly in ML cluster (Fig. 7B, C, Supplementary Data 3). This is particularly relevant because the interplay between TGF-β, BMP, and WNT signaling pathways critically regulates liver-pancreas lineage decisions[35]. Analysis of transcription factor regulons[69] indicated elevated regulatory activity of SMAD9 (also known as SMAD8), SMAD5,

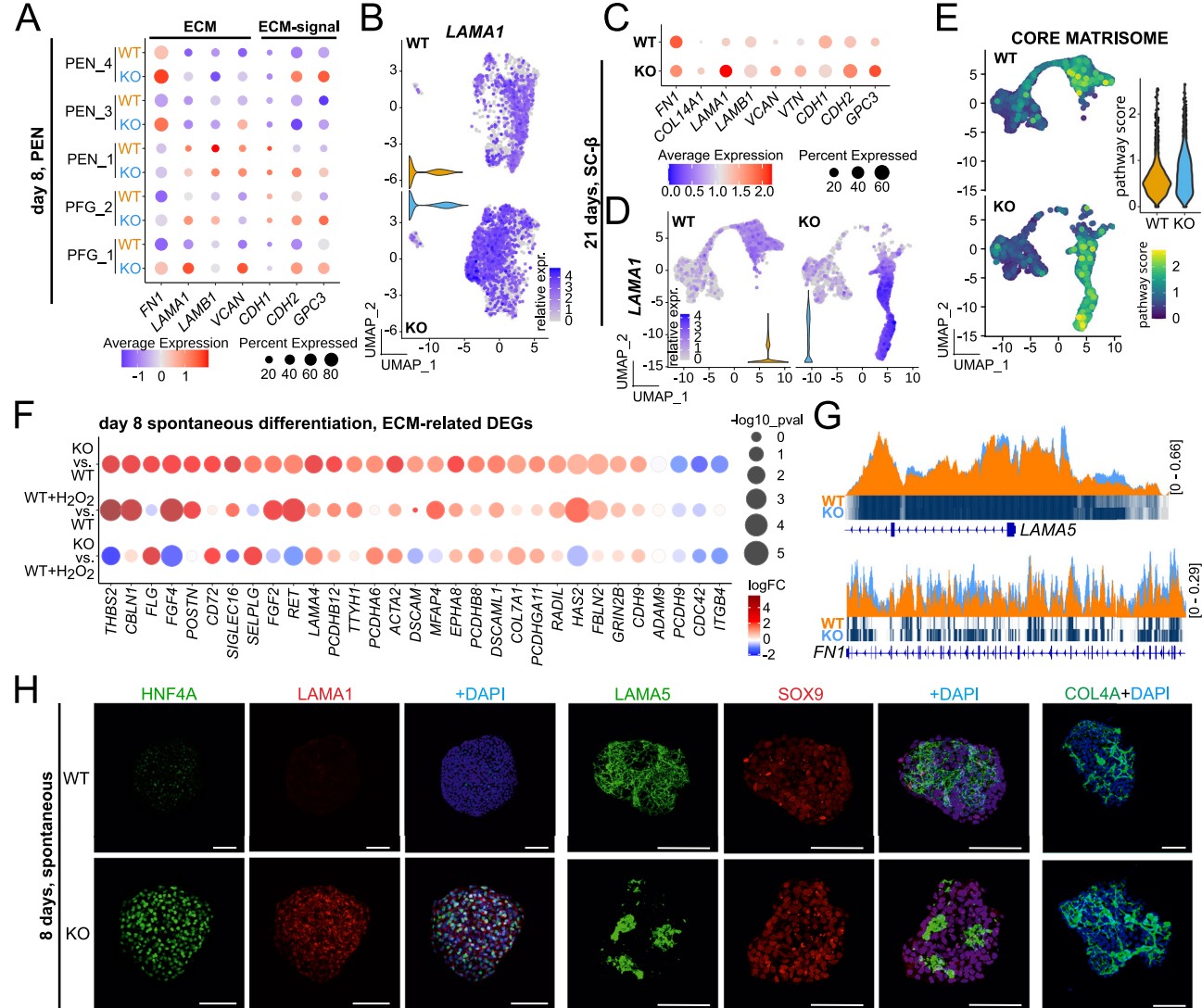

**Fig. 6 | ECM and ECM-associated proteins are dysregulated in GPX2-deficient cells. A** Dot plot represents expression of marker genes associated with ECM (*FN1*, *LAMA1*, *LAMB1*, *VCAN*) and ECM-signals (*CDH1*, *CDH2*, *GPC3*) across PEN_1, PEN_3, PEN_4, PFG_1 and PFG_2 clusters of WT and KO (clonal line KO1) PEN cells. PEN_1 - pancreatic endoderm 1, PEN_3 - pancreatic endoderm 3, PEN_4 - pancreatic endoderm 4, PFG_1 - posterior foregut 1, PFG_2 - posterior foregut 2. Genes with higher average expression are shown in red, and with lower are shown in blue. Dot size represents the percentage of cells expressing selected genes. **B** UMAP embedding projection shows the expression level of ECM marker gene *LAMA1* in PEN cells, based on scRNA-seq results from day 8 of differentiation. Cells with high gene expression are marked in blue. Violin plots representing *LAMA1* abundance are displayed for WT (orange) and KO (clonal line KO1, blue) PEN cells. **C** Dot plot represents the expression of marker genes associated with ECM (*FN1*, *COL14A1*, *LAMA1*, *LAMB1*, *VCAN*, *VTN*) and ECM-signaling (*CDH1*, *CDH2*, *GPC3*) in WT and KO (clonal line KO1) cells at the SC-β cell stage. Genes with higher average expression are shown in red, and with lower expression are in blue. Dot size represents the percentage of cells expressing a given gene. **D** UMAP shows the expression level of ECM marker gene *LAMA1* in WT and KO (clonal line KO1) cells at SC-β cell stage, based on scRNA-seq results from day 21 of differentiation. Cells with high *LAMA1* expression are marked in blue. Violin plots show *LAMA1* abundance in WT (orange)

and KO (blue) cells at the SC-β cell stage. **E** UMAP displays the core matrisome for WT (upper panel) and KO (clonal line KO1, lower panel) cells at SC-β cell stage. Cells exhibiting high pathway scores are marked in yellow, and those with low scores are shown in blue. A violin plot illustrates the distribution of pathway scores, with WT marked in orange and KO in blue. **F** Dot plot illustrates the differential expression of ECM-related genes across the following comparisons: KO (clonal line KO1) vs. WT, WT + $H_2O_2$ vs. WT, and KO vs. WT + $H_2O_2$. WT, WT + $H_2O_2$ and KO cells were differentiated spontaneously until day 8. Genes with higher average expression are depicted in red, and those with lower expression are shown in blue. Dot size corresponds to the -$\log_{10}$(*p*-value). Differences in gene expression between samples were calculated using a two-sided empirical Bayes moderated t-test (limma-voom), with *p*-values adjusted for multiple testing using the Benjamini−Hochberg false discovery rate (FDR) method. **G** ATAC-seq tracks highlight the *loci* of *LAMA5* and *FN1* in WT (orange) and KO (clonal line KO2, blue) cells differentiated spontaneously until day 8. The peaks represent normalized and combined biological replicates (*N* = 2). **H** Representative images of spontaneously differentiated WT and KO (clonal line KO1 and 2) cells (day 8) co-stained for HNF4A (green) and LAMA1 (red), LAMA5 (green) and SOX9 (red) or COL4A (green) proteins. DAPI marks nuclei (blue). Scale bar = 100 μm. *N* = 3 biological repeats. Source data are provided as a Source data file.

and SMAD1, critical for BMP signaling (Supplementary Fig. 8A, Supplementary Data 3). Increased activity of the TGF-β regulon was observed already at the PEN stage (Fig. 3G, H). Spontaneously differentiated *GPX2* KO cells also exhibited higher expression of TGF-β-related genes compared to WT cells (Fig. 7D), and increased activity of the SMAD1 and SMAD9 regulons (Supplementary Fig. 8B). Finally,

$H_2O_2$-treated WT cells displayed similar gene expression alterations, recapitulating the KO phenotype (Fig. 7D, Supplementary Fig. 8B, Supplementary Data 4). ATAC-seq data revealed changes in chromatin accessibility in a cohort of TGF-β regulatory genes, including *SMAD1*, Bone Morphogenetic Protein Receptor Type 2 (*BMPR2*), Left-Right Determination Factor 1 (*LEFTY1*)[70], *NODAL*, and Activin Membrane

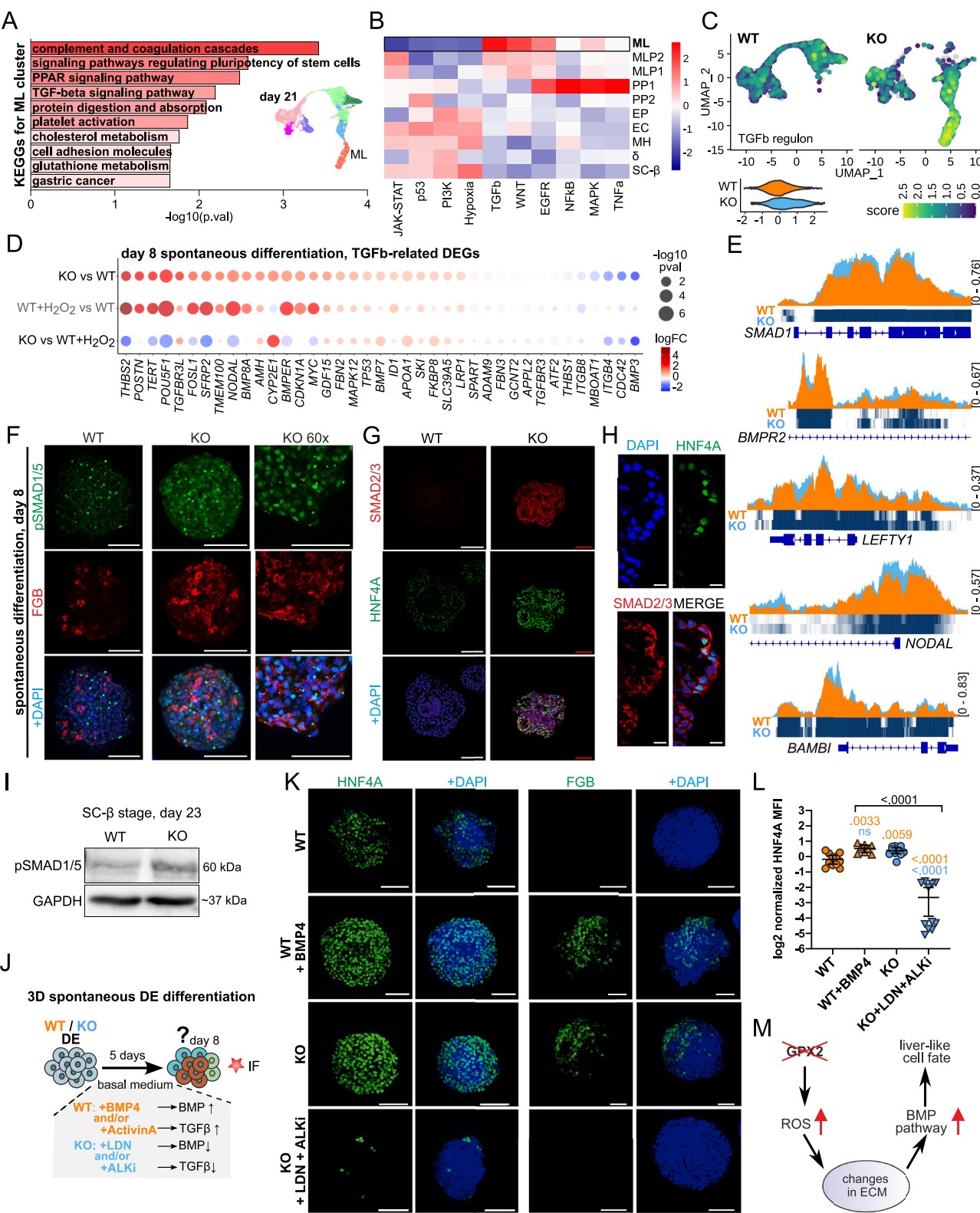

Bound Inhibitor (*BAMBI*)[71] (Fig. 7E, Supplementary Fig. 8C). Changes were also observed in TGF-β/ACTIVIN/NODAL negative regulators (Left-Right Determination Factor 2, *LEFTY2* and TGFB Induced Factor Homeobox 1 and 2, *TGIF1/2*) and BMP signaling enhancers/targets (Repulsive Guidance Molecule BMP Co-Receptor A, *RGMA*, Repulsive Guidance Molecule BMP Co-Receptor B, *RGMB*, Inhibitor of DNA Binding 1, *ID1*, Inhibitor of DNA Binding 2, *ID2*, and Inhibitor of DNA Binding 3, *ID3*) (Supplementary Fig. 8C). Increased expression of *ID1*,

*ID2*, and *ID3* was observed in the ML cluster of KO cells at the SC-β stage (Supplementary Fig. 8C). Finally, we have also identified changes at the *loci* of Fibrillin-1 (*FBN1*)[72] and Thrombospondin Type 1 Domain Containing 4 (*THSD4*, *ADAMTSL-6*)[73], which encode ECM proteins that contribute to ECM structures regulating latent TGF-β availability (Supplementary Fig. 8C). These together suggest that upon ECM remodeling, latent TGF-β complexes are released, leading to TGF-β activation in KO cells.

**Fig. 7 | The BMP pathway, but not the TGF-β pathway, is active in GPX2-deficient cells driving differentiation into liver-like progenitors. A** Bar plot shows 10 first molecular pathways identified by KEGG for the ML cluster of *GPX2* KO (clonal line KO1). Color intensity corresponds to the number of genes identified for each pathway. Enrichment analysis was conducted using only DEGs with a *p*-value of 0. UMAP-based embedding projection of the clustered dataset from the SC-β stage is also displayed. **B** Heatmap presents 10 enriched pathways for clusters identified at the SC-β stage of *GPX2* KO. ML cluster, characteristic for KO cells, is marked with a black frame. Enriched pathways and processes among markers for each cluster and DEGs were identified via PROGENy tool. **C** UMAP projection of the TGF-β regulon for WT and KO (clonal line KO1) cells at differentiation day 21. Cells with high TGF-β regulon activity are marked in yellow, and those with low activity are marked in blue. Each dot represents a single cell. A violin plot illustrates the distribution of pathway scores, with WT marked in orange and KO in blue. **D** Dot plot represents differential expression of TGF-β-related genes across the following comparisons: KO vs. WT, WT + $H_2O_2$ vs. WT, and KO vs. WT + $H_2O_2$, based on RNA-seq data. WT, WT + $H_2O_2$ and KO (clonal line KO1) cells were differentiated spontaneously until day 8. Genes with higher average expression are depicted in red, and with lower in blue. Dot size corresponds to the -$\log_{10}(p$-value). Differences in gene expression between samples were calculated using a two-sided empirical Bayes moderated t-test (limma-voom), with *p*-values adjusted for multiple testing using the Benjamini–Hochberg false discovery rate (FDR) method. **E** ATAC-seq tracks highlight the *loci* of *SMAD1*, *BMPR2P*, *NODAL*, and *BAMBI* in WT (orange) and KO (clonal line KO2, blue) cells differentiated spontaneously until day 8. The peaks represent normalized and combined biological replicates (*N* = 2). **F** Representative IF images of WT and KO spheroids differentiated spontaneously to day 8 and stained for pSMAD1/5 (green) and FGB (red). DAPI marks nuclei (blue). Scale bar = 100 μm. For KO cells (clonal line KO1), zoomed-in inserts are also shown. *N* = 3 biological repeats. **G** Representative IF images of WT and KO (clonal line KO1) spheroids differentiated spontaneously to day 8 and stained for HNF4A (green) and SMAD2/3 (red). DAPI marks nuclei (blue). Scale bar = 100 μm. *N* = 3 biological repeats. **H** Zoom-in of *GPX2* KO (clonal line KO1) cells spontaneously differentiated until day 8, shows cytoplasmic localization of SMAD2/3 (red), co-stained with HNF4A (green). DAPI marks nuclei. Scale bar = 10 μm. *N* = 3 biological repeats. **I** Western blot shows pSMAD1/5 protein levels in WT and KO (clonal line KO1) cells at SC-β stage (day 23). GAPDH was used as a loading control. *N* = 3 biological repeats. **J** Schematic of the experimental design to investigate the influence of BMP and TGF-β signaling pathways on spontaneous differentiation. Cells were differentiated to the DE stage and then maintained in basal medium. To activate BMP and TGF-β signaling, WT cells were treated with recombinant human BMP4 (25 μg/mL on day 4, followed by 3 ng/mL on days 5–8) and/or Activin A (100 ng/mL), respectively. To inhibit BMP and TGF-β signaling, KO cells were treated with LDN193189 (250 nM) and/or Alk5i II (RepSox, 10 μM), respectively. Cells were collected for immunostaining on day 8. **K** Representative images of WT and KO (clonal line KO1) cells spontaneously differentiated for 8 days, with or without modulation of BMP and TGF-β pathways. Cells were stained for HNF4A (green) or FGB (green), and nuclei were counterstained with DAPI. *N* = 3 independent experiments. Scale bar = 100 μm. To activate the BMP pathway, BMP4 protein was added at a concentration of 25 μg/mL on differentiation day 4, followed by 3 ng/mL on days 5–8. BMP pathway was inhibited with 200 nM LDN193189, while the TGF-β pathway was blocked using Alk5i II (10 μM) during differentiation days 5–8. **L** Dot plot shows quantification of mean fluorescence intensity (MFI) of HNF4A protein relative to DAPI in WT (orange) and KO (clonal line KO1, blue) spheroids spontaneously differentiated to day 8, with or without modulation of BMP and TGF-β pathways. Each dot represents one image. *N* = 3 independent experiments. Median values with 95% confidence intervals (CI) are shown. The *p*-values were calculated using an ANOVA test and values are shown on the plot. **M** Schematic overview of the proposed molecular mechanism driving GPX2-deficient cells into liver-like progenitors. *GPX2* KO increases ROS levels, which trigger structural changes in the extracellular matrix (ECM). These changes release BMP activators from the ECM, leading to BMP pathway activation. The activated pathway alters the differentiation process, promoting the development of a hepatic-like or non-pancreatic cell fate. Source data are provided as a Source data file.

The TGF-β signaling family, with its TGF-β/ACTIVIN/NODAL and BMP branches[72,74], uses distinct ligands to activate SMAD2/3[75] or SMAD1/5/9[76] signaling. IF staining revealed nuclear localization of phosphorylated SMAD1/5 (p-SMAD1/5) in FGB⁺ KO cells differentiated spontaneously and in KO cells at SC-β stage (Fig. 7F, I), indicating active BMP signaling. While total SMAD2/3 protein levels were elevated in KO cells, no nuclear localization was observed, suggesting inactive TGF-β signaling (Fig. 7G, H). No significant differences in SMAD4 expression were observed (Supplementary Fig. 8D). Increased BMP activity in KO cells likely results in preferential p-SMAD1/5/8 binding to SMAD4, depleting SMAD4 availability, and suppressing TGF-β activation[77].

To determine whether TGF-β or BMP signaling modulation recapitulates the KO phenotype, we treated WT DE cells with activin A (AA) and/or BMP4 to activate the TGF-β and BMP pathways, respectively. Conversely, KO DE cells were treated with Alk5iII and LDN193189, inhibiting the TGF-β and BMP pathways, respectively (Fig. 7J). BMP4 treatment of WT DE resulted in increased HNF4A and FGB expression. Combined inhibition of TGF-β and BMP signaling with LDN193189 and Alk5iII in KO cells suppressed these markers, reducing their levels below those in WT cells (Fig. 7K, L). Intriguingly, treatment of WT cells with BMP4 and AA, or AA alone, decreased HNF4A expression compared to untreated WT cells, reaching levels comparable to KO cells treated with inhibitors (Supplementary Fig. 8E). LDN193189 or Alk5iII alone in KO cells resulted in similar HNF4A levels as combined inhibition (Supplementary Fig. 8E). These findings suggest that the BMP pathway drives the KO cells towards liver-like lineages. In summary, GPX2 loss in PFG leads to increased oxidative stress, remodeling of ECM composition, allowing BMP release from its latent complex, and enhancing its activity. This ultimately increases the liver-like identity gene expression in PFG, biasing it towards hepatic-like fate (Fig. 7M).

## Discussion

Our investigation provides an exploration of GPX2 pivotal role in endoderm progenitor differentiation, revealing oxidative stress as a critical determinant of lineage-specific cell fate decisions. While previous investigations using *Gpx2* knockout mouse models predominantly focused on intestinal development[78], our research using hPSCs unveils an understanding of GPX2 functional significance in human pancreas organogenesis. Despite the broad expression of several GPX family members, GXP2 loss-of-function clearly skews the human endoderm specification.

During early embryogenesis, the endoderm (formed between E6.5 and E7.5 in mice) initially lacks commitment to specific organ domains. Wells and Melton[33] demonstrated this plasticity through explant studies where E7.5 anterior mouse endoderm, when placed in contact with posterior mesoderm, activates posterior endodermal gene programs and vice versa. While investigations have uncovered various clues about endodermal patterning into distinct organ domains, these findings have not yet coalesced into a comprehensive mechanistic understanding. Importantly, no simple, one-to-one relationship between transcription factors and organ emergence within the endoderm has been reported. Multiple transcription factors, including Gata4, FoxA3, Onecut1, Onecut2, and Prox1, are shared between pancreatic and hepatic endodermal domains rather than being uniquely restricted to either lineage[79]. This transcriptional overlap complicates our understanding of the molecular determinants driving specific organ identity during endodermal development. Here we demonstrate that commonly recognized as simply metabolic byproduct, ROS, play an important role in specification of neighboring domains of PFG endoderm within the hepatopancreatic region.

ROS have emerged as crucial molecular regulators in physiological processes, operating across multiple biological levels: from gene expression and protein translation to intricate protein-protein interactions. Their functionality extends beyond traditional conceptualizations of cellular damage, positioning ROS as signaling molecules that translate environmental cues into precise cellular responses. Hydrogen peroxide predominates as the primary ROS mediating intracellular signaling, functioning as a second messenger.

Its unique properties, including a relatively longer half-life and exceptional membrane permeability, enable $H_2O_2$ to integrate environmental signals and propagate them through downstream signal transduction cascades[80]. This molecular mechanism stands in contrast to historical perspectives that characterized ROS solely as indiscriminate oxidative agents capable of damaging proteins, fatty acids, and nucleic acids. In our study, intracellular oxidative stress was assessed using $H_2DCFDA$, a widely used ROS-sensitive fluorescent probe. While primarily detecting hydrogen peroxide, $H_2DCFDA$ can also respond to other ROS and is subject to time-dependent fluorescence decay. As our analysis focused on overall oxidative stress at a single time point rather than dynamic ROS fluctuations or specific ROS species, these limitations do not affect the interpretation of our findings regarding β-cell development.

Cellular ROS generation occurs within a meticulously regulated framework, primarily centered around the mitochondrial electron transport chain. The delicate balance between ROS production and neutralization is maintained through antioxidant systems. Glutathione, a ubiquitous tripeptide, represents a central component of this regulatory network, working in concert with enzymatic systems like glutaredoxin and thioredoxin to reduce oxidized proteins and hydrogen peroxide. Cytosolic antioxidant enzymes, including GPX2 in PFG, play a critical role in this molecular regulatory process, acting as molecular sentinels, directly neutralizing ROS by accepting electrons and preventing potentially deleterious oxidative modifications. The intricate interplay between ROS generation and scavenging systems underscores the complexity of cellular redox homeostasis.

Our findings contribute to the evolving understanding of cellular differentiation, demonstrating how oxidative stress can serve as a critical determinant in cell fate decisions. By revealing the role of GPX2 in endoderm progenitor differentiation, we provide insights into the molecular mechanisms underlying lineage specification and cellular plasticity. The progressive increase in ROS levels throughout pancreatic differentiation underscores the importance of precise redox balance, resonating with previous findings where decreased ROS production correlates with reduced β-cell differentiation efficiency[81]. Our study suggests that elevated oxidative stress during embryogenesis may disrupt the α/β cell proportion, potentially predisposing individuals to impaired glucose homeostasis and increasing the risk of developing conditions such as type 2 diabetes and metabolic syndrome later in life. Furthermore, GPX2 functions as a temporal gatekeeper of cell fate decisions during endoderm differentiation, with distinct stage-specific roles. While GPX2 deficiency affects lineage commitment at the PFG stage (days 6-8), leading to hepatic-like and multilineage progenitors, we observe comparable β-cell differentiation efficiency at day 21. This suggests that oxidative stress plays different roles at distinct developmental stages, acting as a fate determinant during early specification but potentially serving beneficial functions during later maturation. Critically, INS$^+$ KO cells at the SC-β stage do not co-express hepatic markers such as FGB, demonstrating that pancreatic and hepatic-like populations are distinct. Our scRNA-seq data further support this, showing hepatic-like markers enriched in a separate multilineage cluster while SC-β clusters in KO cells exhibit transcriptional profiles similar, yet not identical, to WT. These observations establish a stage-dependent model where GPX2 and oxidative stress critically determine whether cells enter the pancreatic versus non-pancreatic lineage during early specification, but do not impair β-cell maturation once that fate is established. This temporal specificity identifies the PFG stage as the critical window when oxidative stress management is essential for pancreatic fate specification, with important implications for optimizing directed differentiation protocols.

It remains to be established whether a gradient of oxidative stress exists in the developing endoderm and if it plays a role in modulating cell fate. Increasing evidence implies that controlling the complexity of differentiating cell environments, including biophysical properties, chemical milieu such as oxidative stress or metabolic nuances, is critical for in vitro hPSC differentiation. We recently demonstrated that changes in cell-cell and cell-ECM adhesion impact different events/stages of hPSC differentiation, including three-germ-layer specification following the pluripotency exit in the gastruloid model, but also pancreatic to endocrine progenitor transition[18].

Critical to our understanding are the intricate molecular mechanisms underlying GPX2-mediated fate determination. ROS can signal directly to proteins predominantly via amino acid oxidation, with cysteine residue modification representing the most prevalent mechanism. These oxidative modifications establish ROS as crucial regulators of cellular signaling, transforming proteins into dynamic redox sensors capable of profound functional alterations. Redox-sensitive proteins undergo transformative conformational changes that influence critical cellular processes, including protein function, stability, subcellular localization, and intermolecular interactions. Hypoxia-inducible factor 1α (HIF-1α) exemplifies this regulatory mechanism. For instance, in the heart, oxygen tension fluctuations and subsequent ROS elevation can trigger metabolic reprogramming, driving a shift from fatty acid oxidation to glucose metabolism[82].

Broad-acting phosphatases and kinases, for instance, Cys oxidation abrogates AKT phosphorylation and, therefore, function[83], or mTOR can be activated by Cys oxidation[84,85]. These molecular interactions suggest ROS might serve as a communication network, coordinating mitochondrial activities with nuclear processes, including chromatin remodeling, gene expression, cell cycling, DNA repair, and cellular differentiation. Here, we demonstrate that ECM remodeling emerges as an important driver of lineage plasticity. GPX2 loss disrupts the delicate balance between TGF-β and BMP signaling pathways, with increased oxidative stress facilitating BMP release from latent complexes. Enhanced BMP signaling potentially sequesters SMAD4, a crucial co-factor correlating with preferential hepatic differentiation. This molecular mechanism provides unprecedented insights into the complex process of lineage specification, particularly the intricate balance between pancreatic and hepatic-like cell fate determination.

ROS levels must be tightly regulated to prevent cellular damage. We observed a three-fold increase in ROS levels during the β-cell differentiation. Direct comparison to the magnitude of ROS increases in other cell types or cellular processes is not straightforward; however, the three-fold increase in ROS levels most a substantial change compared to what might be considered baseline or typical fluctuations in many other cell types or developmental stages. Cells with a long lifespan, including stem cell populations, often have mechanisms to minimize ROS accumulation to prevent damage over time[86]. Highly metabolically active cells, such as neutrophils during inflammatory responses or heart muscle cells during periods of increase activity, naturally produce temporary but substantial spikes in ROS as they perform their specialized functions. However, these elevated ROS levels are typically well-controlled by antioxidant defense mechanisms[87,88]. By comparison, a persistent threefold elevation in ROS within a relatively quiescent cell type like a developing β-cell would represent a notably high oxidative state. Cancer cells often exhibit elevated basal ROS levels compared to their normal counterparts. The magnitude of ROS increase in cancer cells can vary widely, but a sustained three-fold increase would be within the range observed in some cancers[89].

In *GPX2* KO cells, the transcriptional landscape is shifted, particularly affecting critical regulatory genes. Decreased HHEX levels in GPX2-deficient cells likely reduce pancreatic differentiation propensity while enabling hepatic-like commitment. This pattern mirrors the

consequences of HHEX deficiency, where *HHEX* KO inhibits efficient pancreatic fate acquisition and promotes hepatic marker expression in hPSC-derived endoderm[7]. Furthermore, human genetic studies identified several pancreatic developmental regulators, mostly transcription factors, including GATA4, GATA6[90–94] and HFN4A[95,96], which exhibit differential dosage-dependent effects between mice and humans, suggesting critical interspecies differences in pancreatic development.

While the role of oxidative stress and GPX2 in endoderm specification is intriguing, further investigations into the molecular mechanisms would strengthen our understanding of this signaling axis. For instance, an examination of epigenetic status might yield important observations. To date, depletion of the m⁶A demethylase ALKBH5 has been shown to impair pancreatic lineage specification, while concurrently promoting the upregulation of hepatocyte-associated genes[97]. Also, BMP signaling may represent just one of several pathways controlling GPX2-dependent induction of liver-associated gene expression in pancreatic foregut.

Our findings contribute to the evolving understanding of cellular differentiation, demonstrating oxidative stress as a critical determinant in cell fate decisions. By revealing the nuanced role of GPX2 in endoderm progenitor differentiation, we provide unprecedented insights into the molecular mechanisms underlying lineage specification and cellular plasticity.

## Methods

### Ethics statement
All experiments with human embryonic stem cells were performed in accordance with the relevant guidelines and regulations of the Local Bioethics Committee of the Medical University of Poznan, Poland. The hESC lines used in this study are HUES8 (sex: male, NIH registry 0021), derived by the Harvard Stem Cell Institute and Dr. Douglas A. Melton laboratory, and H1 (sex: male, NIH registry 0043), derived by the University of Wisconsin and Dr. James Thomson laboratory. The manuscript reports on expression studies with human fetal pancreas. Tissue was obtained under IRB approval IRB-3097 to M.B., and was processed at Baylor College of Medicine, Houston, TX, USA.

### Cell culture
hPSCs used in this study included human H1 (derived by Dr. James Thomson, University of Wisconsin), HUES8 (derived by Dr. Doug Melton at Harvard University), and iCas9-HUES8 (a doxycycline-inducible Cas9-expressing HUES8 line generated by Dr. Danwei Huangfu, Memorial Sloan Kettering Cancer Center) cell lines. The hPSCs were cultured in StemFlex medium (Thermo Fisher Scientific, Netherlands) on Geltrex-coated plates (Thermo Fisher Scientific, Netherlands) at 37 °C in a 5% CO₂ atmosphere. Cells were passaged every 3–4 days using PBS-EDTA and routinely tested for mycoplasma via PCR assay, with all tests returning negative. All experiments with human embryonic stem cells were performed in accordance with the relevant guidelines and regulations of the Local Bioethics Committee of the Medical University in Poznan, Poland.

### *GPX2* gene deletion in hPSCs
Three different sgRNAs targeting the *GPX2* gene were designed using Benchling software and produced in-house, according to the protocol previously published[18]. Shortly, a PCR product consisting of T7 RNA polymerase promoter, target-specific sgRNA, and common sgRNA sequence was used for in vitro transcription to produce sgRNA for lipofection. The sequences of sgRNA are listed in Supplementary Table 1. HUES8-iCas9 cells at the confluence of 70%, treated previously with doxycycline for 24 h, were transfected using Lipofectamine RNAiMax (Thermo Fisher Scientific, Netherlands) and cultured for 1 day. Next, cells were seeded at a single-cell density on a

60 mm culture dish coated with Geltrex and cultured for 7 days. Single colonies were transferred to a 96-well plate. The PCR was performed on genomic DNA isolated from individual colonies with the primers flanking the *GPX2* gene sequence (*GPX2*_F: 5′-TGATGGGGCAAGTAGGAGTC-3′, *GPX2*_R: 5′-CCACACCTGCCCTTTATTGG-3′). The *GPX2* KO hPSC lines were established by selecting homozygous clones with desired mutations, identified based on amplified DNA electrophoresis and Sanger sequencing.

### Pancreatic differentiation of hPSCs in 3D culture
For in vitro pancreatic differentiation in 3D culture, we employed a previously established protocol[98] with modifications[17,18]. hPSCs were dissociated into single-cell suspensions using TrypLE Select (Thermo Fisher Scientific, Netherlands) and plated at a density of $3.5 \times 10^6$ cells/well in E8 medium (Thermo Fisher Scientific, Netherlands) supplemented with 10 μM Y-27632 (Peprotech, USA) on low-adhesion, non-treated 6-well plates (Eppendorf, Germany) placed on an orbital shaker at 37 °C, 110 rpm, under a 5% CO₂ atmosphere. The medium was replaced the next day with E8 without Y-27632. On the subsequent days spheroids were washed in DMEM/F12 medium (Corning, USA) and the medium was changed, as follows: Basal medium: S1 – MCDB131 (Corning, USA), 1% Glutamax (Thermo Fisher Scientific, Netherlands) + 1% Pen/Strep (Thermo Fisher Scientific, Netherlands), 2.44 mM glucose (Merck Millipore, USA), 29 mM sodium bicarbonate (Merck Millipore, USA), 2% BSA (Capricorn Scientific, USA), 2 μl/100 mL ITS-X (Thermo Fisher Scientific, Netherlands). S2 – MCDB131 (Corning, USA), 1% Glutamax (Thermo Fisher Scientific, Netherlands) + 1% Pen/Strep (Thermo Fisher Scientific, Netherlands), 2.44 mM glucose (Merck Millipore, USA), 13 mM sodium bicarbonate (Merck Millipore, USA), 2% BSA (Capricorn Scientific, USA), 2 μl/100 mL ITS-X (Thermo Fisher Scientific, Netherlands). S3 – MCDB131 (Corning, USA), 1% Glutamax (Thermo Fisher Scientific, Netherlands) + 1% Pen/Strep (Thermo Fisher Scientific, Netherlands), 2.44 mM glucose (Merck Millipore, USA), 13 mM sodium bicarbonate (Merck Millipore, USA), 2% BSA (Capricorn Scientific, USA), 500 μl/100 mL ITS-X (Thermo Fisher Scientific, Netherlands). S4 – MCDB131 (Corning, USA), 1% Glutamax (Thermo Fisher Scientific, Netherlands) + 1% Pen/Strep (Thermo Fisher Scientific, Netherlands), 2.44 mM glucose (Merck Millipore, USA), 29 mM sodium bicarbonate (Merck Millipore, USA), 2% BSA (Capricorn Scientific, USA), 500 μl/100 mL ITS-X (Thermo Fisher Scientific, Netherlands), 5 μg/mL heparin sulfate (Sigma Aldrich, USA), S5 – MCDB131 (Corning, USA), 1% Glutamax (Thermo Fisher Scientific, Netherlands) + 1% Pen/Strep (Thermo Fisher Scientific, Netherlands), 2.44 mM glucose (Merck Millipore, USA), 2% BSA (Capricorn Scientific, USA), 260 nM zinc sulfate (Merck Millipore, USA), 5 μg/mL heparin sulfate (Sigma Aldrich, USA), 1x Trace Elements A (Corning, USA), 1x Trace Elements B (Corning, USA). Day 1: S1 + 3 μM CHIR99021 (Peprotech, USA) + 100 ng/mL Activin A (Peprotech, USA) + 250 μM vitamin C (Sigma Aldrich, USA); Days 2-3: S1 + Activin A; Days 4-6: S2 + 50 ng/mL KGF (Peprotech, USA) + 1.25 μM IWP2 (Selleckchem, USA) + 250 μM vitamin C; Day 7: S3 + 50 ng/mL KGF + 2 μM retinoid acid (Peprotech, USA) + 500 nM PdBu (Tocris, United Kingdom) + 250 nM SANT-1 (Tocris, United Kingdom) + 200 nM LDN193189 (Peprotech, USA) + 250 μM vitamin C + 10 μM Y-27632; Days 8-12: S3 + 5 ng/mL Activin A + 50 ng/mL KGF + 100 nM retinoid acid + 250 nM SANT-1 + 1.25 μM IWP2 + 100 ng/mL EGF (Peprotech, USA) + 10 mM nicotinamide (Sigma Aldrich, USA) + 250 μM vitamin C; Days 13-19: S4 + 100 nM retinoic acid (Peprotech, USA) + 250 nM SANT-1 (Tocris, United Kingdom) + 1 μM LY-411575 (Adooq Bioscience, USA) + 1 μM T3 (Sigma Aldrich, USA) + 10 μM Alk5iII (Adooq Bioscience, USA) + 20 ng/mL betacellulin (Peprotech, USA) + 250 μM vitamin C; Day 20-22: S5 without additional components. Experiments were conducted on at least two independent hPSC clonal lines. To induce oxidative stress, H₂O₂ (Sigma-Aldrich, USA) in the final concentration range from 10 nM to 10 μM was added to the medium once on the 5th day of differentiation of

WT cells. To estimate the effect of selenium, sodium selenite (Sigma-Aldrich, USA) in the concentration from 1 pM to 1 nM was supplemented to the medium for 3 days starting from the 5th day of differentiation.

## Pancreatic hPSC differentiation in 2D culture

For in vitro pancreatic differentiation in 2D culture, we employed a protocol established previously[98] with modifications[17,18]. hPSCs were dissociated into single-cell suspensions using TrypLE Select (Thermo Fisher Scientific, Netherlands) and plated at a density of $2 \times 10^6$ cells/well in Stem Flex medium (Thermo Fisher Scientific, Netherlands) supplemented with 10 μM Y-27632 (Peprotech, USA) on Geltrex-coated 6-well plates (Eppendorf, Germany) under a 5% $CO_2$ atmosphere. On the subsequent days cells were washed in DMEM/F12 medium (Corning, USA) and the medium was changed, as follows: Basal medium: S1 – MCDB131 (Corning, USA), 1% Glutamax (Thermo Fisher Scientific, Netherlands) + 1% Pen/Strep (Thermo Fisher Scientific, Netherlands), 8.88 mM glucose (Merck Millipore, USA), 14 mM sodium bicarbonate (Merck Millipore, USA) and 0.1% BSA (Capricorn Scientific, USA). S2 – MCDB131 (Corning, USA), 1% Glutamax (Thermo Fisher Scientific, Netherlands) + 1% Pen/Strep (Thermo Fisher Scientific, Netherlands), 4.44 mM glucose (Merck Millipore, USA), 14 mM sodium bicarbonate (Merck Millipore, USA) and 0.1% BSA (Capricorn Scientific, USA). S3 – MCDB131 (Corning, USA), 1% Glutamax (Thermo Fisher Scientific, Netherlands) + 1% Pen/Strep (Thermo Fisher Scientific, Netherlands), 2.44 mM glucose (Merck Millipore, USA), 20.8 mM sodium bicarbonate (Merck Millipore, USA), 2% BSA (Capricorn Scientific, USA) and 500 μl/100 mL ITS-X (Thermo Fisher Scientific, Netherlands). Day 1: S1 + 3 μM CHIR99021 (Peprotech, USA) + 100 ng/mL Activin A (Peprotech, USA) (Sigma Aldrich, USA); Days 2-4: S1 + Activin A; Days 5-6: S2 + 50 ng/mL KGF (Peprotech, USA) + 1.25 μM IWP2 (Selleckchem, USA) + 250 μM vitamin C; Day 7: S3 + 50 ng/mL KGF + 2 μM retinoid acid (Peprotech, USA) + 1 μM PdBu (Tocris, United Kingdom) + 250 nM SANT-1 (Tocris, United Kingdom) + 200 nM LDN193189 (Peprotech, USA) + 250 μM vitamin C. To induce oxidative stress, $H_2O_2$ (Sigma-Aldrich, USA) in the final concentration range from 10 nM to 10 μM was added to the medium once on the 5th day of differentiation of WT cells. To check the effect of selenium, sodium selenite (Sigma-Aldrich, USA) in the concentration from 1 pM to 1 nM was supplemented to the medium for 3 days starting from the 5th day of differentiation. WT cells treated with 100 pM Se and KO cells treated with 10 nM $H_2O_2$ were used as controls in this experiment.

## Spontaneous differentiation of hPSC-derived endoderm

hPSCs were differentiated to DE as described above. Starting from day 4, cells were maintained in S2 basal medium for 3 days, followed by S3 basal medium for 1 day. Spheroids were harvested on day 8 of differentiation. To evaluate the effects of oxidative stress modifications, we supplemented the basal medium with 10 μM $H_2O_2$ (Sigma-Aldrich, USA) on day 5, or with 100 pM sodium selenite (Sigma-Aldrich, USA) from day 5 to 8. TGF-β and BMP pathways were modified by the medium supplementation with 10 ng/mL Activin A, 25 μg/mL BMP4 (Peprotech, USA) in differentiation day 4 and 3 ng/mL BMP4 in differentiation days 5–8, 200 nM LDN193189 (Peprotech, USA) and 10 μM Alk5i II (RepSox) (Adooq Bioscience, USA) in different combinations.

## Western blot

Cells were lysed in the buffer composed of 60 mM Tris (Bio-Shop, Canada), 2% SDS (Bio-Shop, Canada), 10% sucrose (Bio-Shop, Canada), and 1% protease inhibitor cocktail (Sigma Aldrich, USA), followed by sonication for 15 s in 8 cycles with 1 min pause between cycles and centrifugation at $12,000 \times g$ for 10 min. Protein concentration was determined using a DS-11 spectrophotometer (DeNovix, USA). 30 μg of cell lysates in Bolt LDS buffer (Thermo Fisher Scientific, Netherlands)

were incubated for 5 min at 95 °C and then loaded on a 12% SDS-PAGE gel. Electrophoresis was carried out at 75 V for the first 15 min and 125 V for an additional 1 h, with a constant current intensity of 30 mA. Proteins were transferred onto the PVDF (Thermo Fisher Scientific, Netherlands) membrane using the BioRad Trans-Blot Turbo Transfer System for 10 min. Membranes were blocked with 5% non-fat dry milk or 1–3% BSA in TBS-Tween 20 followed by overnight incubation with a primary antibody. On the next day, membranes were probed with HRP-conjugated secondary antibodies. All antibodies are listed in Supplementary Table 2. Western blot results were visualized using the enhanced chemiluminescent visualization (ECL) system (Thermo Scientific Pierce) or SuperSignal West Femto Maximum Sensitivity Substrate (Thermo Fisher Scientific, Netherlands). Signal detection was conducted using the G: Box System (Syngene, India). Western blots were quantified in ImageJ software.

## Flow cytometry analysis of oxidative stress

The hPSCs and spheroids were dissociated into a single-cell suspension by incubation in TrypLE Select (Thermo Fisher Scientific, Netherlands) for 5 min with shaking, followed by PBS wash. Subsequently, live cells were stained with 0.5 μM H2CDFDA (Invitrogen, Netherlands) to assess ROS level or with 1 μM MitoSOX Mitochondrial Superoxide Indicators (Invitrogen, Netherlands) to detect mitochondrial ROS level. FC analysis was performed using Agilent NovoCyte (Agilent Technologies, USA), acquiring at least 5000 events per sample. Data analysis, including gating, quantification, and generation of density plots/histograms, was performed using the NovoFast program (Agilent Technologies, USA).

## Single-cell RNA sequencing

Spheroids at the PEN and SC-β cell stages, both WT and KO, were dissociated into single cells using 1% trypsin in PBS at 37 °C. scRNA-seq for the PEN stage was done using a combinatorial indexing method from SCALE Biosciences (ScaleBio, USA). Briefly, approximately 250,000 cells of WT PEN and approximately 500,000 cells of KO PEN cells were washed once with ice-cold PBS, centrifuged, and resuspended in 500 μL PBS. Cells were fixed by vortexing with simultaneous addition of Complete Cell Fixation Solution (Fixation Reagent1, ice-cold 100% methanol, Stanlab, Poland) with DEPC, followed by 15 min incubation on ice and then washed with WRS Buffer (Scale Biosciences) and resuspended in WRS Buffer. Libraries for single-cell RNA sequencing of PEN cells were conducted with the ScaleBio Single Cell RNA Sequencing Kit according to the manufacturer's protocol. Libraries for the SC-β cell stage scRNA-seq were prepared using the 10xGenomics kit following the manufacturer's protocol. In this case, the single-cell transcriptome was tagged with a UMI barcode using a droplet-based method in the Chromium system (10xGenomics, USA). For both library preparation methods, the quality and quantity of cDNA libraries were assessed using an Agilent TapeStation with High Sensitivity D1000 ScreenTape (Agilent Technologies, USA). Libraries were paired-end sequenced with a depth of 10,000 (ScaleBio) or 40,000 (10xGenomics) reads per cell. 2500 and 4042 cells from the PEN stage and 6128 and 5610 cells from the SC-β cell stage were sequenced for WT and KO, respectively. Initial processing of acquired data was performed using a manufacturer's pipeline, i.e., NextFlow pipeline for ScaleBio and CellRanger for 10xGenomics, the latter, including WT and KO data aggregation with the "cellranger agrr" algorithm. Further data processing analysis for both platforms, including clustering, visualization, marker identification, differential expression analysis, and pathway enrichment were performed using the Seurat v5.0.3[99], DElegate v1.2.1[100], escape v2.5.5[101], decoupleR v 2.14[69], and UCell v2.12[102] packages in R Studio. Enriched pathways and processes among markers for each cluster and DEGs were identified via the Kyoto Encyclopedia of Genes and Genomes (KEGG), Biological Processes (BP), Wiki Pathway, CollecTRI[103], and PROGENy[41] databases. The functional

enrichment analysis was performed in GeneCodis, Enrichr[104], or Appyter[105] interactive platforms. A pseudotime trajectory was performed using Monocle3[106]. The significance cutoff value (adj. *p*-value < 0.05) was utilized in functional enrichment analysis.

Human fetal pancreas PCW7-11 (OMIX001616)[36] and PCW12-14 (GSE197064)[107], datasets were imported into Seurat, followed by subclustering of the epithelial lineages and further analyses in Seurat as described above. Mouse E6.5-8.5 scRNA-seq data[19] was analyzed using Bioconductor MouseGastrulationData package in R. Mouse E14.5 pancreas scRNA-seq dataset was generated in our previous work[20] (GEO GSM2689399).

## Immunofluorescence staining

Human pancreas sections were processed at Baylor College of Medicine, Houston, TX (USA) with IRB H-3097 approval granted to Dr. Malgorzata Borowiak. Donor identities were encrypted, and the data were analyzed anonymously. The human 10.6- and 13-week fetal pancreas samples were fixed in 4% paraformaldehyde/PBS for 4 h, washed with PBS, soaked in 30% sucrose, and embedded in TissueTek. Sections (12 μm-thick) were cut onto Superfrost Plus-coated glass slides and stored at −80 °C.

Cell monolayers were fixed with 4% paraformaldehyde/PBS for 15 min at room temperature and washed with PBS. Fixed cells were permeabilized by 0.5% Triton X-100 (BioShop, Canada) in PBS for 15 min, blocked for 45 min in 3% BSA (BioShop, Canada) + 0.1% Tween-20 (BioShop, Canada) in PBS, and then incubated overnight at 4 °C with primary antibodies diluted in 5% normal donkey serum (NDS, Jackson Immunoresearch, UK). The next day, cells were washed three times in 0.1% Tween 20 in PBS and incubated with secondary antibodies conjugated with Alexa Fluor 488, TRITC, or Alexa Fluor 647 (diluted 1:400 in 5% NDS) for 2 h at room temperature. The excess secondary antibody was removed by two washes in 0.1% Tween 20 in PBS, and the samples were incubated with DAPI (Sigma Aldrich, USA) as a counterstain. To stain lipids, 1 μg/mL Nile Red (AAT Bioquest, USA) was added to the medium, and cells were incubated at 37 °C for 15 min, then washed with PBS and fixed.

3D spheroids were fixed with 4% paraformaldehyde/PBS for 45 min at 4 °C, washed once with 0.1% Tween-20 in PBS, and suspended in wash buffer (SWB: 2% BSA, 0.1% Triton X-100, 0.025% SDS, 1 × PBS). Fixed spheroids were blocked in SWB for 45 min at 4 °C and incubated overnight with a primary antibody at 4 °C on an orbital shaker. The next day, the spheroids were washed 3 times with SWB solution for 1 h each at 4 °C and incubated with secondary antibodies conjugated with Alexa Fluor 488, TRITC, or Alexa Fluor 647 diluted 1:400 and DAPI (1:10,000) in SWB, overnight at 4 °C on an orbital shaker. On the next day, the excess of the secondary antibody was removed by 2 washes in 0.1% Tween-20 in PBS, first rapid, second for 1 h at 4 °C on an orbital shaker. For imaging, spheroids were kept in PBS on a high-content imaging glass-bottom 96-well plate (Corning, USA). The primary and secondary antibodies used in the study are listed in Supplementary Table 3. Representative images shown in Figures and Supplementary information were chosen from at least 3 biological repeats for each condition and 3 photograpies per repeat, or as stated in the corresponding quantitative analysis.

## Microscopy imaging

Images were obtained with an epifluorescence, confocal, or light sheet microscope. Brightfield images were taken with a Leica DM IL-Led (Leica, Germany) microscope with N Plan Fluor 4x/0.12, N Plan Fluor 10x/0.30, N Plan Fluor 20x/0.40, and N Plan Fluor 40x/0.60 lenses and a JENOPTIK GRYPHAX series ProgRes camera (JENOPTIK, Germany). Fluorescence images were obtained with confocal microscope Nikon A1Rsi (Nikon, Germany) microscope with Plan Plan Apo 10x/0.45 DIC N1, Plan Apo VC 20x/0.75 DIC N2, Apo 40x/1.25 WI λS DIC N2, Plan Apo VC 60x/1.4 Oil DIC N2 and Plan Apo VC 100x/1.4 Oil DIC N2 lenses and with Nikon NIS Elements AR 5.21.01 64-bit software (Nikon, Germany) or with confocal microscope LEICA Stellaris 8 (Leica Camera, Germany) with HC PL APO 20x/0.75 CS2 Air, HC PL IRAPO 40x/1.10 W CORR, PL APO 63x/1.20 W CORR CS2 or PL APO 63x/1.20 W CORR CS2 lenses and with Leica LAS X Office 1.4.7 64-bit software (Leica Camera, Germany) or with light sheet fluorescence microscopy (LSFM) ZEISS Lightsheet 7 (Zeiss, Germany) with W Plan-APO 20x/1,0 DIC lens, in water chamber and with ZEN Black 3.1 software for imaging and ZEN Blue 3.1 software for data analysis (Zeiss, Germany).

## Doxycycline-inducible GPX2 overexpression in hPSCs

*GPX2* cDNA was amplified from total WT PEN cDNA with primers consisting of *GPX2* coding sequence, FLAG sequence at the 3′ end, and adapters to the plasmid. Primers consisted of the following sequences:

F_GPX2_OE: 5′- CTTTAAAGGAACCAATTCAGccaccATGGCTTT CATTGCCAAGTCCTTC −3′,

R_GPX2_OE: 5′- AAGCTGGGTCTAGATATCTTActtatcgtcgtcatccttg taatcacgTATGGCAACTTTAAGGA −3′. Primers were designed using the Benchling Assembly Wizard tool. *GPX2* cDNA was amplified via PCR using Q5 polymerase (NEB, USA) following the manufacturer's instructions, with two-step PCR cycling conditions. The cycling conditions were: 98 °C for 30 s, followed by 15 cycles of 98 °C for 15 s, 63 °C for 20 s, 72 °C for 90 s, 25 cycles of 98 °C for 15 s, 72 °C for 15 s, 72 °C for 90 s, followed by final extension at 72 °C for 5 min. Monarch DNA Gel Extraction Kit (NEB, USA) was used for the resulting *GPX2* PCR product purification. Gateway pENTR 1 A plasmid (Invitrogen, Netherlands) was digested with the restriction enzymes SalI and XhoI (Thermo Fisher Scientific, Netherlands). To ligate the *GPX2* PCR product with vector NEBuilder HiFi DNA Assembly Cloning Kit (NEB, USA) was used. The resulting Gateway pENTR 1 A plasmid containing the *GPX2* cDNA sequence was subjected to an LR recombination reaction with plasmid PB-TA-ERP2[108] (gift from Dr. Knut Woltjen, Addgene, USA, 80477), leading to the acquisition of the final plasmid (pENTR-h*GPX2*-FLAG plasmid) for establishing doxycycline-inducible GPX2 overexpression in HUES8-iCas9 and H1 hPSCs. Next, HUES8-iCas9 and H1 hPSCs were transfected with the pENTR-h*GPX2*-FLAG plasmid and pCMV-hyPB plasmid[109] (gift from Dr. Allan Bradley), carrying the transposase enzyme using the Lipofectamine 3000 kit (Thermo Fisher Scientific, Netherlands), following manufacturer's instructions. Finally, cells were treated with 1 μg/mL puromycin to select positively transformed cells OE hPSCs (GPX2 OE). To induce GPX2 overexpression, cells were treated with 1 μg/mL doxycycline for 24 h.

## Flow cytometry

Cells were dissociated into a single-cell suspension using TrypLE Select (Thermo Fisher Scientific, Netherlands). Fixation was carried out by incubating the cells with 4% paraformaldehyde in PBS containing 0.1% saponin for 45 min at 4 °C, followed by washing with PBS supplemented with 0.1% saponin and 1% BSA (SBP). The cells were then resuspended and incubated overnight at 4 °C on a roller with primary antibodies diluted in SBP. On the following day, cells were washed twice with SBP and incubated for 1 h at room temperature with secondary antibodies (Alexa Fluor 488, or Alexa Fluor 647; Jackson ImmunoResearch, UK) diluted 1:5,000–1:10,000 in SBP. After removing unbound secondary antibodies by a final PBS wash, cells were resuspended in PBS and immediately subjected to FC. Data were acquired using a NovoCyte Flow Cytometer (Agilent Technologies, USA). FC data processing, including gating, quantification, and generation of density plots or histograms, was performed using NovoExpress v1.6.2 (Agilent Technologies, USA). Details of the primary and secondary antibodies are provided in Supplementary Table 4.

## Bulk RNA sequencing

Total RNA was extracted using the TRIzol Reagent (Thermo Fisher Scientific, Netherlands) following the phenol/chloroform method. The RNA was dissolved in RNase-free water and treated with DNase I (Thermo Fisher Scientific, Netherlands). Its concentration and quality were assessed spectrophotometrically by measuring A260/230 and A260/280 ratios using a DS-11 spectrophotometer (DeNovix, USA). We used 500 ng of total RNA isolated from WT and KO cells at PFG stage and day 8 of spontaneous differentiation, treated with $H_2O_2$, to prepare libraries (TruSeq RNA Library Prep Kit v2) according to the manufacturer's protocol. Libraries were quantified with a Qubit fluorometer (TFS), and their quality was assessed with an Agilent High Sensitivity DNA Kit (Agilent Technologies). Libraries were sequenced with an Illumina HiScanSQ sequencer. RNA-seq raw paired-end reads were trimmed with fastp and aligned with the Ensembl GRCh38 human genome with STAR (v2.7), and counts were obtained with feature-Counts v1.6. The raw RNA-seq sequence reads were analyzed via the interactive iDEP9.6 (integrated differential expression and pathway analysis) platform (http://bioinformatics.sdstate.edu/idep96/). Differentially expressed genes (DEGs) between WT and KO, WT + $H_2O_2$ and WT or KO and WT + $H_2O_2$ cells were identified and normalized with the DESeq2 or limma-voom package. DEGs significantly ($p$-value ≤ 0.05) upregulated with a $log_2FC ≥ 0.5$ or downregulated with a $log_2FC ≤ −0.5$ in any of the sequenced samples were selected for further analysis. The analyzed datasets can also be found in Supplementary Data 1–4. The cluster analysis was performed with the Genesis software (http://genome.tugraz.at/genesisclient/genesisclient_description.shtml). Volcano plots were generated with GraphPad Prism 8 software or in Python using Seaborn and Matplotlib packages for genes with normalized reads, with $-log_{10}$ adj $p$-values on the $y$-axis and $log_2$ FC values on the $x$-axis. Genes with $-log10$ adj $p ≥ 1$ and $log_2FC ≥ 0.5$ or $log_2FC ≤ −0.5$ were significantly differentially expressed. The GeneCodis, EnrichR, decoupleR, and Appyters interactive platforms were used for functional enrichment analysis. The cutoff value for significant BP, KEGG, and Wiki pathway results was an adjusted $p$-value ≤ 0.05. Dot plots of representative genes from selected KEGGs, BPs, and Wiki Pathways were plotted in R using the ggplot package. Gene set enrichment analysis (GSEA) was performed with GSEA v4.1.0 software (http://software.broadinstitute.org/gsea/msigdb/index.jsp). GSEA identifies functional enrichment by comparing genes with predefined gene sets. A gene set is a set of genes with similar localizations, pathways, functions, or other features. The input data for the GSEA were normalized counts for the *GPX2* KO vs. WT or for WT + $H_2O_2$ vs. WT gene sets and a reference gene set based on the Molecular Signatures Database set: h.all.v7.4.symbols.gmt (Hallmarks). The permutation number was set to 1000. The enrichment gene sets with a $p$-value ≤ 0.05 and a false discovery rate (FDR) ≤ 0.25 in the GSEA were considered to exhibit statistically significant differences. We used the default parameters of the GSEA software. An enrichment map was used to visualize the GSEA results. The enrichment score (ES) and FDR values were used to sort hallmark pathway-enriched gene sets.

## ATAC-sequencing

The ATAC libraries were generated as described previously[18], following the Omni-ATAC protocol. Shortly, Transposase (Tn5) with assembled Illumina-compatible primers was produced by the Proteomics and Biochemistry Platform of the Andalusian Center for Developmental Biology (CABD, UPO/CSIC/JA) following a published protocol. The WT and KO cells at day 8 of spontaneous differentiation were dispersed to single cells using TrypLE Select (Thermo Fisher Scientific, Netherlands), counted, spun down at $500 \times g$ for 5 min at 4 °C and resuspended at a concentration of $5 \times 104$ cells/50 μL per biological replicate in cold ATAC-RSB (10 mM Tris-HCl (Bio-Shop, Canada) pH = 7.4, 3 mM $MgCl_2$ (Millipore, Germany), 10 mM NaCl (Bio-Shop, Canada), containing 0.1% (v/v) Tween-20 (Roche Diagnostics,

Germany), 0.1% (v/v) NP-40 (Biosciences, USA)). For nuclei extraction, the cells were incubated with 0.1% (v/v) digitonin (Promega, USA) for 3 min at 4 °C and centrifuged for 10 min at 500 x g at 4 °C with 1 mL ATAC-RSB, containing 0.1% (v/v) Tween-20. Nuclei were prepared for the transposition reaction following the previously described protocol[110]. The transposed fragments were purified using the Zymo DNA Clean & Concentrator (Zymo Research, USA), eluted in 20 μL of $H_2O$, and stored at −20 °C. Barcoded adapters were added to the transposed fragments via PCR using the NEBNext Ultra II Q5 2x Master Mix (New England Biolabs, USA), following the manufacturer's protocol. All samples were amplified for 10 cycles, then purified using a double-sided AMPure XP bead cleanup (Beckman Coulter, USA) and eluted in 20 μL of $H_2O$. DNA concentration was measured with the Qubit dsDNA HS Assay Kit (Thermo Fisher Scientific, Netherlands), and sample quality was assessed using the TapeStation HS D1000 Kit (Agilent Technologies, USA). Sequencing was performed on the NovaSeqX platform with 150 bp paired-end reads. The nf-core/atacseq bioinformatics pipeline (version 2.1.2, https://nf-core/atacseq) was used for ATAC-seq data analysis. After adapter trimming with Trim Galore and quality assessment with FastQC, reads were mapped to the human genome (hg38) using Bowtie2 with default paired-end settings. Non-nuclear reads and unmapped paired reads were discarded, and duplicate reads were removed using Picard MarkDuplicates. Further filtering to exclude unmapped regions, mitochondrial DNA, and blacklisted regions was performed with SAMtools. Broad peak calling was conducted using MACS2, while peak annotation (annotatePeaks.pl) and motif enrichment analysis (findMotifsGenome.pl) were performed with HOMER. Normalized bigWig files, scaled to 1 million mapped reads, were generated using BEDTools. Differential accessibility analysis was conducted with DESeq2, and the IGV platform was used to visualize bigWig tracks, peaks, and differential sites. The analyzed dataset can also be found in Supplementary Data 5.

## Live-cell imaging and analysis

The IncuCyte live-cell imaging system (Sartorius, Germany) was used to monitor living cells. hPSCs were seeded at a density of $1.5 \times 10^4$ cells/cm² on 24-well plates coated with Geltrex and cultured in the IncuCyte for up to 6 days at 37 °C in a 5% $CO_2$ atmosphere. Photomicrographs were captured every 2 h, and real-time confluence measurements were performed using the IncuCyte Base Analysis Software (Sartorius, Germany). Cell growth was assessed by analyzing changes in the percentage of the image area occupied by cells (confluence) over time.

## Statistics and reproducibility

All graphs were generated using GraphPad Prism v8.4.2 for Windows (GraphPad Software, Boston, Massachusetts, USA). Statistical analyses were conducted using unpaired two-tailed Student's $t$-tests or one-way ANOVA for multiple comparisons. Data are presented as means ± 95% confidence intervals (CIs) unless otherwise stated. No data was excluded from the analyses. Quantifications were performed independently by two investigators. At least 3 independent biological replicates were performed for each experiment.

## Reporting summary

Further information on research design is available in the Nature Portfolio Reporting Summary linked to this article.

# Data availability

All data generated or analyzed during this study are available. Processed data and all datasets necessary to interpret the results are included in the published article and its Supplementary Information. Raw RNA-seq, scRNA-seq, and ATAC-Seq data generated during the study have been deposited in the NCBI GEO database under accession number GSE291135. Source data are provided as a Source Data file. Raw

image files are available from the corresponding author upon a reasonable request. The authors declare that cell lines are available for the research community upon request from the corresponding author. Other published datasets we used in this study could be obtained from: E-MTAB-6967[19], OMIX001616[36], GSE197064[107], GSM2689399[20], E-MTAB-8210 and E-MTAB-7189[9]. Source data are provided with this paper.

## Code availability

R code generated for RNA-seq and scRNA-seq data analysis are available at GitHub (https://github.com/WJSzlachcic/BorowiakLab_2025_GPX2/).

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

## Acknowledgements

We would like to express our gratitude to Dr. Artur Jankowski for his excellent technical support. We also thank Dr. Ludovic Vallier and all members of the Borowiak lab for valuable discussions, comments, and support. We also thank Dr. Allan Bradley and Dr. Knut Woltjen for sharing plasmids via Addegene. This work was supported by the Polish National Science Center grant OPUS (2020/37/B/NZ3/01917 and 2020/39/B/NZ3/01408 to M. B.), Foundation for Polish Science and EU TEAM Programme (POIR.04.04.00-00-20C5/16-00) to M. B., Polish National Science Center, Miniatura grant (2022/06/X/NZ3/00465) to J. Sz., Polish National Science Center, Sonata grant (2021/43/D/NZ3/02294) to W. J. Sz., and Sonata grant (2022/47/D/NZ3/02068) to K. B.

## Author contributions

J.S.—experimental design and execution, including *GPX2* KO hPSC generation, directed and spontaneous differentiation, scRNA-seq analysis of SC-β cells WT and KO cells, data acquisition and analysis, figure preparation, and manuscript writing; W. J.S.—scRNA-seq analysis of PFG WT and KO cells, figure preparation, and manuscript writing; K.B.—bulk RNA-sequencing and initial analysis, ATAC-seq initial analysis; M.S.—performed and analyzed ATAC-seq; M.Baginska.—participated in experimental execution; M.Borowiak.—experimental design, data analysis, manuscript writing, and funding acquisition.

## Competing interests

The authors declare no competing interests.
