## [Transparent Peer Review file · Nature Communications]

Oxidative stress and GPX2 control pancreatic vs. non-pancreatic cell fate in human endoderm

Corresponding Author: Professor Malgorzata Borowiak

Version 0:

Reviewer comments:

Reviewer #1

(Remarks to the Author)

Review of the NCOMSS-25-32853-T entitled "GPX2 and oxidative stress as gatekeepers of pancreatic cell fate" by J. Szpotkowska et al. The overall goal of this is to assess the role that oxidative stress has on human pancreatic differentiation in the context of a human pluripotent stem cell (hPSC) stepwise differentiation protocol. Two mediators of the oxidative stress pathway, Gpx2 and SOD, are upregulated during human fetal pancreas development. GPX2 protein is expressed from the posterior foregut (PFG) stage through the late EP stages of their SC-B differentiation protocol, suggesting a potential role for this gene in pancreatic differentiation. Intriguingly, a comparison of Gpx2 KO hPSC to WT hPSC that have each been treated with their SC-B protocol, demonstrate that in GPX2 KO cells, differentiation is altered from WT cells beginning at the PFG stage. This alteration includes upregulated expression of the intestinal marker Cdx2 and the liver marker HNF4alpha, while Hhex and Pdx1 are significantly downregulated. Remarkably, after 21 days of stepwise differentiation toward a pancreatic beta cell fate, Gpx2 loss of function cells are capable of contributing to the pancreatic beta cell population as efficiently as control cells. Single cell transcriptional analysis of WT and Gpx^{-/-} 21 Day cultures reveals that while the differentiated beta cell populations display a similar transcriptional profile and found at a similar frequency, the less differentiated progenitor states found in WT cultures are dramatically reduced and are replaced by several populations of novel "multilineage" progenitors. The authors also demonstrate that, Gpx^{-/-} hPSC derived DE, cultured in a basal environment and allowed to differentiate on their own for 8D under excessive oxidative stress conditions (WT + peroxide or Gpx2^{-/-}) upregulates pro-liver and pro-intestinal markers. The paper suggests that oxidative stress is required for appropriate hPSC differentiation and that Gpx2 is in part responsible for mitigating the oxidative stress response required during hPSC derived posterior foregut differentiation.

My major critiques of the paper are as

- 1) This paper difficult to follow, the hypothesis tested difficult to pin down and the significance of the conclusions drawn dampened by existing literature (some of which is outlined in 2) or by weakly supportive or poorly justified experiments.
- 2) While the role of oxidative stress on pancreatic differentiation has not been previously identified, the idea that the strict control of oxidative stress is critical for fate acquisition in a developmental context (Timme-Laragy et al, 2017; PMID: 2897759) or in disease been explored in depth in a variety of contexts. Strict control of oxidative stress is known to be altered in various gut diseases and in a variety of different developmental context, including intestine development.
- 3) In particular it is difficult to determine the relevance of making any conclusions on stages past the PFG stage because their biological relevance of the conclusions from results that are produced after the primary defect at the PFG stage has occurred, are difficult to determine and question the biological/experimental relevance.
- 4) Finally, I am particularly flummoxed by the concluding experiments (summarized in Figures 6, 7). While it may be true that ECM and BMP signaling are upregulated in Gpx2 deficient DE differentiation, no direct evidence links these two alterations to one another (as outlined in 7M).

What follows is non-exhaustive list of more minor that focus more on the paper up to the end of Figure 4. I have more numerous concerns with the conclusions made and the data produced that was used to produce the last 3 figures but these concerns are only briefly touched on in this review.

- 1) How many independent cell lines were used throughout this work. Was only a single WT and mutant hPSC line used?

- 2) Figure 1 establishes a potential unique role of Gpx2 in development of the human pancreas in vivo and in vitro by defining the expression window of the transcript or the protein. However inconsistencies in these data are not explored for example: GPX2 protein is abundantly detected in week 10 and 13 fetal pancreas while abundant transcripts are detected in published data sets at week 6 of fetal pancreas, but is severely downregulated after week 9.
- 3) A concern is the difference between the two WT biological replicates used in the bulk RNA seq experiments in Fig 2 and how these inconsistencies exacerbate/complicate the trajectory of Gpx2 deficient cells under similar conditions.
- 4) In general the methods used to assess oxidative stress nor their significance/caveats are highlighted in this proposal (Fig 1B, 2B, 3K).
- 5) Figure 2 demonstrates that Gpx2 loss effects the initial DE PFG stage upregulating HNF4, CDX2 and Pdx1 while downregulating Hhex compared with WT hES. It is curious that Pdx expression is upregulated in these cultures and this expression is termed precocious because later in differentiation, these cultures appear to have less Pdx expression than controls.
- 6) The use of "altered spatial patterning" in PFG is confusing and inaccurate. If individual spheroids demonstrate a reproducible "pattern" in WT cultures this might be correct but the images shown demonstrate that individual spheroids exhibit wide-spread cell to cell heterogeneity thus making conclusions from scRNA seq about patterning is not accurate. Temporal patterns might be a more accurate term but "spatial patterning" is certainly not correct, given the data presented.
- 7) A concern for the PEN (pancreatic endoderm) experiments outlined in Figure 3 includes the difference between the two WT replicates in the PC analysis at the PFG stage.
- 8) Figure 3K is inadequately described. The plots depicting cell program use is hard to read.
- 9) Is Bmp6 normally upregulated by the endoderm during intestine or liver development? BMPs are typically added to the media to induce PFG to induce hepatic fate. This suggests that the bias in these cultures is due to aberrant gene expression caused by oxidative stress conditions rather than a normal developmental process.
- 10) Fig 3L how was the period of oxidative stress induction chosen?
- 11) Fig 3M- it is unclear what groups the p values provided are comparing. Important controls that are missing include the addition of 0.01 uM peroxide to KO cultures and the addition of 100pM selenium to WT cultures.
- 12) While the images shown in figure 3N highlight the points outlined in the paper in the absence of quantitation, they lack significance.
- 13) Why does the DAPI staining of the spheroids in Fig 3N, representing PEN, look so different from the cultures in 4A, representing SC-B spheroids? 3N looks tubular and 4A looks circular? They all seem to be taken at a similar magnification.
- 14) There is an apparent hepatic bias to the Gpx2 KO (versus WT) hP cells during pancreas-induction. In addition, a subset of intestinal and bile duct gene programs are also upregulated. Are these each represented in one of the 3 distinct GPX KO ML populations present after pancreatic differentiation in Figure 4? Or are these ML populations entirely multilineage?
- 15) There is no direct evidence to support their other conclusion that ECM alterations drive BMP expression that in turn promotes hepatic fates.
- 16) Oxidative stress is a well-known inducer of fibrosis (excess ECM)- could the changes noted in the ECM be the result of a fibrotic response?
- 17) The authors demonstrate that in the absence of Gpx2 hPSC will turn on a number of hepatic genes. This cell population is termed "hepatic progenitors". How close are hPSC induced hepatic progenitors or in vivo examined hepatic progenitors to the hepatic progenitors produced by Gpx2 deficiency herein?
- 18) The conclusion that "ECM remodeling emerges as a key driver of lineage plasticity" is not well-supported by the data presented and appears mainly correlative.
- 19) The connection between ECM production, TGFb superfamily signaling and hepatic fate has not been established.

Reviewer #2

(Remarks to the Author)

In this study, Szpotkowska et al. investigated the role of GPX2 in human pancreatic differentiation and revealed oxidative stress as a key determinant of pancreatic versus non-pancreatic cell fate. The authors demonstrated that GPX2-deficient cells performed increasing hepatic differentiation within a pancreatic differentiation system, highlighting the critical role of GPX2 in guiding lineage commitment during progenitor differentiation. Mechanistically, they showed that loss of GPX2 induces extracellular matrix (ECM) remodeling, which in turn activates BMP signaling and diverts differentiation away from the pancreatic lineage. Furthermore, manipulating ROS levels, either by induction or alleviation, was sufficient to mimic or rescue the phenotypes associated with GPX2 deficiency. Overall, the data are comprehensive, clearly presented, and logically organized, offering novel insights into the regulation of endodermal differentiation through redox signaling. In addition, the single-cell dataset provide rich information for the pancreatic differentiation field for future exploration. However, several key issues need to be addressed. Here are the detailed comments:

Major comments:

1. In Figure 1I, the Western blot control is inconsistent. GAPDH expression in SC- β cells is much lower than in other samples, which may compromise the conclusion that SC- β cells do not express GPX2. More uniform loading or replicate blots would strengthen this claim.
2. In Figures 2F and 2G, although some HNF4A⁺/FGB⁺ cells are detected, the number of positive cells appears very low. Quantitative comparison using FACS is recommended to more accurately assess the differences between WT and KO cells.
3. In Figure 3N, it is noted that WT PEN cells show no PDX1 expression, especially since PDX1 was clearly detected in Figure 2E. This inconsistency raises concerns about the stability and efficiency of the differentiation system, and warrants further explanation or validation.
4. In Figures 4A and 4B, the authors report an increase in AFP⁺ and FGB⁺ liver progenitors in KO cells. However, the immunostaining shows very few AFP⁺ cells. Quantitative methods such as FACS or image-based quantification should be used to confirm these differences between WT and KO.
5. From Figures 4A-B, GPX2 KO does not appear to affect the proportion of pancreatic β -like cells. However, the functional capacity of SC- β cells remains unclear. Does GPX2 loss influence β cell function, such as insulin content, mitochondrial morphology and number, or glucose-stimulated insulin secretion (GSIS)? Functional validation is recommended.

Minor comments:

1. In Figure 1A, it is suggested to use double marker combinations to define cell stages more clearly—for example, DE (SOX17/FOXA2), PP (PDX1/NKX6.1), EP (CHGA/NKX2.2). Alternatively, FACS can be used to show stage-specific differentiation efficiency.
2. Cite PMID: PMC9293889, which also discusses pancreatic versus hepatic fate decisions at the progenitor stage.
3. In Figure 6G and Figure 7E, some of the displayed loci show minimal or unclear differences in chromatin accessibility. If possible, the authors should highlight more pronounced differential peaks or adjust the visualization (e.g., by changing scale, zooming in, or using a different normalization method) to better emphasize significant changes. Alternatively, a brief explanation for why these loci were selected (e.g., biological relevance despite modest changes) would be helpful.
4. In the figure legends, specify whether bar plots represent mean \pm SD or mean \pm SEM to improve clarity.

Reviewer #3

(Remarks to the Author)

Reviewer #4

(Remarks to the Author)

This is a technically rigorous study that identifies GPX2, a redox-sensitive glutathione peroxidase, as a key regulator of human endodermal lineage decisions during pancreatic differentiation from human pluripotent stem cells (hPSCs). The authors demonstrate that GPX2 deficiency skews differentiation toward hepatic and intestinal fates, even under pro-pancreatic conditions, by altering oxidative stress levels and disrupting the cellular microenvironment, particularly ECM composition and BMP signaling. The work is conceptually novel and contributes important mechanistic insight into how redox balance regulates human cell fate determination, with potential implications for regenerative medicine and disease modeling.

Major concerns:

1. The manuscript could more clearly discuss whether the liver- and intestine-like cells represent mature or immature progenitor states. This is particularly relevant given the absence of ALB transcripts and the enrichment of multilineage progenitor markers.
2. Figures summarizing scRNA-seq cluster proportions, pseudotime trajectories, and pathway enrichment could benefit from additional quantitative clarity. Including statistical comparisons between WT and KO proportions across clusters would enhance interpretability.

Minor concerns:

1. Clarify in the abstract that the knockout of GPX2 leads to a bias in lineage commitment rather than a complete block in pancreatic differentiation.
2. Improve consistency in gene nomenclature and italicization throughout the manuscript.
3. Confirm whether the observed increase in α cells (GCG⁺) in KO samples is statistically significant and discuss its implications.

Version 1:

Reviewer comments:

Reviewer #1

(Remarks to the Author)

The authors have addressed many of the concerns listed in the previous review. Several concerns remain and should be addressed.

1) I still have reservations about the cell lines used. The authors originally omitted any information on which cell lines they used. The authors now state that the cell lines used for the KO experiments were produced from the HUES8 parental line (the methods state that the Cas9-HuES8 line was used to generate the knockout) and the over expression lines produced in the H1 and HUES8 background. Were the experiments with the KO cells performed with a single clone or were they mainly from 1 of the 2 KO clones? Which cell line was used as the control for the KO experiments? There are methods describing the creation and use of the DOX inducible HUES8 cell line but no mention of how the H1 over-expressing cell line was created. This information should be provided in the methods or in the body of the paper. Because my original concerns were not addressed; please provide details on the exact cell lines (and KO clone) used to generate the data in Figs 2-5.

2) The title of the paper seems misleading. "Gpx2 and oxidative stress as gatekeepers of pancreatic fate" specifically the term gatekeeper is ambiguous/non-scientific and seems to suggest that oxidative stress levels prevent pancreatic differentiation.

3) The new text states that 0.1 10 uM H202 is used to treat WT cells however the graph in the figure 3 (old 3M) that WT cells treated with 0.01 uM demonstrate an elevated level of Pdx1.

Reviewer #2

(Remarks to the Author)

The authors have added more data and provided further evidences. Overall, the revised manuscript has been improved. However, there are still some questions which need to be addressed:

1. For Western blotting experiments, all samples for comparison should be performed and presented in a single gel rather than in a separated manner.
2. The FACS results should be quantified and statistically evaluated.
3. There is no GSIS result in the rebuttal Fig. 12. Additionally, the results of functional evaluation of β cells are important and should be added to the manuscript.
4. GPX2 KO notably affected the cell-fate choices during pancreatic differentiation, but at the SC- β cell stage (day 21) the authors did not observe significant changes in β cell differentiation efficiency and function between WT and GPX2 KO cells. These observations are confusing, and need more explanation and discussion.

Reviewer #3

(Remarks to the Author)

Reviewer #4

(Remarks to the Author)

Most of this reviewer's comments have been addressed.

Version 2:

Reviewer comments:

Reviewer #1

(Remarks to the Author)

My concerns with the manuscript have been resolved with the revisions.

Reviewer #2

(Remarks to the Author)

The authors have addressed our comments.

Reviewer #3

(Remarks to the Author)

We thank the Reviewers for their time and valuable input on our manuscript. We are delighted that Reviewers 2 and 4 find our work exciting and suitable for publication in Nature Communications. We appreciate all suggestions and hope Reviewer 1 will be satisfied with our additions and revisions. We have addressed all your comments; please find the schedule of our responses below, highlighted in blue.

Reviewer #1 (Remarks to the Author):

Review of the NCOMSS-25-32853-T entitled “GPX2 and oxidative stress as gatekeepers of pancreatic cell fate” by J. Szpotkowska et al. The overall goal of this is to assess the role that oxidative stress has on human pancreatic differentiation in the context of a human pluripotent stem cell (hPSC) stepwise differentiation protocol. Two mediators of the oxidative stress pathway, Gpx2 and SOD, are upregulated during human fetal pancreas development. GPX2 protein is expressed from the posterior foregut (PFG) stage through the late EP stages of their SC-B differentiation protocol, suggesting a potential role for this gene in pancreatic differentiation. Intriguingly, a comparison of Gpx2 KO hPSC to WT hPSC that have each been treated with their SC-B protocol, demonstrate that in GPX2 KO cells, differentiation is altered from WT cells beginning at the PFG stage. This alteration includes upregulated expression of the intestinal marker Cdx2 and the liver marker HNF4alpha, while Hhex and Pdx1 are significantly downregulated. Remarkably, after 21 days of stepwise differentiation toward a pancreatic beta cell fate, Gpx2 loss of function cells are capable of contributing to the pancreatic beta cell population as efficiently as control cells. Single cell transcriptional analysis of WT and Gpx^{-/-} 21 Day cultures reveals that while the differentiated beta cell populations display a similar transcriptional profile and found at a similar frequency, the less differentiated progenitor states found in WT cultures are dramatically reduced and are replaced by several populations of novel “multilineage” progenitors. The authors also demonstrate that, Gpx^{-/-} hPSC derived DE, cultured in a basal environment and allowed to differentiate on their own for 8D under excessive oxidative stress conditions (WT + peroxide or Gpx2^{-/-}) upregulates pro-liver and pro-intestinal markers. The paper suggests that oxidative stress is required for appropriate hPSC differentiation and that Gpx2 is in part responsible for mitigating the oxidative stress response required during hPSC derived posterior foregut differentiation.

My major critiques of the paper are as

1) This paper difficult to follow, the hypothesis tested difficult to pin down and the significance of the conclusions drawn dampened by existing literature (some of which is outlined in 2) or by weakly supportive or poorly justified experiments.

We thank the Reviewer for their critical assessment of our manuscript, particularly regarding its clarity, the clarity of the hypothesis, and the perceived significance of our conclusions. We acknowledge that these aspects are crucial for effective scientific communication.

Our central hypothesis, which we aim to convey more clearly in the revised MS, is that GPX2 plays a specific and critical role in regulating oxidative stress during the early stages of human pancreatic fate determination from hPSCs, thereby influencing lineage commitment. While the broader role of oxidative stress in development and disease is established, our work provides novel insights by identifying (i) GPX2 as a key molecular regulator and (ii) within this precise

developmental context. This distinction is vital as it moves beyond general observations to pinpoint a specific molecular mechanism. Furthermore, the role of oxidative stress in the pancreatic development and in human pancreatic *in vitro* differentiation is understudied.

Regarding the experimental support, a multi-faceted experimental approach robustly underpins our conclusions. As detailed in our response to other comments, our findings from bulk RNA-seq are corroborated by comprehensive single-cell RNA sequencing (scRNA-seq) performed at two different time points utilizing both *GPX2* KO and overexpression (OE) hPSC clonal lines, alongside ATAC-seq, and immunofluorescence (IF) staining, as well as both directed and spontaneous differentiation experiments of WT and *GPX2* KO endoderm. These collective data sets consistently demonstrate the altered differentiation trajectory of *GPX2* KO endoderm, specifically its bias towards hepatic lineages, thereby providing strong and consistent evidence for our conclusions. Furthermore, our study provides valuable insights into the molecular mechanism underlying the role of GPX2, although we acknowledge that these experiments do not exhaust all possibilities for further exploration. In revised MS and in rebuttal letter we have added the new experimental data on link ECM and BMP signaling as well as AFP and FGB expression. Together, these diverse experimental validations strengthen the significance of our findings beyond what might be inferred from any single experimental modality or existing broader literature. We are pleased that other Reviewers have provided generally positive feedback on the manuscript. We have made considerable efforts to revise the manuscript based on all feedback, and we hope that the main message is now more clearly presented, additional data strengthened the main conclusion and its overall significance.

2) While the role of oxidative stress on pancreatic differentiation has not been previously identified, the idea that the strict control of oxidative stress is critical for fate acquisition in a developmental context (Timme-Laragy et al, 2017; PMID: 2897759) or in disease been explored in depth in a variety of contexts. Strict control of oxidative stress is known to be altered in various gut diseases and in a variety of different developmental context, including intestine development.

We appreciate the Reviewer's comment and agree that oxidative stress plays a crucial role in developmental processes and various diseases, as established in many studies. Our manuscript acknowledges this broader context in the introduction. Our research specifically investigates the role of oxidative stress during early pancreatic fate determination in human pluripotent stem cell (hPSC) differentiation, an area that is, to our knowledge, understudied. We found that GPX2 is a key regulator of oxidative stress during this critical phase of pancreatic development, establishing a novel mechanistic link and highlighting the need for tight control over reactive oxygen species (ROS) for successful differentiation. While the importance of managing oxidative stress is well recognized, our study provides unique insights into its regulation within the early human pancreatic lineage. By identifying GPX2 as a direct molecular player, we offer a concrete target for understanding and potentially manipulating this process. This research has significant translational relevance, as the dysregulation of oxidative stress is linked to numerous pancreatic diseases, including diabetes and pancreatitis. Our findings shed light on the early developmental vulnerabilities that may contribute to these conditions, making a

valuable contribution to understanding pancreatic development and its susceptibility to oxidative stress-related issues.

3) In particular it is difficult to determine the relevance of making any conclusions on stages past the PFG stage because their biological relevance of the conclusions from results that are produced after the primary defect at the PFG stage has occurred, are difficult to determine and question the biological/experimental relevance.

We appreciate the Reviewer insightful comment regarding the relevance of our conclusions on stages past the PFG stage. We agree that the initial PFG defect is a critical finding, and our decision to extend the analysis was specifically aimed at addressing a key biological question: Can the developing pancreas compensate for or recover from an early, oxidative stress-induced insult? Do the ML progenitors persist or represent temporary alternations in PFG endoderm development.

The Reviewer correctly highlights the challenge of interpreting results after the primary defect has occurred. Our investigation into these later stages was not a simple continuation but a deliberate effort to understand the system capacity for adaptation and compensatory differentiation. The formation of functional β cells, despite the early PFG defect and the shift toward hepatic and ML cell types, is a crucial finding. It provides direct evidence that the system possesses a remarkable ability to overcome or mitigate an early developmental defect partially. This is highly relevant as it models a fundamental biological process: how an organism responds to and recovers from early-life stressors.

Our rationale is also supported by the fact that GPX2 is expressed beyond the PFG stage, and we demonstrated that oxidative stress is a factor throughout the entire pancreatic differentiation process. Therefore, understanding the long-term consequences of GPX2 deletion - and whether the initial PFG changes are exacerbated or lead to new pathologies - is crucial for a comprehensive understanding of its function. Therefore, the later stages aren't simply a post-defect analysis; they represent a crucial part of the GPX2 expression timeline and the ongoing putative role of oxidative stress.

4) Finally, I am particularly flummoxed by the concluding experiments (summarized in Figures 6, 7). While it may be true that ECM and BMP signaling are upregulated in Gpx2 deficient DE differentiation, no direct evidence links these two alterations to one another (as outlined in 7M).

Thank you for highlighting this critical point. We wanted to highlight the newly described role of GPX2 in our MS rather than focusing on follow-up questions, such as the ECM and BMP link, given that this link was also initially reported in the literature. Having said this, in response to your comment, we demonstrate that ECM remodeling leads to BMP activation and subsequent hepatic cell fate commitment by treating WT DE cells during spontaneous differentiation with matrix metalloproteinase 2 (MMP2). MMP2 remodels key ECM components, type IV and V collagen, laminin, fibronectin and elastin¹. We initiated the MMP2 treatment on day 5 of differentiation, at the same time point at which H₂O₂-induced oxidative stress was induced in our experiments. As markers of BMP signaling activation and hepatic

commitment, we analyzed protein levels of pSMAD1/5 and FGB, respectively, at 72 hours post-treatment. IF and quantification showed that MMP2 treatment altered the structure of COL4A and increased pSMAD1/5 and FGB protein levels by 1.3- and 11.5-fold, respectively (Rebuttal Fig. 1). Additionally, we conducted spontaneous 2D differentiation of WT DE cells seeded on various coatings: Geltrex (our standard coating), laminin, and vitronectin. Differentiation on different coatings affected COL4A structure and pSMAD1/5 levels, with the most pronounced increase observed in cells grown on laminin-coated surfaces. Interestingly, our scRNA-seq data revealed increased expression of the *LAMAI* gene in ML, MP1, and MP2 clusters in *GPX2* KO. The FGB expression levels also varied depending on the coating, with the lowest protein levels observed on vitronectin and the highest on laminin (Rebuttal Fig. 2). Vitronectin on polymers forms multimeric globules that prevent the exposure of its functional domains to cells², potentially reducing the accessibility of its binding sites. Geltrex, on the other hand, is a mixture containing, among others, laminins and collagens - proteins known to induce BMP signaling effectively. Together, we interpret these results as indicating that ECM composition and structure are correlated with BMP signaling activity and FGB expression in hPSC-derived DE *in vitro*. Therefore, by inducing changes in the ECM, one can modulate further differentiation of DE.

The literature further supports our conclusion that the ECM can regulate BMP activity. It is now widely recognized that the ECM is not merely a passive reservoir but actively regulates the biological activity of BMPs and other signaling pathways. Fibrillar ECM proteins can directly bind BMPs and/or their signaling molecules, thereby controlling the dose and availability of BMPs³. Examples include collagen IV and fibronectin. Collagen IV has been shown to bind BMP4 at its C-terminal region⁴, and BMP2 is known to bind fibronectin. Moreover, cell-sized micropatterns of BMP2 trapped in fibronectin matrices induce distinct reorganization of actin and adhesion structures, leading to phosphorylation and nuclear translocation of SMAD1/5/8⁵. The relationship between BMPs and liver cell fate is also well established. Utilizing a mouse *Bmp4* null mutation and a BMP inhibitor (*Xenopus* noggin protein, Xnoggin), it was demonstrated that BMP signaling is essential for inducing liver-specific gene expression in the endoderm and suppressing a pancreatic developmental program. BMP signaling also plays a crucial role in driving the morphogenetic expansion of the hepatic endoderm into the liver bud⁶. Inhibition of BMP signaling results in a reduced number of induced hepatoblasts⁷ and disrupts the typical structure of hepatic sinusoids, leading to liver congestion and impaired fibronectin deposition⁸. hPSC differentiation toward hepatocytes proceeds through a series of well-defined stages: hPSC → definitive endoderm (DE) → posterior foregut (PFG) → liver bud (LB) → hepatic progenitors (HP) → hepatocyte-like cells. A critical decision point that directs differentiation toward either pancreatic progenitors or liver bud occurs during the posterior foregut stage. In numerous protocols, at this stage, BMP4 protein is added to the differentiation medium (along with other factors) to guide cells toward hepatic progenitors⁹⁻¹³. This is the same time point at which we manipulate oxidative stress and BMP levels in our differentiation. Lastly, multiple hPSC pancreatic differentiation protocols incorporate BMP pathway inhibitors, such as Noggin, dorsomorphin, or LDN193189, to preferentially direct cells toward pancreas-like cells rather than liver-like

cells^{14,15}. The role of BMP signaling in liver development has been extensively reviewed by others^{6,7,16,17}.

To summarize, we hope that these new experimental data and a simplified overview of the literature support the inclusion of the ECM-BMP link in the summary scheme.

Rebuttal Fig. 1. Connection between ECM, BMP pathway, and liver fate. **A.** WT untreated and MMP2 (140 ng/mL)-treated cells differentiated spontaneously on Geltrex until day 8, were stained for FGB (green), pSMAD1/5 (red) and COL4A (white). DAPI marks nuclei (blue). Scale bar = 100 μ m. Zoom inserts are also provided. **B.** Quantification of the mean fluorescence intensity of FGB and pSMAD1/5 relative to DAPI shown as dot plots. Each dot represents one image, with mean and 95% confidence intervals (CIs) indicated. An unpaired t-test was used to determine statistical significance. $N = 3$ independent biological repeats.

Rebuttal Fig. 2. Connection between ECM changes, BMP pathway, and liver fate. **A.** WT cells were seeded on different coatings, such as Geltrex, laminin and vitronectin and differentiated spontaneously until day 8, stained for FGB (green), pSMAD1/5 (red), and COL4A (white). DAPI marks nuclei (blue). Scale bar = 100 μ m. **B.** Quantification of the mean fluorescence value of pSMAD1/5 relative to DAPI for WT cells on Geltrex (orange), laminin (green), and vitronectin (purple), shown as a dot plot. Each dot represents one image, with mean and 95% confidence intervals (CIs) indicated. An unpaired t-test was used to determine statistical significance. **C.** UMAPs show the expression level of ECM marker genes *LAMA1*, *VTN*, and *COL4A1* in WT and KO cells at SC- β cell stage, based on scRNA-seq results from day 21 of differentiation. Cells with high gene expression are marked in blue. $N = 3$ biological repeats.

What follows is non-exhaustive list of more minor that focus more on the paper up to the end of Figure 4. I have more numerous concerns with the conclusions made and

the data produced that was used to produce the last 3 figures but these concerns are only briefly touched on in this review.

1) How many independent cell lines were used throughout this work. Was only a single WT and mutant hPSC line used?

Regarding the cell lines utilized in our study, we employed a set of human pluripotent stem cell lines to ensure the generalizability and reproducibility of our findings. Specifically, our study utilized HUES8 (derived by Dr. Melton at Harvard University), H1 (derived by Dr. Thomson at the University of Wisconsin), and iCas9-HUES8¹⁸ (derived by Dr. Huangfu, Memorial Sloan Kettering Cancer Center) as WT hESCs. For our experimental manipulations, we generated two independent *GPX2* KO clonal lines based on the HUES8 parental background and two independent *GPX2* overexpression clonal lines derived from the H1 and HUES8 parental hESCs. We have added a more detailed description to the method parts.

In the revised manuscript, the “Material and Methods” section now reads: “*hPSCs used in this study included human H1 (derived by Dr. James Thomson, University of Wisconsin), HUES8 (derived by Dr. Doug Melton at Harvard University) and iCas9-HUES8 (a doxycycline-inducible Cas9-expressing HUES8 line generated by Dr. Danwei Huangfu, Memorial Sloan Kettering Cancer Center) cell lines.*”

2) Figure 1 establishes a potential unique role of Gpx2 in development of the human pancreas in vivo and in vitro by defining the expression window of the transcript or the protein. However inconsistencies in these data are not explored for example: GPX2 protein is abundantly detected in week 10 and 13 fetal pancreas while abundant transcripts are detected in published data sets at week 6 of fetal pancreas, but is severely downregulated after week 9.

We thank the Reviewer for the comments regarding the potential inconsistencies between *GPX2* mRNA and protein expression data and the limitations of scRNA-seq depth. The reviewer correctly highlights a potential discrepancy where GPX2 protein is abundantly detected at weeks 10 and 13 of fetal pancreas development, while published transcriptomic data show abundant transcripts at week 6, with severe downregulation after week 9. We agree that such differences can arise due to several biological factors:

1. **Post-transcriptional and translational regulation:** mRNA levels do not always directly correlate with protein levels. Gene expression is a highly regulated process, and significant control occurs at the post-transcriptional and translational levels. Factors such as mRNA stability, translational efficiency, microRNA regulation, and protein degradation rates can lead to temporal lags or quantitative differences between transcript and protein abundance. It is possible that *GPX2* mRNA, transcribed at earlier stages (e.g., week 6), is stable and translated into protein that persists and accumulates in the cells, leading to higher protein levels at later time points (weeks 10-13) even if mRNA levels have declined.

2. **Protein half-life:** Proteins have varying half-lives. If GPX2 protein has a relatively long half-life, it could accumulate and remain detectable for a period even after its corresponding mRNA has been downregulated. Conversely, if mRNA has a short half-life, its transient presence might not immediately translate into high protein levels if protein synthesis is slow, or protein degradation is rapid.
3. **Single-cell RNA sequencing resolution/depth:** while powerful for resolving cellular heterogeneity, scRNA-seq often suffers from "dropout" events, where transcripts are not detected due to technical limitations (e.g., low capture efficiency, insufficient sequencing depth). Thus, low-abundance transcripts or transcripts in cells with low RNA content might be missed, leading to an apparent downregulation or absence of expression even if they are present at low levels. This could potentially explain why *GPX2* transcripts appear severely downregulated after week 9 in published datasets, while the protein remains abundant. The scRNA-seq data might not fully capture the complete transcriptional landscape, especially for genes expressed at moderate to low levels.
4. **Cellular context and heterogeneity:** The published scRNA-seq data may represent a snapshot of gene expression across a heterogeneous cell population, while protein detection might be more localized or concentrated in specific cell types that are particularly relevant at later developmental stages. Differences in cell composition or specific cellular states between the time points of mRNA and protein analysis could also contribute to observed variations.

We acknowledge these considerations and believe that our protein expression data, combined with the published transcriptomic data, collectively suggest a dynamic and potentially complex regulatory mechanism for GPX2 during human pancreatic development.

3) A concern is the difference between the two WT biological replicates used in the bulk RNA seq experiments in Fig 2 and how these inconsistencies exacerbate/complicate the trajectory of Gpx2 deficient cells under similar conditions.

We appreciate the Reviewer's concern regarding the differences observed between the two WT biological replicates in the bulk RNA-seq experiments presented in Figure 3B. Furthermore, we acknowledge that the early developmental stages of hPSC differentiation are highly dynamic, and even slight variations in growth dynamics or initial cell density can have a subtle impact on gene expression profiles. As the biological replicates were from independent differentiation experiments, not just two different wells performed in parallel, some differences can be observed. However, we wish to emphasize that our conclusions regarding the trajectory of GPX2-deficient cells are further supported by a comprehensive suite of additional experiments, including scRNA-seq, ATAC-seq, and IF staining, as well as directed and spontaneous differentiation of WT and KO endoderm. Collectively, these independent experimental approaches consistently demonstrate the altered differentiation of *GPX2*

knockout endoderm and its bias towards hepatic lineages, thereby robustly validating our findings despite the inherent variability of early *in vitro* differentiation systems.

4) In general the methods used to assess oxidative stress nor their significance/caveats are highlighted in this proposal (Fig 1B, 2B, 3K).

We thank the Reviewer for pointing out the lack of detail in our description of the oxidative stress-related experiments. In Figures 1B and 2B, we assessed intracellular oxidative stress using 2',7'-dichlorodihydrofluorescein diacetate (H₂DCFDA), a widely used fluorescent probe in ROS research^{19–22}. H₂DCFDA is a membrane-permeable compound that enters cells and is hydrolyzed by intracellular esterases to form non-fluorescent DCFH, which becomes trapped in the cytoplasm. DCFH is subsequently oxidized by ROS to produce the fluorescent molecule 2',7'-dichlorofluorescein (DCF). The fluorescence intensity, which reflects intracellular peroxide levels, was quantified by flow cytometry.

We acknowledge the limitations associated with H₂DCFDA, including the time-dependent decay of DCF fluorescence, which may limit its utility in dynamic or long-term measurements. While H₂DCFDA is primarily sensitive to hydrogen peroxide, it can also detect other reactive oxygen species, such as hydroxyl radicals. Importantly, our study does not aim to discriminate between individual ROS types, but rather to assess the overall impact of oxidative stress on SC-β cell differentiation. Moreover, ROS levels were measured at a single time point per cell, without tracking fluctuations over time in each cell. Therefore, these limitations did not compromise the interpretation of our results.

Shorter version of method characteristics now included in the Discussion: *“In our study, intracellular oxidative stress was assessed using H₂DCFDA, a widely used ROS-sensitive fluorescent probe. While primarily detecting hydrogen peroxide, H₂DCFDA can also respond to other ROS and is subject to time-dependent fluorescence decay. As our analysis focused on overall oxidative stress at a single time point rather than dynamic ROS fluctuations or specific ROS species, these limitations do not affect the interpretation of our findings regarding β cell development.”*

In Figure 3K, terms related to oxidative stress (“reactive oxygen species pathway”, “hypoxia”, “fatty acid metabolism”) were among the top scores for MSigDB-Hallmark annotations in differential pathway analysis of the KO and WT trajectories and PEN/PFG clusters. The analysis was done using Enrichr in Seurat.

5) Figure 2 demonstrates that Gpx2 loss effects the initial DE◊ PFG stage upregulating HNF4, CDX2 and Pdx1 while downregulating Hhex compared with WT hES. It is curious that Pdx expression is upregulated in these cultures and this expression is termed precocious because later in differentiation, these cultures appear to have less Pdx expression than controls.

We appreciate your comment on our use of "precocious" in Fig. 2. We initially chose this term because PDX1 expression is clearly increased in KO PFG at earlier stages. And as you note,

these expression levels do become more similar between WT and KO at later stages. Further, we also observed that HHEX is downregulated in KO PFG. Since HHEX is known to inhibit pancreatic fate, its downregulation further supports a stronger, earlier activation of the pancreatic developmental program.

Our interpretation is that KO PFG rapidly and aberrantly upregulates markers and programs characteristic of various endoderm lineages, including HNF4A, AFP, FGB, and CDX2 (as shown in Figure 5B-C). These endodermal cells co-expressing PROX1, PDX1 HNF4A, HHEX, CDX2, AFP, FGB also exist in developing human pancreas at week 4 (see Rebuttal Fig. 3). Previous studies showed that the ventral foregut endoderm is multipotent for the hepatic and pancreatic programs in mice²³⁻²⁵. However, recent work by Dr. Francesca Spagnoli laboratory revealed mutually exclusive signaling signatures in hepatic and pancreatic progenitors²⁶.

Rebuttal Fig. 3. Marker gene expression in the developing human pancreas at week 4²⁷.
A. Proportions of endoderm-derived progenitor lineages (pancreatic non-endocrine, duodenal, and hepatic) within the developing human pancreas at PCW 4.
B. Expression patterns of selected marker genes in progenitor cells differentiating toward pancreatic non-endocrine, duodenal, and hepatic lineages.

C. Co-expression of lineage-specific marker genes in the developing human pancreas at PCW4. Pancreatic non-endocrine (PEN) cells are marked in red, duodenum in green and liver in blue.

Once cells commit to hepatic or intestinal fates, or express these markers, they cannot fully respond to retinoic acid and revert to a pancreatic cell fate under our differentiation conditions. Our scRNA-seq data analyses further confirm that the proportion of "hepatic-like" or "multilineage" cells remains at later developmental stages (day 21). We acknowledge the possibility that cells formed at the PFG stage could be eliminated, and new multilineage/hepatic-like progenitors might form at later stages.

To avoid any potential misconceptions and reflect the dynamic nature of PDX1 expression more accurately, we have changed "precocious" to "altered" in the title of Revised Fig. 2. This modification better encompasses the observed changes without implying a permanent or strictly accelerated development, aligning with our understanding that the KO leads to an alternation rather than just an earlier onset of pancreatic differentiation.

Revised Figure 2 title reads "*GPX2 deficiency results in altered differentiation of the posterior foregut endoderm.*"

6) The use of "altered spatial patterning" in PFG is confusing and inaccurate. If individual spheroids demonstrate a reproducible "pattern" in WT cultures this might be correct but the images shown demonstrate that individual spheroids exhibit wide-spread cell to cell heterogeneity thus making conclusions from scRNA seq about patterning is not accurate. Temporal patterns might be a more accurate term but "spatial patterning" is certainly not correct, given the data presented.

We thank the Reviewer for this suggestion regarding the use of "altered spatial patterning." We agree that, given the observed cell-to-cell heterogeneity within spheroids and the fact that an *in vitro* hPSC system does not precisely reflect *in vivo* development, the term "spatial patterning" might be confusing in the context of the presented data. Therefore, we have revised the manuscript accordingly, and where appropriate, we have rephrased our descriptions to reflect "temporal patterns" rather than "spatial patterning."

7) A concern for the PEN (pancreatic endoderm) experiments outlined in Figure 3 includes the difference between the two WT replicates in the PC analysis at the PFG stage.

WT and KO cells at the PEN stage (day 8 of differentiation) were analyzed using scRNA-seq. For dimensionality reduction and visualization, we used UMAP analysis, which is a nonlinear extension of the PCA method. UMAP simplifies high-dimensional data while preserving local structure and relationships between individual cells, providing clearer separation and more intuitive interpretation of cell clusters²⁸. In contrast, Fig. 3B presents PCA analysis based on bulk RNA-seq data, performed in duplicates for both WT and KO cells at the PFG stage. Please see our response to your comment number 3 for additional context.

8) Figure 3K is inadequately described. The plots depicting cell program use is hard to read.

Thank you for letting us know this was not clearly presented. In response to your comment, we updated the Revised Fig. 3K and the related plots to include cell fate trajectories (Revised Fig. 3J). Please find the updated plots below.

Revised Fig. 3. J. UMAP plot visualizes the differentiation trajectory from foregut to pancreatic endoderm. Cells are color-coded by genotype. The black arrow indicates the trajectory. Gray cells are not assigned to the trajectory. Each dot represents a single cell. **K.** The plots show the mean score of selected hallmark gene sets from the Molecular Signature Database along pseudotime for WT and KO trajectories, as depicted in (J). Grey areas depict 95% confidence intervals. The violin plots at the bottom represent the pseudotime distribution of analyzed clusters for cells within the trajectories.

In Fig. 3K we present curves for WT and KO cells showing mean MSigDB terms scores along pseudotime. The plots include only cells belonging to a PEN → PFG trajectory (now shown in the Revised Fig. 3J). We have modified the colors in Revised Fig. 3J for WT and KO cells in the trajectory to match the line colors in Revised Fig. 3K. We also added pseudotime values along the drawn trajectory line in Revised Fig. 3J to correlate specific pseudotime points in Revised Fig. 3K with cells shown in the UAMP space in Revised Fig. 3J. For the same reason, the violin plot was initially included in Fig. 3J to show how clusters are aligned within the plotted trajectory. To increase the clarity of Revised Fig. 3K, we have now moved this violin plot to the bottom right part of Revised Fig. 3K and simultaneously adjusted the x/y-axes labels. Importantly, our analysis is supported by statistical analysis, which we have now emphasized by including information in the Revised Fig. 3K caption that: “Gray areas depict 95% confidence intervals.”.

Figures 3J and K have been updated in the revised MS.

9) Is *Bmp6* normally upregulated by the endoderm during intestine or liver development? BMPs are typically added to the media to induce PFG to induce hepatic fate. This suggests that the bias in these cultures is due to aberrant gene expression caused by oxidative stress conditions rather than a normal developmental process.

Thank you for bringing this point up. Indeed, the primary source of BMPs in liver development is the septum transversum mesenchyme (STM) adjacent to the developing liver and these signals are recapitulated *in vitro* by the addition of BMP4 to the medium.

Interestingly, we found that some early PFG epithelial and endodermal cells also expressed BMPs. For this, we have analyzed publicly available human PCW5-6 scRNA-seq dataset from Prof. Ludovic Vallier group (please see below Rebuttal Fig. 4) and found that the main source of BMPs is stellate cells. At the same time, endoderm-derived hepatocytes and cholangiocytes also express some BMPs (e.g. BMP4, and, specifically, BMP7), and BMP receptors along with SMADs.

Rebuttal Fig. 4. BMP expression in developing human liver. A-B UMAP plots of PCW5-9+5 liver cells subset from a comprehensive fetal (PCW5-17) and adult liver scRNA-dataset²⁹ showing **A.** PCW distribution and **B.** Cell types. **C.** Expression of *BMP*, *BMPR*, and *SMAD* transcripts in PCW5-6 liver. There were no endothelial cells sequenced at these ages.

We observed similar expression patterns of *BMPs*, *SMADs*, and *BMPRs* in human PCW4 fetal scRNA-seq data as below (Rebuttal Fig. 5):

Rebuttal Fig. 5. BMP expression in fetal PCW4 PFG. A-B. UMAP plots of PCW4 epithelial and endoderm subset from a comprehensive whole gastrulating embryo (PCW3-12) scRNA-dataset³⁰ showing: **A.** selected lineages and **B.** identified Seurat clusters. **C.** Feature plot maps showing expression of transcription factors expressed in PFG. **D.** Expression of *BMP*, *BMPR*, and *SMAD* transcripts in PCW4 PFG cluster 3 and early endodermal clusters (mostly, early liver endoderm).

However, the expression of BMPs in hPSCs differentiated towards PFG endoderm was also observed by Funa et al. 2024³¹. A panel from Suppl. Fig. 5 from this publication is provided below, showing BMP ligand genes expression at days 5 and 7, with a peak in *BMP6* expression at day 5, corresponding to our day 6 PFG results, included below as Rebuttal Fig. 6.

[FIGURE REDACTED]

Rebuttal Fig. 6. Expression of BMP ligand genes on days 5 and 7 during HUES4 hESC differentiation immature β cells. Data are presented as mean \pm SD. Rebuttal Fig. 6 corresponds to Supplementary Figure 5S from the publication by Funa *et al.* (2024), entitled “TGF- β modulates cell fate in human ES cell-derived foregut endoderm by inhibiting Wnt and BMP signaling”.³¹

Interestingly, we found that among *PRDXs*, *GPXs*, and *SODs* genes expressed in the developing human liver based on PCW5-6 scRNA-seq dataset²⁹, the *GPX2* transcript is the only one expressed at a very low level by liver cells, as shown below. Thus, *in vitro* *GPX2* KO might promote liver cell fate by recapitulating its absence in early liver cells. We have added these results to the supplementary data (Revised **Suppl. Fig. 6H**).

Revised Suppl. Fig. 6H. Oxidative stress response genes expression in early PCW5-6 human liver. The higher average gene expression is shown in red, and lower expression in blue.

The following sentence has been added to the Results section: “Interestingly, analysis of a scRNA-seq dataset from developing human liver (PCW5–6) reveals that among the *PRDX*, *GPX*, and *SOD* gene families, *GPX2* is the only transcript expressed at a very low level in liver cells (**Suppl. Fig. 6H**).”.

10) Fig 3L how was the period of oxidative stress induction chosen?

Oxidative stress was induced on day 5, based on the dynamics of *GPX2* expression. On day 6, we observed the appearance of *GPX2* protein during pancreatic *in vitro* differentiation. As

GPX2 acts as antioxidant, we assumed that its expression may be triggered by increased oxidative stress, which led us to induce oxidative stress shortly before GPX2 protein expression starts. In our experiments, repeated treatment with H₂O₂ caused cell death.

The following sentence is now included in the revised MS: “We determined that GPX2 protein expression emerges at day 6 of pancreatic differentiation (Fig. 11). As GPX2 acts as antioxidant, we assumed that its expression may be triggered by increased oxidative stress, which led us to induce oxidative stress shortly before GPX2 expression starts.”

11) Fig 3M- it is unclear what groups the p values provided are comparing. Important controls that are missing include the addition of 0.01 uM peroxide to KO cultures and the addition of 100pM selenium to WT cultures.

We thank the Reviewer for this valuable suggestion. We fully agree that these additional controls are important. To address this point, we repeated the pro-pancreatic differentiation experiments up to day 8, as described in the manuscript, but included the suggested controls: 100 pM selenium was added to WT cells, and 10 nM H₂O₂ was added to KO cells at the selected time point (differentiation day 5). We did not observe any changes upon these control treatments. The results of our analysis are presented below (Rebuttal Fig. 7) and also included to the manuscript in Revised Fig. 3N. The following sentences have been added to the Results section: “As controls, we treated WT cells with 100 pM Se and KO cells with 10 nM H₂O₂.” and “PDX1 expression was not changed in applied control conditions.”

Rebuttal Fig. 7. Quantification of PDX1 protein levels based on IF staining in WT and KO PENs differentiated with or without oxidative stress modulation. WT and KO cells were treated

with 10 nM H₂O₂ and 100 pM selenium. WT cells are represented by white dots, and KO cells by gray dots. Each dot corresponds to one image; the mean and 95% confidence intervals (CIs) are indicated. Statistical significance was assessed using the ANOVA test, with *p*-values shown on the right side of the graph.

To clarify which groups the provided *p*-values correspond to, we have modified the sentence in Revised Fig. 3N caption, which now states “*The p-values are colored according to the control to which the comparison refers, i.e. orange means comparison to WT without added H₂O₂ and blue to KO without selenium*”, as well as added legend in the right top corner of the Revised Fig. 3N as below:

Revised Fig. 3N. Quantification of IF staining for WT and KO PENs differentiated with or without oxidative stress modulation. WT and KO cells were treated with varying concentrations of H₂O₂ and selenium, respectively. As controls, WT cells were treated with 100 pM Se and KO cells with 10 nM H₂O₂. WT is marked in orange, and KO is in blue. Each dot represents one image, with mean and 95% confidence intervals (CIs) indicated. N = 3 independent experiments. An unpaired t-test was used to determine statistical significance. The *p*-values are colored according to the control to which the comparison refers, i.e. orange means comparison to WT without added H₂O₂ and blue to KO without selenium.

12) While the images shown in figure 3N highlight the points outlined in the paper in the absence of quantitation, they lack significance.

The quantification was included in the original submission as in Figure 3M (Revised Fig. 3N).

13) Why does the DAPI staining of the spheroids in Fig 3N, representing PEN, look so different from the cultures in 4A, representing SC-B spheroids? 3N looks tubular and 4A looks circular? They all seem to be taken at a similar magnification.

The different appearance of nuclei in Figures 3N and 4A results from the different differentiation formats: Fig. 3N is in 2D (Revised Fig. 3M), and Fig. 4A is in 3D. Please, also note that even in 2D format cells are not growing as single-cell layer but some miniature 3D formation can be formed. To address your concerns, we are now providing the zoomed inserts for Fig. 3N (Revised Fig. 3M) and below.

Revised Fig. 3M. Representative images of WT and KO PENs differentiated with or without regulation of oxidative stress. To modulate oxidative stress, 10 nM H₂O₂ or 100 pM Se was added to WT and KO cells, respectively. Cells were stained for PP marker PDX1 (red) or ECM component, COL4A1 (green). DAPI stains nuclei (blue). Scale bar = 100 μ m. *N* = 3 independent experiments. Zoomed-in inserts are also shown.

14) There is an apparent hepatic bias to the Gpx2 KO (versus WT) hP cells during pancreas-induction. In addition, a subset of intestinal and bile duct gene programs are also upregulated. Are these each represented in one of the 3 distinct GPX KO ML populations present after pancreatic differentiation in Figure 4? Or are these ML populations entirely multilineage?

Analysis of all cell clusters present at day 21 scRNA-Seq suggested that there were ML subpopulations that distinctly expressed liver, bile duct, and duodenal marker genes (Fig. 4M). To get a more detailed look at the progenitor cells, we reanalyzed the MLP2 and ML clusters after separating them from the rest of the dataset. We didn't include the MLP1 cells in this analysis because they don't show markers for the liver, bile ducts, or duodenum. We have reclustered these populations and identified two MLP2 and three ML1 subpopulations (please see below, Rebuttal Fig. 8 A and B). While MLP2 subpopulations do not express lineage markers at high levels, ML_1 subcluster has the highest expression of liver and intestinal markers, while ML_2-3 subclusters are enriched in bile duct markers (Rebuttal Fig. 8C).

Rebuttal Fig. 8. Subclustering and re-analysis of ML and MLP2 clusters from KO cells at SC- β stage. A. UMAP showing original identities MLP2 and ML of subclustered cells. **B.** UMAP showing MLP2 and ML subclusters. **C.** Dot plot shows the expression of liver, bile duct (bd) and duodenal markers as in Revised Fig. 4M.

15) There is no direct evidence to support their other conclusion that ECM alterations drive BMP expression that in turn promotes hepatic fates.

Please see our response to question 4.

16) Oxidative stress is a well-known inducer of fibrosis (excess ECM)- could the changes noted in the ECM be the result of a fibrotic response?

Thank you for this interesting suggestion. We thoroughly evaluated the expression of a cohort of fibrosis-related genes, including COL1A1, ACTA2³², CYGB, TAGLN³³, POSTN³⁴ and PRSS1³⁵. For most of these genes, we did not observe the upregulation in the KO subpopulation. However, for some, such as CASR and SPINK1³⁵, the scRNA-seq showed

higher expression in the KO (Rebuttal Fig. 9). Please note that these genes are not only associated with fibrosis, but CASR is a key regulator of parathyroid hormone (PTH) secretion³⁶, while SPINK1 plays a protective role in exocrine pancreatic tissue by inhibiting the premature activation of trypsin within the pancreas³⁷. Furthermore, the KEGG and GO analyses of differentially expressed genes in KO did not indicate significant changes in fibrosis-related terms. Lastly, fibrosis is less frequently detected during development due to the different repair and regeneration capacity of forming organs vs. postnatal ones. Overall, we are inclined to conclude that a primary fibrosis response is not observed in KO.

Rebuttal Fig. 9. Expression levels of fibrosis-related genes in WT and KO cells at SC- β cell stage, based on scRNA-seq data from day 21 of differentiation. Expression is visualized using a dot plot, and selected genes are further shown on UMAPs.

17) The authors demonstrate that in the absence of Gpx2 hPSC will turn on a number of hepatic genes. This cell population is termed “hepatic progenitors”. How close are hPSC induced hepatic progenitors or *in vivo* examined hepatic progenitors to the hepatic progenitors produced by Gpx2 deficiency herein?

We thank the Reviewer for this highly accurate question. We would like to clarify that it was not our intention, nor did we expect, to generate pure or highly comparable hepatic progenitors to those derived through directed *in vitro* differentiation protocols or *in vivo* sources. Rather, our scRNA-seq analysis revealed the presence of KO-specific cell population expressing liver-associated genes, even under conditions strongly promoting pancreatic differentiation. This observation suggests that KO cells may exhibit predisposition toward hepatic lineage differentiation compared to WT cells. To explore this possibility experimentally and assess the differentiation bias of KO cells, we performed a spontaneous differentiation assay. Both WT and KO hPSCs were differentiated to the DE stage and subsequently cultured in a basal differentiation medium without the addition of lineage-specific small molecules, allowing for unbiased spontaneous differentiation. By day 8 of this spontaneous differentiation, we observed a significant upregulation of HNF4A and FGB protein expression in KO cells, along with a trend toward increased AFP expression, relative to WT cells (Fig. 5B-D in the manuscript). Notably, PDX1 expression was absent in WT and KO cells under these conditions (Fig. 5B). In contrast, overexpression of GPX2 resulted in a significant downregulation of AFP and FGB compared to WT cells (Fig. 5E), further supporting the role of GPX2 in modulating hepatic lineage potential. Taken together, these findings suggest that *GPX2* KO cells may have an enhanced tendency or efficiency to differentiate into liver-like cells compared to their WT counterpart. However, to eliminate any confusion and to emphasize that we do not expect our differentiation to yield hepatic progenitors, we have modified the manuscript by changing the terms ‘hepatic progenitors’ to ‘hepatic-like progenitors’, ‘liver progenitors’ to ‘liver-like progenitors’ and “hepatic lineage” to “hepatic-like lineage”.

Additionally, to check how close *GPX2*-deficient progenitors are to hepatic progenitors from *in vitro* differentiation or *in vivo* sources, we re-analyzed our and publicly available scRNA-seq data. Specifically, we integrated three datasets (Rebuttal Fig. 9A):

1. Our *GPX2* KO day 21 data for MLP1, MLP2 and ML clusters, together with endocrine cell clusters, the latter as control, as they should cluster separately from ML and hepatic lineages.
2. Ori *et al.*³⁸ dataset of *in vitro* differentiation days 7 to 15, which is to our knowledge the only scRNA-seq dataset for early *in vitro* liver progenitors, yet these cells were found off-target cells during lung differentiation. For integration we selected foregut endoderm (Ori_4, Ori_12) and early liver-like (Ori_0 and Ori_10) clusters.
3. Weasley *et al.*²⁹, fetal liver subset of PCW5-6 hepatoblast clusters (HB1, HB2).

We then integrated these datasets using the STACAS method and identified 18 clusters (Rebuttal Fig. 10A, B). We found that fetal hepatoblasts clustered separately with some overlap with the Ori dataset, while MLP1 and ML clusters had some similarity with the *in vitro* clusters from Ori; this overlap constituted clusters 0, 1, 3 and 4. As expected, our endocrine cells clustered separately from hepatoblasts and *in vitro* derived hepatic progenitors. We then have analyzed clusters 0, 1, 3, and 4 in terms of cell proportions from each dataset (Rebuttal Fig. 10C) and found clusters 0 and 1, predominantly populated by ML and MLP cells to include ~20-25% of Ori cells, while cluster consisting mainly Ori cells contained ~8% of ML cells. We then analyzed cells from Ori and Weasley within these clusters by time (Rebuttal Fig. 10D) and found cluster 1 to include more immature cells from days 7-11, while cluster 0 cells from days 10-15. Clusters 4 and 3 represented increasing maturation of *in vitro* liver-like cells, the latter including mainly fetal hepatoblasts with only some older Ori cells, suggesting that even in this liver progenitor permissive protocol, at day 15, these cells do not achieve huge proportion of PCW5-6 like cells. We then looked at cell type proportions within clusters 0, 1, 3 and 4 (Rebuttal Fig. 10E), and confirmed cluster 1 consists of less mature cells, i.e. mostly MLP1 and Ori_4/12, while cluster 0 of more mature MLP2 and ML cells together with liver progenitor Ori_0 cluster.

Together, this analysis suggest that the most mature *GPX2* KO ML cells are similar to some extent to liver progenitors at the days 10-15 of *in vitro* differentiation, yet most of ML cells persist in a less advanced stage, likely due to the fact our protocol was strongly promoting pancreatic endocrine cell formation.

Rebuttal Fig. 10. Similarity of *GPX2* deficient ML cells to *in vitro* derived and fetal human hepatoblasts. **A.** UMAP showing distribution of STACAS-integrated datasets, with denoted lineages from our data, **B.** identified Seurat clusters shown in the UMAP space, **C.** Bar plot

showing normalized proportion of cells within clusters shared by our and Ori *et al.* cells, **D.** Bar plot showing normalized proportion of cell ages from Ori *et al.* and Weasley *et al.* datasets within selected clusters, **E.** Bar plot showing normalized proportion of cell types within selected clusters.

18) The conclusion that “ECM remodeling emerges as a key driver of lineage plasticity” is not well-supported by the data presented and appears mainly correlative.

We refer the Reviewer to our answer to comment number 4. Further, we have replaced the word “key” to “important”.

19) The connection between ECM production, TGF β superfamily signaling and hepatic fate has not been established.

We kindly refer the Reviewer to our response to Question 4.

Reviewer #2 (Remarks to the Author):

In this study, Szpotkowska *et al.* investigated the role of GPX2 in human pancreatic differentiation and revealed oxidative stress as a key determinant of pancreatic versus non-pancreatic cell fate. The authors demonstrated that GPX2-deficient cells performed increasing hepatic differentiation within a pancreatic differentiation system, highlighting the critical role of GPX2 in guiding lineage commitment during progenitor differentiation. Mechanistically, they showed that loss of GPX2 induces extracellular matrix (ECM) remodeling, which in turn activates BMP signaling and diverts differentiation away from the pancreatic lineage. Furthermore, manipulating ROS levels, either by induction or alleviation, was sufficient to mimic or rescue the phenotypes associated with GPX2 deficiency. Overall, the data are comprehensive, clearly presented, and logically organized, offering novel insights into the regulation of endodermal differentiation through redox signaling. In addition, the single-cell dataset provide rich information for the pancreatic differentiation field for future exploration. However, several key issues need to be addressed. Here are the detailed comments:

Major comments:

1. In Figure 1I, the Western blot control is inconsistent. GAPDH expression in SC- β cells is much lower than in other samples, which may compromise the conclusion that SC- β cells do not express GPX2. More uniform loading or replicate blots would strengthen this claim.

We thank the Reviewer for this suggestion. We have replicated the WB and now are presenting the SC- β cell sample with more comparable GAPDH expression to the other stages. Here is updated WB, as shown in revised Figure 1I below.

Revised Fig. 1I. Western blot depicts GPX2 protein levels during hPSC pancreatic differentiation. Days of cell collection are shown. GAPDH was used as a loading control.

2. In Figures 2F and 2G, although some HNF4A⁺/FGB⁺ cells are detected, the number of positive cells appears very low. Quantitative comparison using FACS is recommended to more accurately assess the differences between WT and KO cells.

Thank you for this suggestion. As recommended, we performed flow cytometry analysis of FGB and HNF4A expression on days 6 and 8 of WT and KO cell differentiation. These experiments confirmed our IF-based observation, specifically the upregulation of FGB⁺ and HNF4A⁺ cells, and cells co-expressing FGB and HNF4A, at both time points in KO cells. The effect was more pronounced on day 8. We have included the representative FACS plots for days 6 and 8, in Revised Fig. 2H. For convenience we combined this panel with corresponding histograms and scatter plots as Rebuttal Fig. 11. The gating strategy is presented in Revised Suppl. Fig. 2F and Rebuttal Fig. 11.

gating strategy

day 8

day 6

Rebuttal Fig. 11. HNF4A and FGB expression levels in WT and KO cells at differentiation days 6 and 8. For each condition (WT and KO, days 6 and 8), data are shown as histograms and scatter plots. WT cells are depicted in yellow, KO cells in blue, and secondary antibody controls in gray. For each sample, the percentage of cells co-expressing FGB and HNF4A is indicated in blue on the diagram. Gating strategy is also shown.

3. In Figure 3N, it is noted that WT PEN cells show no PDX1 expression, especially since PDX1 was clearly detected in Figure 2E. This inconsistency raises concerns about the stability and efficiency of the differentiation system, and warrants further explanation or validation.

Thank you for your feedback regarding the PDX1 signal intensity and representativeness. We appreciate you bringing this to our attention. We have conducted over 15 biological replicates of the differentiation to pancreatic endocrine cells (PEN), each time verifying PDX1 expression. These extensive experiments consistently demonstrate clear PDX1 expression in WT cells and, importantly, increased PDX1 protein levels in KO cells. We have replaced the original submission images with new, more accurate, and clearer immunofluorescence images that reflect these consistent findings. We kindly refer the Reviewer to our response to Reviewer 1, Question 4 or to Revised Figure 3M.

4. In Figures 4A and 4B, the authors report an increase in AFP⁺ and FGB⁺ liver progenitors in KO cells. However, the immunostaining shows very few AFP⁺ cells. Quantitative methods such as FACS or image-based quantification should be used to confirm these differences between WT and KO.

As recommended, we quantified the IF staining for AFP at day 23 (SC-β cell stage) of WT and KO differentiation and observed a significant, almost threefold increase in AFP mean fluorescent intensity. We attempted to validate this finding using flow cytometry; however, the AFP antibody used in our study did not perform reliably in FACS, despite multiple optimization efforts. However, we confirmed using flow cytometry the presence of FGB⁺ cells exclusively in KO cells at day 27 (stage SC-β cells). These quantitative assessments support the observed differences between WT and KO cells. We have included this new data in Revised Suppl. Fig. 5D, E.

Revised Suppl. Fig. 5D-E. Changes in selected protein levels in GPX2-deficient cells. D. FGB expression was measured using flow cytometry, and results are presented on a histogram and scatter plot. WT and KO cells are presented in orange and blue, respectively. Control cells (stained only with secondary antibody) are marked in red. The percentage of cells that express FGB protein is also displayed. $N = 2$ biological repeats. E. Dot plot represents the quantification of mean fluorescence intensity of AFP relative to DAPI for WT (orange) and KO (blue) cells at the SC- β cell stage. Data are presented as mean on the plot; error bars are 95% CIs. The p -values were calculated using an unpaired t-test. $N = 4$ biological repeats.

5. From Figures 4A-B, GPX2 KO does not appear to affect the proportion of pancreatic β -like cells. However, the functional capacity of SC- β cells remains unclear. Does GPX2 loss influence β cell function, such as insulin content, mitochondrial morphology and number, or glucose-stimulated insulin secretion (GSIS)? Functional validation is recommended.

We thank the Reviewer for raising this important question. While we had not initially planned to investigate this in depth, your comment prompted us to examine the functional aspects of the cells. We have now assessed a cohort of genes associated with insulin secretion and glucose sensing, including *INS*, *MAFA*, *NEUROD1*, and *SLC2A2* at the mRNA level, and the results are presented in the panel below. The changes in mRNA expression levels for these genes in KO vs. WT SC- β cells were not significant.

Furthermore, we quantified mitochondrial content using MitoTracker Green and found no statistically significant differences between KO and WT SC- β cells, as shown in the accompanying panel. Finally, we measured total insulin secretion via ELISA following KCl-induced membrane depolarization, normalizing the data to cell number, and did not observe a significant increase in insulin secretion (see Rebuttal Fig. 12). Together, we concluded that KO SC- β cells are functionally at least comparable with WT cells.

Rebuttal Fig. 12. Functional comparison of WT and KO SC- β cells. **A.** ELISA results for total insulin secretion in WT and KO SC- β cells. $N = 3$ independent experiments. **B.** Dot plot representing mitochondrial mass (i.e., total mitochondrial content per cell) in WT (orange) and KO (blue) SC- β cells, assessed by flow cytometry using MitoTracker Green. Mean values and 95% confidence intervals (CIs) are shown. $N = 3$ independent experiments. Statistical significance was determined using an unpaired t -test. **C.** Representative histogram of total mitochondrial mass measured for WT (orange) and KO (blue) SC- β cells. **D.** Violin plots represent the expression of genes associated with insulin secretion and glucose sensing in WT (pink) and KO (green) SC- β cells.

	avg_log2FC	p-val_adj
PCSK1	nd	nd
PCSK2	7,07E-12	1,99E-07
CPE	0,377224416	2,85E-05
CHGA	0,319796584	0,797882327
CHGB	nd	nd
INS	0,477460829	2,87E-14
GCK	0,227650216	1
KCNJ11	0,394544704	1
SLC2A2	1,615009767	1
MAFA	0,649074233	1
NEUROD1	0,181827013	1
GLP1R	nd	nd

Rebuttal Table 1. Statistical significance and expression changes of selected genes involved in insulin secretion and glucose sensing. Adjusted *p*-values (p_val_adj) and average log₂ fold changes (avg_log2FC) are reported for the comparison between WT and KO cells.

Minor comments:

1. In Figure 1A, it is suggested to use double marker combinations to define cell stages more clearly—for example, DE (SOX17/FOXA2), PP (PDX1/NKX6.1), EP (CHGA/NKX2.2). Alternatively, FACS can be used to show stage-specific differentiation efficiency.

As requested, we are now including representative images of immunofluorescent staining for distinct stages, and along with flow cytometry plots showing the co-expression of stage-specific markers. IF staining using double-marker combinations are shown in Revised Fig. 1A, and the corresponding flow cytometry plots are presented in Revised Suppl. Fig. 1A. For convenience we are also showing this data below as Rebuttal Figure 13.

Rebuttal Fig. 13. Schematic representation of the stepwise differentiation of hPSCs into SC- β cells, with time points of sample collection indicated. Stage-specific marker expression is shown by immunofluorescence staining and quantitatively by flow cytometry. Day 4 (definitive endoderm) cells express SOX17 and FOXA2; day 12 (pancreatic progenitors) are PDX1⁺ and NKX6-1⁺, day 19 (endocrine progenitors) co-express PDX1 and CHGA; and day 24 (SC- β cells) co-express NKX6-1 and C-peptide. Flow cytometry plots corroborate the co-expression of these markers at each differentiation stage. Additionally, the flow cytometry plot for differentiation day 8 (pancreatic endoderm) shows co-expression of PDX1 and SOX9.

2. Cite PMID: PMC9293889, which also discusses pancreatic versus hepatic fate decisions at the progenitor stage.

Thank you for bringing our attention to this study. We have now referred to this work in MS.

The following sentence has been added to the Discussion section: “*To date, depletion of the m⁶A demethylase ALKBH5 has been shown to impair pancreatic lineage specification, while concurrently promoting the upregulation of hepatocyte-associated genes*³⁹.”

3. In Figure 6G and Figure 7E, some of the displayed loci show minimal or unclear differences in chromatin accessibility. If possible, the authors should highlight more pronounced differential peaks or adjust visualization (e.g., by changing scale, zooming in, or using a different normalization method) to better emphasize significant changes. Alternatively, a brief explanation for why these loci were selected (e.g., biological relevance despite modest changes) would be helpful.

Thank you for letting us know that the ATAC-seq data was not optimally presented. We have now updated the plots with tracks (see Revised Figures 5K, 6G, 7E and below combined as Rebuttal Fig. 14).

Rebuttal Fig. 14. ATAC-seq tracks highlight the loci of marker genes related to (A) liver cell fate, (B) ECM, and (C) TGF- β signaling in WT (orange) and KO (blue) cells differentiated spontaneously until day 8. Peaks represent normalized data combined from biological replicates ($N = 2$).

4. In the figure legends, specify whether bar plots represent mean \pm SD or mean \pm SEM to improve clarity.

We thank the Reviewer for bringing it up. In Figure 1B, bar plots represent mean \pm SEM. For the remaining graphs, we have specified that bar plots represent mean \pm 95% confidence

intervals (CIs). We have now updated the description for Figure 1B. We think that CIs directly communicate the precision of the estimated mean by showing a plausible range for the actual population mean. Thus, CIs are more informative for inferring statistical significance and understanding the certainty of your findings than standard deviation (SD) or standard error of the mean (SEM).

Caption for Figure 1B now reads: “*Changes in oxidative stress levels during in vitro SC-β cell differentiation measured by H₂DCFDA intensity relative to hPSCs. Data on the plot are presented as mean ± SEM.*”

Reviewer #3 (Remarks to the Author):

Reviewer #4 (Remarks to the Author):

This is a technically rigorous study that identifies GPX2, a redox-sensitive glutathione peroxidase, as a key regulator of human endodermal lineage decisions during pancreatic differentiation from human pluripotent stem cells (hPSCs). The authors demonstrate that GPX2 deficiency skews differentiation toward hepatic and intestinal fates, even under pro-pancreatic conditions, by altering oxidative stress levels and disrupting the cellular microenvironment, particularly ECM composition and BMP signaling. The work is conceptually novel and contributes important mechanistic insight into how redox balance regulates human cell fate determination, with potential implications for regenerative medicine and disease modeling.

Major concerns:

1. The manuscript could more clearly discuss whether the liver- and intestine-like cells represent mature or immature progenitor states. This is particularly relevant given the absence of ALB transcripts and the enrichment of multilineage progenitor markers.

Thank you for bringing up this crucial point. We agree that clarifying the maturity of the liver- and intestine-like cells is crucial, especially given the absence of ALB transcripts and the enrichment of multilineage progenitor markers. We would like to emphasize that throughout the manuscript, we consistently use terms such as "hepatic progenitors" or "hepatic-like progenitors" when referring to *GPX2* KO cells. This is precisely because we concur with your assessment: these cells represent an immature progenitor state, not mature, functional hepatocytes or intestinal cells.

It is also important to consider our differentiation protocol. We specifically designed it to induce pancreatic cell fates using a precise combination of growth factors and small molecules. Therefore, it would be highly unexpected for the loss of GPX2 and the resulting increase in oxidative stress alone to drive endodermal cells into fully mature hepatocyte-like or intestine-like cells. Our findings are consistent with these cells remaining in an earlier, progenitor state.

2. Figures summarizing scRNA-seq cluster proportions, pseudotime trajectories, and pathway enrichment could benefit from additional quantitative clarity. Including statistical comparisons between WT and KO proportions across clusters would enhance interpretability.

Thank you for this valuable comment. We have added plots showing cell frequencies according to genotype contribution within each cluster for both day 8 (Revised Suppl. Fig. 4H) and day 21 scRNA-seq datasets (Revised Suppl. Fig. 6G). For convenience we are also showing these data below as Rebuttal Figure 15.

:

Rebuttal Fig. 15. The bar plot shows the percentage of clusters assigned to each genotype - WT (orange) and KO (blue) - based on scRNA-seq data from days 8 and 21. Data are normalized to the total number of cells sequenced for each genotype.

To remove bias caused by the different number of cells sequenced for each genotype, the data is normalized to the number of cells sequenced for each genotype.

Regarding pseudotime trajectory analysis (Revised Fig. 3K), the existing gray areas define 95% confidence interval, which supports our conclusions. We have now included this missing information in Revised Fig. 3K caption by adding the sentence: “*Gray areas depict 95% confidence intervals.*”

Conversely, pathway enrichment analysis, e.g., Fig. 3G, and Fig. 7A are based on statistical analyses and terms *p*-values are plotted.

Minor concerns.

1. Clarify in the abstract that the knockout of GPX2 leads to a bias in lineage commitment rather than a complete block in pancreatic differentiation.

Thank you for your suggestion. Abstract has been updated accordingly.

Abstract now reads: “*Cell fate decisions in human endoderm development are tightly regulated with extensive transcriptional networks described, yet the mechanistic role of metabolic products in this process remains elusive. Endodermal posterior foregut (PFG) gives rise to multiple organs, like the pancreas, liver, and intestine. Here, we identify Glutathione Peroxidase 2 (GPX2) as a critical regulator of human PFG differentiation, revealing oxidative stress as a key determinant of pancreatic versus non-pancreatic cell fate. We demonstrate that GPX2-deficient hPSCs under pancreas-promoting conditions differentiate also into hepatic-like progenitors. Through bulk and single-cell RNA-sequencing, ATAC-sequencing, and functional studies, we reveal that GPX2 orchestrates lineage commitment by regulating PDX1 and other key transcription factors, shaping PFG patterning and leading to the emergence of multilineage pro-liver and pro-intestinal progenitors. Mechanistically, GPX2 deficiency triggers extracellular matrix (ECM) remodeling, activating BMP signaling, and skewing differentiation away from the pancreatic lineage. Manipulating oxidative stress, either by inducing or alleviating ROS levels, recapitulates or rescues the effects of GPX2 loss, establishing oxidative stress as a gatekeeper of pancreatic fate. These findings establish GPX2 as a critical redox-sensitive determinant of human pancreatic differentiation and underscore the impact of oxidative stress on human endoderm lineage specification. Controlling oxidative stress is a crucial tool for endoderm differentiation and could provide enabling technology in regenerative medicine.*”

2. Improve consistency in gene nomenclature and italicization throughout the manuscript.

Thank you for bringing it to our attention. Gene nomenclature and italicization are now uniformly present in the manuscript.

3. Confirm whether the observed increase in α cells (GCG⁺) in KO samples is statistically significant and discuss its implications.

Yes, the observed increase in GCG⁺ cells following GPX2 loss was statistically significant, with a *p*-value of 0.0004, as determined by an unpaired t-test from *N* = 3 independent experiments. Quantification of GCG⁺ cells was performed based on immunofluorescence staining, as shown in Supplementary Fig. 5C. Consistent with these findings at the protein level, scRNA-seq analysis at day 21 revealed an increased proportion of GCG⁺ cells within the multihormonal cluster in KO populations (4.2% vs. 3% in WT) (Rebuttal Fig. 16). Together, these results suggest that elevated oxidative stress during embryogenesis may skew the α/β cell proportion, as evidenced by the upregulation of α cell marker genes *ARX* and *ALDH1A1*⁴⁰ (Rebuttal Fig. 16). This shift may predispose individuals to abnormal glucose homeostasis and

potentially increasing susceptibility to conditions such as type 2 diabetes and metabolic syndrome in adulthood.

We have added the following sentences to the discussion: “*Our study suggests that elevated oxidative stress during embryogenesis may disrupt the α/β cell proportion, potentially predisposing individuals to impaired glucose homeostasis and increasing the risk of developing conditions such as type 2 diabetes and metabolic syndrome later in life.*”

Rebuttal Fig. 16. Expression of α cell marker genes *GCG*, *ARX*, and *ALDH1A1* expression in WT and KO cells at the SC- β cell stage, visualized on UMAPs. Cells that express marker gene are highlighted in purple.

Literature:

1. Wolosowicz, M., Prokopiuk, S. & Kaminski, T. W. The Complex Role of Matrix Metalloproteinase-2 (MMP-2) in Health and Disease. *Int J Mol Sci* **25**, 13691 (2024).
2. Xiao, Y. *et al.* Material-driven fibronectin and vitronectin assembly enhances BMP-2 presentation and osteogenesis. *Mater Today Bio* **16**, 100367 (2022).
3. Sedlmeier, G. & Sleeman, J. P. Extracellular regulation of BMP signaling: welcome to the matrix. *Biochem Soc Trans* **45**, 173–181 (2017).
4. Wang, X., Harris, R. E., Bayston, L. J. & Ashe, H. L. Type IV collagens regulate BMP signalling in *Drosophila*. *Nature* **455**, 72–77 (2008).
5. Fitzpatrick, V. *et al.* Signal mingle: Micropatterns of BMP-2 and fibronectin on soft biopolymeric films regulate myoblast shape and SMAD signaling. *Sci Rep* **7**, 41479 (2017).
6. Rossi, J. M., Dunn, N. R., Hogan, B. L. & Zaret, K. S. Distinct mesodermal signals, including BMPs from the septum transversum mesenchyme, are required in combination for hepatogenesis from the endoderm. *Genes Dev* **15**, 1998–2009 (2001).
7. Palaria, A., Angelo, J. R., Guertin, T. M., Mager, J. & Tremblay, K. D. Patterning of the hepato-pancreatobiliary boundary by BMP reveals heterogeneity within the murine liver bud. *Hepatology* **68**, 274–288 (2018).
8. Tsai, M.-S. *et al.* Proper BMP Signaling Levels Are Essential for 3D Assembly of Hepatic Cords from Hepatoblasts and Mesenchymal Cells. *Dig Dis Sci* **60**, 3669–3680 (2015).

9. Imamura, S., Yoshimoto, K., Terada, S., Takamuro, K. & Kamei, K.-I. In vitro culture at 39 °C during hepatic maturation of human ES cells facilitates hepatocyte-like cell functions. *Sci Rep* **12**, 5155 (2022).
10. Ang, L. T. *et al.* A Roadmap for Human Liver Differentiation from Pluripotent Stem Cells. *Cell Rep* **22**, 2190–2205 (2018).
11. Zhao, D. *et al.* Promotion of the efficient metabolic maturation of human pluripotent stem cell-derived hepatocytes by correcting specification defects. *Cell Res* **23**, 157–161 (2013).
12. Si-Tayeb, K. *et al.* Highly efficient generation of human hepatocyte-like cells from induced pluripotent stem cells. *Hepatology* **51**, 297–305 (2010).
13. Hannan, N. R. F., Segeritz, C.-P., Touboul, T. & Vallier, L. Production of hepatocyte-like cells from human pluripotent stem cells. *Nat Protoc* **8**, 430–437 (2013).
14. Ziojła, N. M. *et al.* ETVs dictate hPSC differentiation by tuning biophysical properties. *Nat Commun* **16**, 1999 (2025).
15. Functional, metabolic and transcriptional maturation of human pancreatic islets derived from stem cells | Nature Biotechnology. <https://www.nature.com/articles/s41587-022-01219-z>.
16. Lotto, J., Stephan, T. L. & Hoodless, P. A. Fetal liver development and implications for liver disease pathogenesis. *Nat Rev Gastroenterol Hepatol* **20**, 561–581 (2023).
17. Choi, T.-Y. *et al.* Bone morphogenetic protein signaling governs biliary-driven liver regeneration in zebrafish through *tbx2b* and *id2a*. *Hepatology* **66**, 1616–1630 (2017).
18. Soh, C.-L. & Huangfu, D. CRISPR/Cas9-Mediated Mutagenesis of Human Pluripotent Stem Cells in Defined Xeno-Free E8 Medium. *Methods Mol Biol* **1498**, 57–78 (2017).
19. Zhang, J. *et al.* Systematic identification of anticancer drug targets reveals a nucleus-to-mitochondria ROS-sensing pathway. *Cell* **186**, 2361-2379.e25 (2023).

20. Song, J. *et al.* FOXO-regulated OSER1 reduces oxidative stress and extends lifespan in multiple species. *Nat Commun* **15**, 7144 (2024).
21. Guo, Y. *et al.* Oxidative stress-induced FABP5 S-glutathionylation protects against acute lung injury by suppressing inflammation in macrophages. *Nat Commun* **12**, 7094 (2021).
22. Zhang, Y., Cai, Y. & Chen, Z. Community-specific diffusion characteristics determine resistance of biofilms to oxidative stress. *Sci Adv* **9**, eade2610 (2023).
23. Deutsch, G., Jung, J., Zheng, M., Lóra, J. & Zaret, K. S. A bipotential precursor population for pancreas and liver within the embryonic endoderm. *Development* **128**, 871–881 (2001).
24. Tremblay, K. D. & Zaret, K. S. Distinct populations of endoderm cells converge to generate the embryonic liver bud and ventral foregut tissues. *Dev Biol* **280**, 87–99 (2005).
25. Miki, R. *et al.* Fate maps of ventral and dorsal pancreatic progenitor cells in early somite stage mouse embryos. *Mech Dev* **128**, 597–609 (2012).
26. Willnow, D. *et al.* Quantitative lineage analysis identifies a hepato-pancreato-biliary progenitor niche. *Nature* **597**, 87–91 (2021).

REVIEWER COMMENTS

Reviewer #1 (Remarks to the Author):

The authors have addressed many of the concerns listed in the previous review. Several concerns remain and should be addressed.

1) I still have reservations about the cell lines used. The authors originally omitted any information on which cell lines they used. The authors now state that the cell lines used for the KO experiments were produced from the HUES8 parental line (the methods state that the Cas9-HuES8 line was used to generate the knockout) and the over expression lines produced in the H1 and HUES8 background. Were the experiments with the KO cells performed with a single clone or were they mainly from 1 of the 2 KO clones? Which cell line was used as the control for the KO experiments? There are methods describing the creation and use of the DOX inducible HUES8 cell line but no mention of how the H1 over-expressing cell line was created. This information should be provided in the methods or in the body of the paper. Because my original concerns were not addressed; please provide details on the exact cell lines (and KO clone) used to generate the data in Figs 2-5.

Thank you for your valuable feedback and for giving us the opportunity to address your concerns more thoroughly. We sincerely apologize for not providing sufficient detail about the cell lines in our original submission.

To ensure clarity, we provide here a complete breakdown of the cell lines used in each figure:

Fig. 2A: clonal line KO1

Fig 2B: clonal line KO1

Fig. 2C: clonal line KO1, and in corresponding **Suppl. Fig 2F:** data for KO2 and KO3

Fig. 2D: clonal line KO1, and in corresponding **Suppl. Fig 2G:** data for KO2

Fig. 2E-G: clonal line KO1

Fig. 2H: clonal line KO2, and in corresponding **Suppl. Fig 2I, J:** data for KO1 and KO2

Fig. 3A: N/A, a scheme

Fig. 3B-K: clonal line KO1

Fig. 3L: N/A, a scheme

Fig. 3M-O: clonal line KO1, and in corresponding **Suppl. Fig. 4J:** data for KO2

Fig. 4A-C: clonal line KO1, and in corresponding **Suppl. Fig. 5A-E:** data for KO1, and **Suppl. Fig. 5F-G:** data for KO2

Fig. 4D-N: clonal line KO1, and in corresponding **Suppl. Fig. 6A-H:** data for KO1

Fig. 5A: N/A, a scheme

Fig. 5B: clonal line KO1

Fig. 5C: clonal line KO2

Fig. 5D: clonal line KO1 and KO2

Fig. 5E-F: OE in HUES8, and corresponding **Suppl. Fig. 7A-C:** OE in HUES8 background and **Suppl. Fig. 7D:** OE in HUES8 and H1 hESC background

Fig. 5G-H: clonal line KO2

Fig. 5I-J: clonal line KO1, and corresponding **Suppl. Fig. 7E-I:** clonal line KO1

Fig. 5K-L: clonal line KO2, and corresponding **Suppl. Fig. 7J-N:** clonal line KO2

Fig. 6A-F: clonal line KO1, and corresponding **Suppl. Fig. 7E-I:** clonal line KO1

Fig. 6G: clonal line KO2

Fig. 6H: HNF4A, LAMA1, LAMA5, SOX9: clonal line KO1, COL4A: clonal line KO2

Fig. 7A-D: clonal line KO1 and corresponding **Suppl. Fig. 8A-B:** clonal line KO1

Fig. 7E: clonal line KO2 and corresponding **Suppl. Fig. 8C:** clonal line KO2

Fig. 7F-L: clonal line KO1 and corresponding **Suppl. Fig. 8D-E:** clonal line KO1.

While in our initial submission we did not clarify exactly KO clonal lines, during the study, we performed validation experiments across multiple independent clones. Specifically:

- All initial experiments: such as hPSC characterization, phenotyping (pancreatic differentiation and stage-specific markers evaluation) were performed on at least two independent *GPX2* KO clonal lines (KO1 and KO2), both on the HUES8 background,
- For resource-intensive experiments (e.g., scRNA-seq), we selected KO1 based on preliminary validation showing consistent phenotypes between clones,
- We have now included data from KO2 in Supplementary Figures 2F-G, 2I-J, 5D, and 5G, demonstrating that the effects are reproducible across independent clones.

In response to your comments, we have added more data on KO2:

- Suppl. Fig. 2C: Growth curves of WT, KO1, and KO2 lines by live-cell imaging,
- Suppl. Fig. 2F: Mean fluorescence intensity (MFI) quantification of stage-specific marker protein level at DE stage for WT, KO2 and KO3
- Suppl. Fig. 2G and 5F: Mean fluorescence intensity (MFI) quantification of stage-specific marker protein level at PEN and SC- β stages for WT and KO2
- Suppl. Fig. 2I-J and 5G: FGB protein levels in WT, KO1, and KO2 lines at PFG, PEN, and SC- β stages assessed by flow cytometry (also shown in Rebuttal Figure 2)
- Suppl. Fig. 7D: Pancreatic differentiation results from the H1-based *GPX2* OE line (also shown in Rebuttal Figure 1D)

Inducible *GPX2* overexpression in H1 hESC line generation: We apologize for this omission. The H1 doxycycline-inducible *GPX2* OE hESC line was generated using the same methodology

as the HUES8 OE HESC line, which is described in the Methods section. We have now clarified this explicitly in the revised Methods section.

Rationale for hPSC line selection: We selected H1 and HUES8 hESC lines because: (1) both are well-characterized hESC lines routinely used for pancreatic differentiation, (2) our laboratory has two decades of experience with these lines, ensuring reliable differentiation efficiency and growth dynamics, and (3) both lines have been validated for normal karyotype and routinely tested negative for mycoplasma contamination.

The consistent results across multiple independent clones, including loss- and gain-of-function approaches, and two different parental hESC lines confirm the reproducibility and robustness of the observed effects. We believe these additional data comprehensively address your concerns and strongly support our conclusions. We hope this clarification resolves your reservations.

The revised Results section now reads as follows: “*The resulting Gateway pENTR 1A plasmid containing the GPX2 cDNA sequence was subjected to an LR recombination reaction with plasmid PB-TA-ERP2*¹ (gift from Dr. Knut Woltjen, Addgene, USA, 80477), leading to the acquisition of the final plasmid (pENTR-hGPX2-FLAG plasmid) for establishing doxycycline-inducible GPX2 overexpression in HUES8-iCas9 and H1 hPSCs. Next, HUES8-iCas9 and H1 hPSCs were transfected with the pENTR-hGPX2-FLAG plasmid and pCMV-hyPB plasmid² (gift from Dr. Allan Bradley), carrying transposase enzyme using Lipofectamine 3000 kit (Thermo Fisher Scientific, Netherlands), following manufacturer’s instructions” and “Results” section has been revised as follows: “*To investigate GPX2 suppressive role in liver-like fate acquisition, we generated doxycycline-inducible GPX2 overexpression (OE) hPSC line on HUES8 and H1 background (Suppl. Fig. 7A-D).*”

Rebuttal Fig. 1. Additional clonal line data showing subsequent differentiation stages, GPX2 overexpression, and growth curves of WT and two hPSC clonal lines monitored by live imaging. (A) Dot plot showing the quantification of SOX17 mean fluorescence intensity (MFI) at the DE differentiation stage, normalized to DAPI, in WT (orange) and *GPX2* KO (KO1, KO2, KO3 - blue) spheroids. Analyzed KO spheroids were derived from three independent hPSC clonal lines (KO1, KO2, KO3). Each dot represents one image, with mean and 95% confidence intervals (CIs) indicated. $N = 3$ independent experiments. One-way ANOVA test was used to determine statistical significance. **(B)** Dot plot showing the quantification of PDX1 and SOX9 MFI, normalized to DAPI, at the pancreatic endoderm (PEN; day 8) differentiation stage in WT (orange), KO1 (blue) and KO2 (blue) spheroids. Each dot represents one image, with mean and 95% confidence intervals (CIs) indicated. $N = 3$ independent experiments. One-way ANOVA test was used to determine statistical significance. **(C)** Dot plot showing the quantification of PDX1, CHGA, and C-PEP MFI in WT (orange), KO1 (blue) and KO2 (blue) spheroids at the SC- β cell differentiation stage. Each dot represents one image. The median with 95% CI is shown on the graph. The p -values were calculated using one-way ANOVA test. $N = 3$ independent experiments. **(D)** Quantification of AFP and FGB MFI

in WT (orange) and OE (pink) spheroids derived from H1 and HUES8 hPSC lines shown as a dot plot. Each dot represents one image. The median with 95% CI is shown on the graph. The *p*-values were calculated using one-way ANOVA test. *N* = 3 independent experiments. (E) Representative growth curves of WT (red) and KO1 (green) and KO2 (blue) hPSCs monitored by live imaging during a 5.5-day period. Confluence values, normalized to the starting time point, showed no significant differences between WT, KO1 and KO2 hPSCs. *N* = 3 independent experiments.

2) The title of the paper seems misleading. "Gpx2 and oxidative stress as gatekeepers of pancreatic fate" specifically the term gatekeeper is ambiguous/non-scientific and seems to suggest that oxidative stress levels prevent pancreatic differentiation.

We thank the reviewer for this observation. Following your suggestion, we have revised the title to: "*Oxidative stress and GPX2 control pancreatic vs non-pancreatic cell fate in human endoderm*".

3) The new text states that 0.1 10 uM H2O2 is used to treat WT cells however the graph in the figure 3 (old 3M) that WT cells treated with 0.01 uM demonstrate an elevated *level* of Pdx1.

We apologize for this error and appreciate your careful assessment. MS, line 266 has been now updated to 0.01 μ M.

We have revised the Results section as follows: "*To further explore the role of oxidative stress, we treated WT and KO cells with varying concentrations of hydrogen peroxide (H₂O₂; 0.01–10 μ M) to induce oxidative stress, and selenium (Se; 1–1000 pM) to reduce it.*"

Reviewer #2 (Remarks to the Author):

The authors have added more data and provided further evidence. Overall, the revised manuscript has been improved. However, there are still some questions which need to be addressed:

1. For Western blotting experiments, all samples for comparison should be performed and presented in a single gel rather than in a separated manner.

Thank you. We have repeated all the timepoints in the same run and updated the WB in Fig 11 and source data. We have also included this panel below.

Fig. 11. Western blot analysis of GPX2 protein levels during hPSC pancreatic differentiation. Cell lysates were collected at various time points, as indicated. GAPDH was used as a loading control.

2. The FACS results should be quantified and statistically evaluated.

Thank you for your suggestion. Below we present the quantification and statistical analyses for the flow cytometry evaluation of i) pancreatic differentiation assessment (Suppl. Fig. 1B) and ii) HNF4A and FGB expression at different timepoints during differentiation for WT and two independent clonal KO lines (Rebuttal Fig. 4 and Suppl. Fig. 2H-I and 5G). Cytometry analyses of FGB⁺, HNF4A⁺, and FGB⁺/HNF4A⁺ cells at the PFG and PEN stages are presented in Supplementary Figures 2H–I, while results for FGB⁺, INS⁺, and FGB⁺/INS⁺ cells at the SC- β cell stage are shown in Supplementary Figure 5G. For both clones, we observed upregulation of FGB expression at SC- β cell stage, however, most INS⁺ cells did not co-express the FGB.

Suppl. Fig. 1B. Quantification of flow cytometry data during pancreatic differentiation at key developmental stages: pancreatic endoderm (PEN, day 8), pancreatic progenitor (PP, day 12), late endocrine progenitor (late EP, day 19), and SC- β cell stage (day 24). Expression and co-expression of stage-specific markers were analyzed as follows: day 8 – PDX1 and SOX9; day 12 – PDX1 and NKX6-1; day 19 – PDX1 and CHGA; day 24 – INS and NKX6-1. Co-expression of markers also indicated. Each dot represents one biological replicate; median values and 95% confidence intervals (CIs) are shown. $N = 3$ independent experiments.

Rebuttal Fig. 2. Stage-specific flow cytometry–based quantification of FGB, HNF4A, and INS expression during pancreatic differentiation. The percentages of FGB⁺, HNF4A⁺, and FGB⁺HNF4A⁺ double-positive cells are shown for the primitive foregut (PFG; **A**) and pancreatic endoderm (PEN; **B**) stages. Two *GPX2* KO clonal cell lines (KO1 and KO2) were analyzed in each experiment. Wild-type (WT) cells are shown in orange, and *GPX2* KO cells are presented in blue. Each dot represents one biological replicate; median values and 95% confidence intervals (CIs) are indicated. One-way ANOVA test was used to determine statistical significance. *N* = 3 independent experiments.

3. There is no GSIS result in the rebuttal Fig. 12. Additionally, the results of functional evaluation of β cells are important and should be added to the manuscript.

To assess β cell function following *GPX2* deletion, we performed glucose-stimulated insulin secretion (GSIS) assays. Control and *GPX2* KO β cells were sequentially exposed to low glucose (2 mM), high glucose (20 mM), exendin-4 (50 nM), and KCl (30 mM), and the levels of secreted insulin were measured after each treatment. Both WT and KO β cells exhibited a progressive increase in insulin secretion in response to the stimuli, demonstrating a normal secretory response (**Rebuttal Fig. 3A**). The insulin stimulation index was calculated by normalizing the insulin secretion after each stimulus to the basal secretion measured at 2 mM glucose (**Rebuttal Fig. 3B**). Our results show that the stimulation index increased with each successive treatment, confirming proper β cell functionality. However, no significant difference in insulin secretion or stimulation index was observed between SC-derived WT and *GPX2* KO β cells.

To further validate the lack of effect of *GPX2* loss on mature β cell function, we performed the same analysis in an independent cellular model. For this purpose, we generated *GPX2* knockdown (KD) EndoC- β H1 cells. EndoC- β H1 cells are an immortalized human β cell line commonly used for studies of insulin secretion and drug screening, derived from human fetal pancreas obtained after elective termination of pregnancy between 7 and 11 weeks gestation³. EndoC- β H1 cells were transduced with lentiviral particles carrying either *GPX2* shRNA or a control (GIPZ) construct. Successful *GPX2* knockdown was confirmed at the RNA and protein levels, showing a 47% decrease in mRNA expression and 44% reduction in protein abundance (**Rebuttal Fig. 3C-D**). Cells were treated with low (2.8 mM) and high (16.8 mM) glucose concentrations in Krebs buffer, and insulin secretion levels were measured following each treatment. Both *GPX2* KD EndoC- β H1 and control cells showed a stepwise increase in insulin release in response to glucose stimulation, indicating a normal secretory response (**Rebuttal Figure 3E**). Unexpectedly, GSIS analysis revealed that the insulin stimulation index was significantly higher in *GPX2* KD cells compared to controls (**Rebuttal Figure 3F**), contrasting with the results obtained from *GPX2* KO β cells. Given these contrasting outcomes, we conclude that further investigation is required to fully elucidate the mechanisms by which *GPX2* modulates β cell function and insulin release.

Rebuttal Fig. 3. Functional assessment of insulin secretion *in vitro* following *GPX2* loss. (A) hPSC-derived WT and *GPX2* KO β cells at day 45 of differentiation were treated with 2 mM glucose, 20 mM glucose, 50 nM exendin-4 or 30 mM KCl in Krebs buffer, and insulin concentration in the medium was assessed by ELISA. WT SC- β cells are shown in orange, and *GPX2* KO (clonal line KO1 and KO2) cells are presented in blue. Each dot represents an individual biological replicate (average of two technical replicates) normalized to the cell number. The median values and 95% confidence intervals (CIs) are indicated. Statistical significance was determined using two-way ANOVA test. $N = 7$ independent replicates. (B) The stimulation index for WT and *GPX* KO SC- β cells, defined as fold change in insulin secretion at 20 mM glucose, 50 nM exendin-4 and 30 mM KCl over the basal condition (2 mM glucose). WT cells are shown in orange, and *GPX2* KO cells are marked in blue. Each dot represents an individual biological replicate (average of two technical replicates) normalized to the cell number. The median values and 95% confidence intervals (CIs) are indicated. Statistical significance was determined using two-way ANOVA test. (C) Relative mRNA expression of *GPX2*. mRNA level was assessed by qPCR and normalized to GAPDH. (D) Western blot analysis validates the absence of *GPX2* protein in KD EndoC- β H1 cells. Quantification is presented on the right side of the membrane. Results of *SPOCK2* KD are also shown - not relevant to this study. (E) To assess glucose-stimulated insulin secretion, EndoC- β H1 control (CTRL, GIPZ) and *GPX2* KD cells were exposed to 2.8 mM (basal) or 16.8 mM (stimulatory) glucose in Krebs buffer, and insulin levels in the supernatant were quantified by ELISA. Control cells are shown in orange, and *GPX2* KD EndoC- β H1 cells are presented in blue. Each dot represents an individual replicate normalized to the cell number. The median values and 95% confidence intervals (CIs) are indicated. Statistical significance was determined using an unpaired t-test. $N = 3$ independent biological replicates. (F) The stimulation index for control and *GPX* KO EndoC- β H1 cells, defined as fold change in insulin secretion at 16.8 mM glucose over the basal condition (2.8 mM glucose). Control cells are shown in orange, and *GPX2* KO EndoC- β H1 cells are marked in blue. Each dot represents an individual replicate normalized to the cell number. The median values and 95% confidence intervals (CIs) are indicated. Statistical significance was determined using an unpaired t-test. $N = 3$ independent biological replicates.

4. *GPX2* KO notably affected the cell-fate choices during pancreatic differentiation, but at the SC- β cell stage (day 21) the authors did not observe significant changes in β cell differentiation efficiency and function between WT and *GPX2* KO cells. These observations are confusing, and need more explanation and discussion.

We thank the Reviewer for this important observation. We agree that this requires clarification and have added discussion to address this point. Based on all gathered data, we are convinced that the key to understanding of these observations lies in the temporal role of *GPX2* during differentiation. Our data demonstrate that *GPX2* functions primarily as a fate decision regulator at the posterior foregut (PFG) stage (day 6), rather than as a controller of subsequent β cell maturation. At day 8 (PEN stage), *GPX2* KO cells exhibit a mixed population with increased hepatic-like progenitors alongside pancreatic progenitors (Figs. 2-3), representing a critical fate decision window where oxidative stress levels determine lineage commitment. Importantly, this fate decision appears largely irreversible once established. By the SC- β cell stage (day 21), INS^+ cells in KO samples do not co-express hepatic markers such as FGB, demonstrating that cells

committed to the pancreatic lineage are a distinct population from those that adopted hepatic-like fate. This is further confirmed by scRNA-seq analysis, which reveals that hepatic-like and intestinal markers are enriched in a separate multilineage (ML) cluster in KO cells (Fig. 4), while SC- β clusters remain distinct. The PPs that successfully navigate the early fate decision despite GPX2 deficiency proceed through subsequent differentiation stages without major impairment. Our scRNA-seq at day 21 show that SC- β cell clusters in KO cells exhibit transcriptional profiles similar, yet not identical, to WT. This explains why we observe overall similar KO β cell function at day 21 as WT cells despite the profound effects on lineage choice at earlier stages. This stage-specific function is consistent with developmental principles, as many developmental regulators control lineage specification without affecting terminal differentiation. Once cells pass through the critical decision point and commit to a specific fate, subsequent maturation can proceed through compensatory mechanisms or GPX2-independent pathways. Thus, GPX2 acts as an early gatekeeper determining whether cells enter pancreatic lineage but does not significantly regulate β cell maturation once that fate is chosen.

The revised discussion now reads: *“Furthermore, GPX2 functions as a temporal gatekeeper of cell fate decisions during endoderm differentiation, with distinct stage-specific roles. While GPX2 deficiency affects lineage commitment at the PFG stage (days 6-8), leading to hepatic-like and multilineage progenitors, we observe comparable β cell differentiation efficiency at day 21. This suggests that oxidative stress plays different roles at distinct developmental stages, acting as a fate determinant during early specification but potentially serving beneficial functions during later maturation. Critically, INS⁺ KO cells at the SC- β stage do not co-express hepatic markers such as FGB, demonstrating that pancreatic and hepatic-like populations are distinct. Our scRNA-seq data further support this, showing hepatic-like markers enriched in a separate multilineage cluster while SC- β clusters in KO cells exhibit transcriptional profiles similar, yet not identical, to WT. These observations establish a stage-dependent model where GPX2 and oxidative stress critically determine whether cells enter the pancreatic versus non-pancreatic lineage during early specification, but do not impair β cell maturation once that fate is established. This temporal specificity identifies the PFG stage as the critical window when oxidative stress management is essential for pancreatic fate specification, with important implications for optimizing directed differentiation protocols”.*

Reviewer #3 (Remarks to the Author):

Thank you for your evaluations and contributions to our manuscript. We have improved our manuscript because of your review.

Reviewer #4 (Remarks to the Author):

Most of this reviewer’s comments have been addressed.

We appreciate the time and effort you invested in reviewing our manuscript. Your constructive feedback has led to meaningful improvements throughout the work.

1. Kim, S.-I. *et al.* Inducible Transgene Expression in Human iPS Cells Using Versatile All-in-One piggyBac Transposons. *Methods Mol Biol* **1357**, 111–131 (2016).
2. Yusa, K., Zhou, L., Li, M. A., Bradley, A. & Craig, N. L. A hyperactive piggyBac transposase for mammalian applications. *Proc Natl Acad Sci U S A* **108**, 1531–1536 (2011).
3. Ravassard, P. *et al.* A genetically engineered human pancreatic β cell line exhibiting glucose-inducible insulin secretion. *J Clin Invest* **121**, 3589–3597 (2011).